# From Euclidean to Graph-Structured Data:
# A Survey of Collaborative Learning

**Rémi Bourgerie**                                                                 *remibo@kth.se*
*School of Electrical Engineering and Computer Science, and Digital Futures*
*KTH Royal Institute of Technology*
*Stockholm, Sweden*

**Šarūnas Girdzijauskas**                                                         *sarunasg@kth.se*
*School of Electrical Engineering and Computer Science, and Digital Futures*
*KTH Royal Institute of Technology*
*Stockholm, Sweden*

**Viktoria Fodor**                                                                 *vfodor@kth.se*
*School of Electrical Engineering and Computer Science, and Digital Futures*
*KTH Royal Institute of Technology*
*Stockholm, Sweden*

**Reviewed on OpenReview:** *https://openreview.net/forum?id=vj9l8AjLT6&noteId=VC1pHzl5Pm*

## Abstract

The conventional approach to machine learning, that is, collecting data, training models, and performing inference in a single location, faces fundamental limitations, including scalability and privacy, that restrict its applicability. To address these challenges, recent research has explored *collaborative learning* approaches, including *federated learning* and *decentralized learning*, where individual agents perform training and inference locally, with limited collaboration. Most collaborative learning research focuses on Euclidean data with regular, grid-like structure (e.g., images, text). However, these approaches fail to capture the relational patterns in many real-world applications, best represented by graphs. Learning on graphs relies on message-passing mechanisms to propagate information between connected nodes, making it conceptually well-suited for collaborative environments where agents must exchange information. Yet, the opportunities and challenges of learning on graph-structured data in collaborative settings remain largely underexplored. This survey provides a comprehensive investigation of collaborative learning from Euclidean to graph-structured data, aiming to consolidate this emerging field. We begin by reviewing its foundational principles for Euclidean data, organizing them along three core dimensions: learning effectiveness, efficiency, and privacy preservation. We then extend the discussion to graph-structured data, introducing a taxonomy of graph distribution scenarios, characterizing associated statistical heterogeneities, and developing standardized problem formulations and algorithmic frameworks. Finally, we systematically identify open challenges and promising research directions. By bridging established techniques for Euclidean data with emerging methods for graph learning, our survey provides researchers and practitioners with a well-structured foundation of collaborative learning, supporting further development across a wide range of scientific and industrial fields. **Resources are available at https://github.com/remibourgerie/collaborative_gnns.**

# 1 Introduction

Over the past decade, Deep Learning has turned into a success story, largely due to the combination of four factors (Goodfellow et al., 2016): (i) neural networks as *universal* function approximators (Rosenblatt, 1958; Rumelhart et al., 1986) (ii) *sample efficient* training algorithms (Bottou, 2010), (iii) *parallelization* schemes that align with hardware architectures (Hooker, 2021); and (iv) the availability of *massive* amounts of high-quality data.

While the first three factors have been consolidated over the years, the fourth requirement, *access to massive datasets*, remains a major obstacle. Indeed, today data ecosystem remains highly fragmented, constrained by privacy legislation (European Union, 2016), data ownership concerns, integration complexity, and the inherent coordination difficulties of multi-agent systems. This access to data is partially addressed for LLMs by scaling on *public* web data (Naveed et al., 2023; Baack, 2024), but the fundamental challenge persists: the absence of mechanisms to collect, align, and utilize data across organizational and technical boundaries.

*Collaborative learning*, often termed under the umbrella of *federated* or *decentralized learning*, provides a framework to address this challenge by allowing multiple agents to train a shared model directly on their private datasets. In this framework, only model updates or aggregated information are exchanged between agents rather than entire datasets, drastically reducing the amount of data communicated. Moreover, by keeping data at its source, this framework holds the promise of privacy preservation.

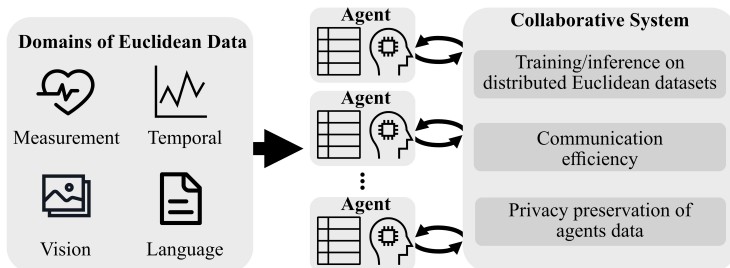

**(a) Collaborative Learning on Euclidean Data**

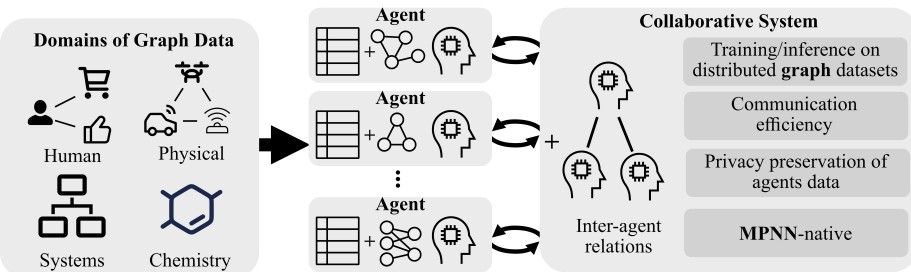

**(b) Collaborative Learning on Graph Data**

Figure 1: **Collaborative Learning from Euclidean to Graph Data.** Collaborative learning refers to a system of agents that collectively train and infer ML models on their local private data. Collaborative learning has been extensively studied for Euclidean data (a), comprising highly popular domains like vision, language, temporal sequences, and measurement data. Conversely, collaborative learning on graph data (b) is an emerging field. Graphs canonically represent data structured with relations, arising in domains including human behavior, physical systems, chemistry, and information systems. Compared to *centralized learning*, collaborative learning offers three main advantages: enabling training and inference on distributed datasets, communication efficiency, and the promise of privacy preservation of agents data. Collaborative learning on graph data aligns naturally with *Message Passing Neural Networks (MPNNs)*.

Collaborative learning solutions can be divided into two categories based on data structure: *Euclidean data*, which exhibits regular, grid-like structure (e.g., images as 2D grids, text as 1D sequences), and *non-Euclidean data*, where relationships are defined by irregular structures such as *graphs* (Bronstein et al., 2021), as illustrated in Figure 1.

For Euclidean data, typical use cases include cross-device scenarios involving numerous agents, each possessing limited data insufficient for generating accurate models independently. These agents may include mobile phones (Xu et al., 2023), autonomous vehicles (Hellström et al., 2022), mobile base stations (Zhang et al., 2022), or Internet of Things (IoT) devices (Zhou et al., 2023). Cross-silo applications involve large datasets held by distinct actors who cannot share data to preserve privacy, prevalent in healthcare and finance (Li et al., 2020). Collaboration typically involves exchanging and aggregating model parameters during training, while inference is generally performed locally by individual agents.

For graph-structured data, two distinct scenarios emerge. The first scenario is learning on a large collection of independent graph instances, typical in computational biology (e.g., predicting molecular properties) (He et al., 2022), involves learning parameters of a graph-based model (e.g., a GNN) using tools conceptually similar to those for Euclidean data, as each graph instance can be treated analogously to a single data sample. The second scenario arises when agents hold a portion of a *shared* global graph, both inference and learning necessitate information exchange among agents, as the graph topology encodes critical relationships spanning agent boundaries. Applications include collaborative autonomous agents in IoT systems (Liu et al., 2025), autonomous vehicles (Blumenkamp et al., 2022; Pan et al., 2023), and privacy-preserving learning over large-scale social network or recommender system graphs (He et al., 2021; Dong et al., 2023; Agrawal et al., 2024; Han et al., 2024).

While these diverse applications demonstrate broad potential, their implementation presents significant challenges that differ fundamentally between Euclidean and non-Euclidean settings. Modern machine learning (ML) relies predominantly on iterative optimization algorithms (Bottou et al., 2018), which have long posed difficulties for distributed systems (Bertsekas, 2011; Ram et al., 2009; Low et al., 2010; Xing et al., 2016). These challenges are exacerbated in collaborative learning due to the substantial size of the models that need to be communicated and inherent system discrepancies, driving active research in system architecture and algorithm design (Kairouz et al., 2021; Daly et al., 2024).

For graph-structured data, the challenges are further amplified by the non-Euclidean nature of the data. The graph topology is often captured by *Message Passing Neural Networks* (MPNNs) (Gilmer et al., 2017; Hamilton et al., 2017), where the nodes iteratively exchange and aggregate messages with their neighbors. These MPNNs are inherently distributed iterative algorithms, which makes them conceptually well-suited for collaborative settings. However, when each agent holds a partition of the shared global graph and cannot freely share data, the requirement for information to propagate through the graph topology creates novel challenges beyond those in Euclidean settings, leading to a rapidly expanding area of research.

## 1.1 The Objectives of the Survey and Related Work

While existing surveys have extensively covered *federated learning* on Euclidean data, and to a lesser extent on graph data, none have systematically analyzed their intersection. This survey bridges that gap by providing a unified view that maps the design space of collaborative learning from Euclidean to graph-structured data. We outline how different approaches relate to each other and identify research challenges that remain unanswered for the less understood case of graphs. Crucially, while methods for Euclidean data are well-established, graph-structured data introduces unique complexities that draw upon knowledge from multiple domains. Collaborative learning for graph data stands on three foundational pillars, borrowing from distinct research areas as illustrated in Figure 2:

- **Diffusion Algorithms on Networks** are distributed iterative methods that compute node-level or network-level functions through local message passing. These methods were initially developed for distributed decision problems in networked systems (e.g., multi-agent systems, sensor networks) (Tsitsiklis et al., 1986; Olfati-Saber et al., 2007), and later adopted in large-scale net-

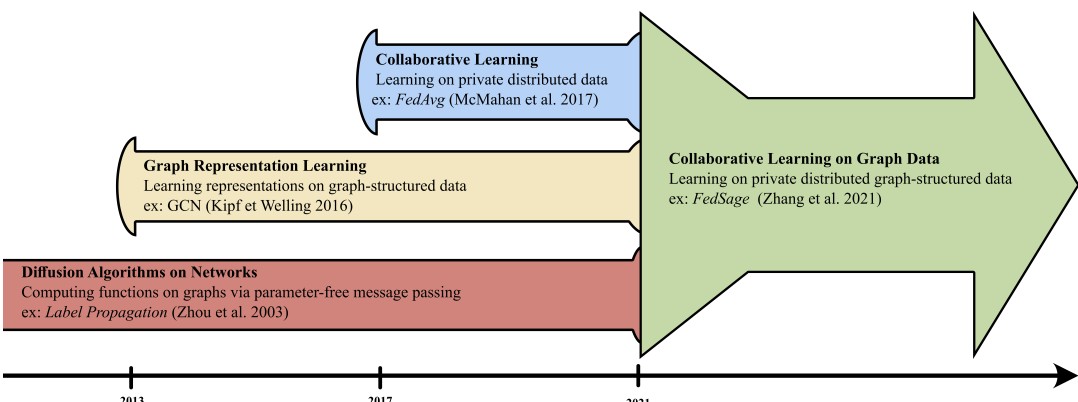

Figure 2: **Collaborative Learning on graph data and its three pillars.** Collaborative learning on graph data relies on three distinct pillars: diffusion algorithm on networks, graph representation learning, and collaborative learning.

work analytics and learning through closely related random-walk-based processes (e.g., ranking, recommendation, semi-supervised inference on graphs) (Zhou et al., 2003).

- **Graph Representation Learning** employs deep neural architectures to learn representations from graph-structured data (Kipf & Welling, 2017). Applications where relational structure is intrinsic (e.g., social networks, molecular modeling, infrastructure systems) have driven this area, which emerged at the intersection of graph signal processing and machine learning (Bruna et al., 2013).

- **Collaborative Learning**, for example *federated learning* (McMahan et al., 2017), allows training and inference on private distributed datasets through centralized or decentralized approaches. Collaborative learning is motivated in two key practical cases: when sensitive data cannot be shared between actors (healthcare, finance, user data) and when natively distributed data cannot be collected due to communication constraints (robotics, IoT). The supporting disciplines span machine learning, distributed optimization, information theory, and communication systems.

- **Collaborative learning on Graph Data** (Zhang et al., 2021) studies learning on private distributed graph-structured data. It intersects the three areas above, by combining the message-passing principles of diffusion algorithms on networks, the neural architectures of graph representation learning, and the distributed optimization frameworks of collaborative learning.

This survey connects these three domains to examine how established results can accelerate research in emerging fields and novel applications. Table 1 presents representative surveys and foundational papers across these pillars, providing comprehensive details that were necessarily excluded to maintain focus. The table also includes works on parallel processing of big data, an area with partial overlap with collaborative learning. To limit the scope of the survey, we focus on supervised learning. Interested readers may find references for collaborative unsupervised and self-supervised learning in (Uludag et al., 2025; Zhang et al., 2025; Ji et al., 2024), for multi-agent reinforcement learning in (Gronauer & Diepold, 2022; Chen et al., 2025), and for reinforcement learning on graph-structured data in (Nie et al., 2023; Liu et al., 2024).

## 1.2 The Organization of the Paper

In this survey, we provide a structured overview of learning from distributed private datasets, including the basic concepts, the research challenges, and proposed solutions, starting from classical collaborative learning, to learning over distributed private graph datasets. The skeleton of the paper is shown in Figure 3. In the first part of the paper, we discuss collaborative learning over Euclidean data, that is, classical centralized FL and its distributed counterparts. We formulate the learning problem, discuss how the data can be partitioned across the agents, and what heterogeneity aspects need to be considered. Then we discuss

| Topic | Representative Works |
|---|---|
| **Diffusion Algorithms on Networks** | |
| Distributed averaging and consensus | **(Bertsekas & Tsitsiklis, 2015; Olfati-Saber et al., 2007)** (Yang et al., 2019) |
| Label propagation | **(Zhou et al., 2003)** |
| **Graph Representation Learning** | |
| Graph Neural Architectures | **(Kipf & Welling, 2017)** (Wu et al., 2020) |
| Graph Foundation Models | (Wang et al., 2025) |
| **Collaborative Learning on Euclidean Data** | |
| Federated Learning | **(McMahan et al., 2017)** (Kairouz et al., 2021), (Zhang et al., 2021; Yang et al., 2019; Liu et al., 2022) |
| Vertical Federated Learning | (Liu et al., 2024) |
| Data heterogeneity and personalization | (Zhu et al., 2021; Tan et al., 2022; Liu et al., 2024) |
| Privacy and security | (Lyu et al., 2020; Daly et al., 2024) |
| Parallel training | (Xing et al., 2016) |
| Machine Learning in wireless networks | (Hellström et al., 2022) |
| **Collaborative Learning on Graph Data** | |
| Federated Learning with GNNs | **(Zhang et al., 2021)** (He et al., 2021; Zhang et al., 2021) (Liu et al., 2025; Fu et al., 2024) |
| Applications and use cases | (He et al., 2021; Dong et al., 2023) |
| GNNs in wireless communications | (He et al., 2021; Lee et al., 2022) |
| Parallel training for large GNNs | (Shao et al., 2024; Lin et al., 2023; Liu et al., 2022) |

Table 1: **Key surveys and seminal works (in bold) on collaborative learning on graph data**, including the three pillars of *Collaborative Learning*, *Diffusion algorithms on Networks*, and *Graph Representation Learning*.

proposed solutions (i) for effective learning, that is, to achieve models with high accuracy despite the various forms of heterogeneity, (ii) for efficiency, in terms of the use of communication and computation resources, and finally, (iii) solutions for privacy preservation. Since collaborative learning over Euclidean data is an established research area, we review the key contributions in this part.

In the second part of the paper, we follow the same structure to discuss the emerging research field of collaborative learning on graph-structured data. The ways data (or information) can be distributed are significantly more varied now than in the Euclidean case, so we introduce a taxonomy for data partition. Then, we follow the structure of the first part to find relevant research results and questions that are still open for the design of collaborative learning systems on graph data. This part is based on a rigorous survey of all recent papers with keywords *federated graph* and *decentralized graph learning*. The taxonomy of solutions for effective, efficient, and privacy-preserving collaborative learning for Euclidean and graph data is provided in Appendix A, together with the notations and the glossary of key terminology used throughout this survey.

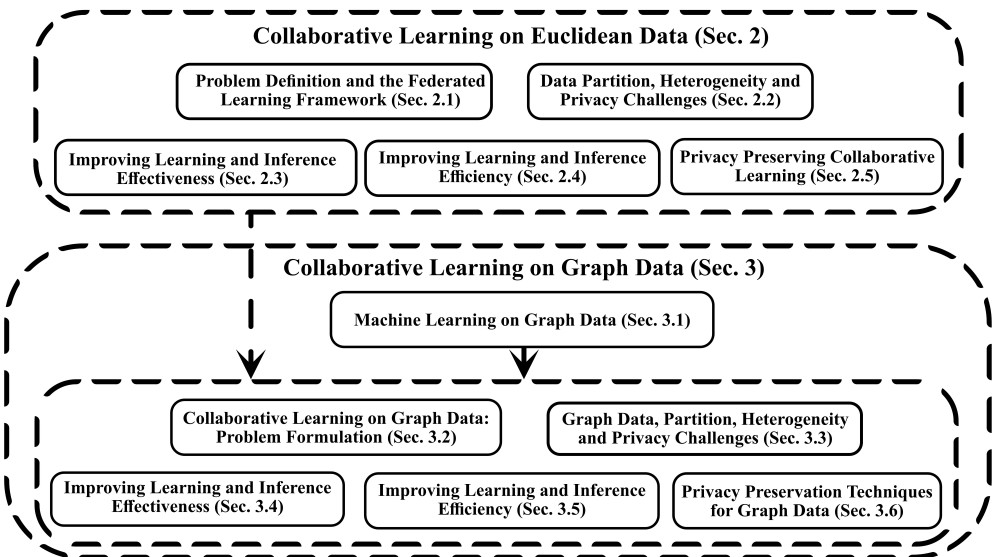

Figure 3: **The structure of the survey**. Our contributions reside in extending and mapping the concepts found for collaborative learning on Euclidean data to collaborative learning on graph data, thereby consolidating insights across these separate research areas. Our analysis of the design choices focuses on three directions: improving learning and inference *effectiveness*, improving learning and inference *efficiency*, and *privacy*-preservation techniques.

## 2 Collaborative Learning on Euclidean Data

The concept of *collaborative learning* emerges from multiple taxonomies, which are all cast under the umbrella term of *distributed machine learning* as outlined in Figure 4. The term *distributed machine learning* originally

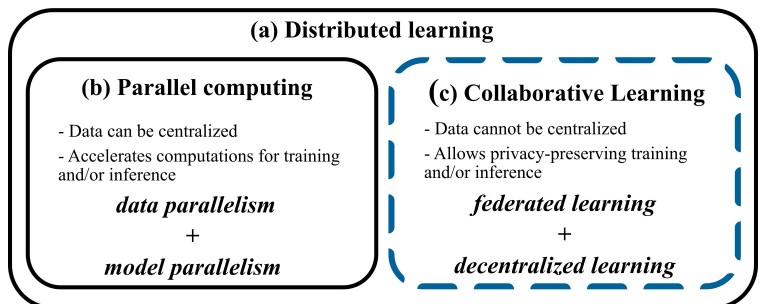

Figure 4: **Taxonomies of distributed learning and the scope of our survey**. *Distributed learning* (a) is often used as an umbrella term encompassing various scenarios and techniques. In particular, *parallel computing* (b) aims to accelerate training and/or inference under the assumption that the data can be centralized, and includes both *data-parallelism* and *model-parallelism* approaches. Our focus, however, is on a different scenario that we term *collaborative learning* (c), which addresses cases where the data remains distributed and private to its owner, covering both *federated learning* and *decentralized learning*.

served as a broad label for methods designed to train and/or infer ML models on massive datasets by distributing computations across multiple machines, often in a cluster (Xing et al., 2016; Kairouz et al., 2021). Distribution is then largely associated with parallelization, combining *data parallelism*, where datasets are divided among machines, and *model parallelism*, where parts of the model are divided among machines and updated in parallel.

The term *collaborative learning* refers specifically to scenarios where local data remains strictly private to its owner. For clarity, we assume that local data is generated locally at multiple agents, where it persists throughout training and inference phases while remaining exclusively accessible to the data owner, a case commonly termed *federated learning* or *decentralized learning* in the literature. This privacy constraint necessitates precise problem formulation and distinct design methodologies to address real-world implementation challenges, which we develop in this section. Our focus here is on *Euclidean data*, defined as data where each sample is represented as a feature vector in an Euclidean space (in practice, some $\mathbb{R}^d$ with the standard inner product).

## 2.1 Collaborative Learning: Problem Formulation and the Federated Learning Framework

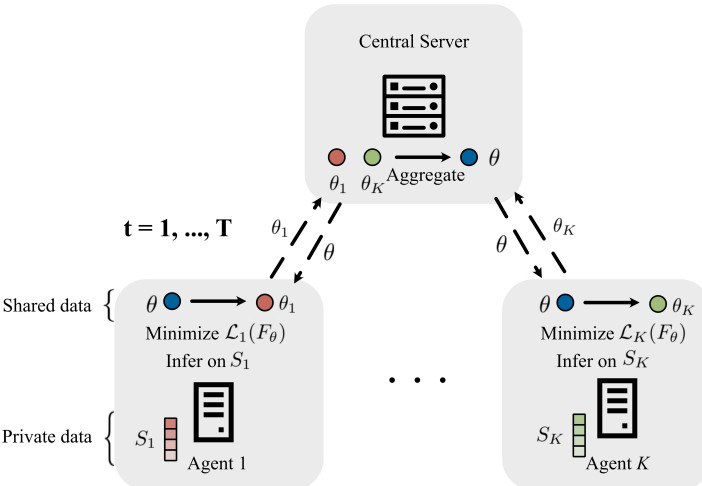

Figure 5: *Federated learning*, the basic approach to collaborative learning represented here for $K$ agents. Agents possess private local datasets $S_k$. In each round $t$, they train their local model $F_\theta$ on their local loss and communicate the model parameter updates $\theta_k(t)$. The central server aggregates in turn the local models into a global model with parameters $\theta(t)$, which is then shared again with the agents for another round $t+1$.

**Problem definition.** Consider a set of agents, denoted by $\mathcal{K} = \{1, \ldots, K\}$, as illustrated in Figure 5, where the data of each agent is sampled from an unknown distribution $\mathcal{D}_k$ on $\mathcal{X} \times \mathcal{Y}$ where $\mathcal{X}$ is the feature Euclidean domain and $\mathcal{Y}$ the label domain. Given a loss function $\ell : \mathcal{X} \times \mathcal{Y} \to \mathbb{R}^+$, each agent seeks to learn a mapping $F : \mathcal{X} \to \mathcal{Y}$ among a class of function $\mathcal{F}$ that minimizes the local population loss

$$\mathcal{L}_k(F) := \mathbb{E}_{(x,y) \sim \mathcal{D}_k} \ell(F(x), y). \tag{1}$$

If the data distributions $\mathcal{D}_k$ were known, agents could determine an optimal solution $F_k^* \in \mathcal{F}$, by directly minimizing their respective local loss (1) on $\mathcal{F}$. However, in practice, the true distributions $\mathcal{D}_k$ are unknown. Instead, agents rely on empirical samples forming local datasets $S_k$, drawn i.i.d. from $\mathcal{D}_k$. These datasets allow agents to construct a surrogate $\hat{\mathcal{L}}_k$, which represents an empirical approximation of the true loss $\mathcal{L}_k$ based on their local data only

$$\hat{\mathcal{L}}_k(F) := \frac{1}{|S_k|} \sum_{(x,y) \in S_k} \ell(F(x), y). \tag{2}$$

This approach provides a local estimate $\hat{F}_k^* \in \mathcal{F}$. However, the local datasets $S_k$ are typically too small to accurately approximate the underlying distribution $\mathcal{D}_k$. As a result, the local estimate $\hat{F}_k^*$ poorly approximates the true optimal function $F_k^*$, leading to high estimation error (Valiant, 1984; Bottou & Bousquet, 2007). To address this issue, a key assumption is that although the distributions $\mathcal{D}_k$ may differ across agents, they

share strong similarities. Therefore, collaboration among agents enables them to benefit from the aggregated knowledge contained in the union of local datasets $\{S_k\}_{k \in \mathcal{K}}$.

In the most favorable case, when the local distributions $D_k$ are the same, an effective solution is to collaborate on building **a single model common to all agents**. The problem can then be framed as identifying $F^* \in \mathcal{F}$ that minimizes the average of local population losses among all agents

$$\mathcal{L}(F) := \frac{1}{K} \sum_{k=1}^{K} \mathcal{L}_k(F). \tag{3}$$

Since the population losses $\mathcal{L}_k$ are generally uncomputable, (3) is typically approximated using local empirical losses. Most existing work focuses on minimizing (4), which is computationally feasible, rather than the true loss

$$\hat{\mathcal{L}}(F) := \frac{1}{K} \sum_{k=1}^{K} \hat{\mathcal{L}}_k(F). \tag{4}$$

This line of work typically assumes that $\hat{F}^* \approx F^*$, that is, minimizing (4) provides a good approximation of minimizing (3). This assumption holds only when the generalization gap $|\mathcal{L}(\hat{F}^*) - \hat{\mathcal{L}}(\hat{F}^*)|$ is negligible (Bottou et al., 2018).

To minimize (4), it can be convenient to parametrize $F_\theta$ with $\theta \in \Theta$ where $\Theta$ is the domain parameter. Typically, $F$ can be a neural network, parametrized by its learnable weights $\theta$. The problem can be framed as identifying $\theta^*$, a solution of

$$\min_{\theta \in \Theta} \hat{\mathcal{L}}(F), \text{ or equivalently, } \quad \min_{\theta \in \Theta} \frac{1}{K} \sum_{k=1}^{K} \hat{\mathcal{L}}_k(F_\theta). \tag{5}$$

**Federated learning framework.** The basic approach for collaborative learning on distributed datasets is the *Federated Learning (FL)* framework (Konečný et al., 2016; McMahan et al., 2017; Kairouz et al., 2021). The key idea of federated learning is to *federate* (McMahan et al., 2017) the training of agents through a common round-based protocol coordinated by a central server, as shown in Figure 5. At the end of training, each agent receives the jointly trained model, which can be used to perform inference independently on local data.

**Federated Averaging (FedAvg) protocol.** The most popular and simplest implementation of FL is *Federated Averaging (FedAvg)* (McMahan et al., 2017) detailed in Algorithm 1. It proceeds in $T$ communication rounds between a leader, referred to as the server, and the agents. At the beginning of each round $t$, the server randomly selects $K_{\max}$ agents (line 2) and communicates the latest global model parameters $\theta(t-1)$ to them (line 3). Upon reception, each selected agent initializes its temporary local model $\theta_k(t - \frac{1}{2})$ to the received global model $\theta(t-1)$. Then, each agent $k$ performs $E$ local update steps (7) on its local dataset $S_k$ to minimize its empirical loss (2) using the *Stochastic Gradient Descent (SGD)* algorithm (Robbins & Monro, 1951). After completing the local updates, each agent sends its resulting local model $\theta_k(t)$ back to the server (line 10). The server then aggregates all received models to compute the updated global model $\theta(t)$ via averaging, denoted by $\bar{\theta}(t)$ (11). Under specific conditions, this iterative process converges to $\theta^*$ a solution of (5) (Yu et al., 2019; Stich, 2019).

---

**Algorithm 1** Federated Averaging (FedAvg) adapted from (McMahan et al., 2017)

---

**Require:** Set of agents $\mathcal{K} = \{1, \ldots, K\}$, local datasets $S_k$, total number of communication rounds $T$, number of local training iterations $E$, number of sampled clients $K_{max}$, initial global model $\theta(0)$, learning rate $\eta$

1: **for** each round $t = 1, \ldots, T$ **do**

    Server:

2:      Sample uniformly a random subset $\mathcal{K}(t) \subseteq \mathcal{K}$ of $m$ clients ($|\mathcal{K}(t)| = \mathcal{K}_{max}$)

3:      Distribute global model parameters $\theta(t-1)$ to all sampled clients $k \in \mathcal{K}_t$

4:      **for** each agent $k \in \mathcal{K}(t)$ **in parallel do**

5:          Upon reception of $\theta(t-1)$, initialize local model parameter $\theta_k(t - \frac{1}{2}) = \theta(t-1)$

6:          **for** $e = 1, \ldots, E$ **do**

7:              Update local model using gradient descent [1]:

$$\theta_k(t - \frac{1}{2}) \leftarrow \theta_k(t - \frac{1}{2}) - \eta \nabla \left( \sum_{(x,y) \in S_k} \ell(F_{\theta_k(t-\frac{1}{2})}(x), y) \right)$$

8:          **end for**

9:      **end for**

10:      Communicate local model $\theta_k(t) = \theta_k(t - \frac{1}{2})$ to the server

    Server:

11:      Upon reception of models $\{\theta_k(t)\}_{k \in \mathcal{K}(t)}$, average local models:

$$\bar{\theta}(t) = \sum_{k \in \mathcal{K}(t)} \frac{n_k}{n} \theta_k(t)$$

12:      Set global model: $\theta(t) \leftarrow \bar{\theta}(t)$

13: **end for**

14: **return** Final global model $\theta(T)$

---

## 2.2 Data Partition, Heterogeneity and Privacy Challenges

The basic approach for collaborative learning, as defined in *FedAvg* (Algorithm 1), builds on several unrealistic assumptions. In the sequel, we discuss the characteristics of collaborative learning that need to be considered to address realistic use cases, that is, **data partition**, **statistical imbalance**, **system heterogeneity**, and **privacy vulnerabilities**.

### 2.2.1 Data Partition

The first deviation from the ideal assumptions outlined in Section 2.1 concerns data-related aspects, notably the partitioning of data across agents and the resulting statistical imbalance.

FL assumes a significant shared knowledge among the agents through the homogeneity of the local datasets. However, in practice, local datasets $\{S_k\}_{k \in \mathcal{K}}$ often vary significantly across the agents, with heterogeneity arising from the partition of the data or statistical imbalance. An example of these partitions for two agents is shown in Figure 6, with a summary provided in Table 2. In the native scenario (Section 2.1), agents collect similar types of features and labels, represented by a common feature and label spaces $\mathcal{X} \times \mathcal{Y}$. However, inconsistency in the features and labels collected is frequent in data collection practices. While these challenges have long been recognized in centralized ML (Pan & Yang, 2009; Zhu, 2005), they are amplified in collaborative settings where the dataset $S$, that would ideally be centrally available, is instead partitioned across agents in local datasets $\{S_k\}_{k \in \mathcal{K}}$ which stay private to each agent. To formalize, we

---

[1]In practice and as proposed in the original algorithm(McMahan et al., 2017), the dataset $S_k$ is divided into batches of fixed size and the model is updated with the gradient on each batch of data, elided here for simplicity.

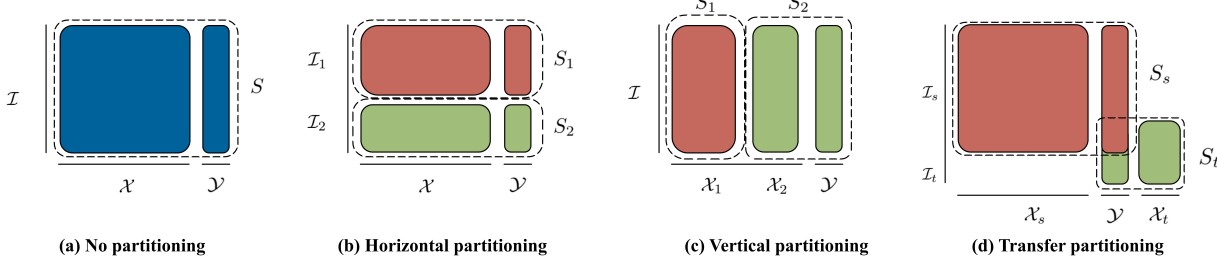

Figure 6: The different partitions of Euclidean data for a two-agent case, where $S_k$ denotes the dataset of agent $k$. Compared to *no partitioning* (a), *horizontal partitioning* (b) splits the sample space $\mathcal{I}$, *vertical partitioning* (c) splits the feature space $\mathcal{X}$, and *transfer partitioning* (d) splits both, with the source agent $s$ holding most of the data.

assume the existence of a global dataset

$$S = \left\{(x^i, y^i)\right\}_{i \in \mathcal{I}}.$$

where $\mathcal{I}$ is the global sample index, and each agent $k \in \mathcal{K}$ holds a private partition $S_k$ of $S$. Different partitions exist for $S$:

- *Horizontal partition.* Under horizontal partition, the index $\mathcal{I}$ of $S$ is partitioned across agents into local indexes $\mathcal{I}_k \subset \mathcal{I}$ such as $\cup_{k \in \mathcal{K}} \mathcal{I}_k = \mathcal{I}$

$$S_k = \left\{(x^i, y^i)\right\}_{i \in \mathcal{I}_k}.$$

  Therefore, agents possess data about distinct samples indexed by $\mathcal{I}_k$, but with the same feature $\mathcal{X}$ and label space $\mathcal{Y}$. This is the default setting of FL presented in Algorithm 1, and the most studied partition (Kairouz et al., 2021) in the FL literature. This partition further distinguishes *cross-device* scenarios, where millions of agents with limited data (e.g., mobile devices) collaborate, from *cross-silo* scenarios, where a few institutions holding vast amounts of data "silos" (e.g., banks, hospitals, e-commerce) collaborate.

- *Vertical partition.* Under vertical partitions, the domains $\mathcal{X}$ and $\mathcal{Y}$, usually some Euclidean vectorial spaces, are partitioned across agents into local feature $\mathcal{X}_k$ and label $\mathcal{Y}_k$ domains such that $\mathcal{X} = \oplus_{k \in \mathcal{K}} \mathcal{X}_k$ and $\mathcal{Y} = \oplus_{k \in \mathcal{K}} \mathcal{Y}_k$. Consequently, each sample $(x^i, y^i)$ is partitioned into $(x_k^i, y_k^i) \in \mathcal{X}_k \times \mathcal{Y}_k$ to form the local datasets

$$S_k = \left\{(x_k^i, y_k^i)\right\}_{i \in \mathcal{I}}.$$

  Therefore, agents observe the same samples but with distinct views, as represented by different feature and label domains (Liu et al., 2024). This naturally arises in multi-modal scenarios where agents collect features and labels specific to their modality (e.g., a hospital will diagnose respiratory diseases using lung radiography techniques, while another will use respiratory tests)

  A special case of vertical partition is the *transfer partition*. It occurs when one or more *source* agents (e.g., agent $s \in \mathcal{K}$) hold a dataset from one feature space $\mathcal{X}_s$, while another *target* agent (e.g., agent $t \in \mathcal{K}$) holds a dataset from a different feature space $\mathcal{X}_t$. For simplicity, we describe the case involving two agents $s$ and $t$

$$\forall k \in \{s, t\}, \quad S_k = \left\{(x^i, y^i) \in \mathcal{X}_k \times \mathcal{Y}_k\right\}_{i \in \mathcal{I}_k}.$$

  Despite differences in feature domains, the agents may share a subset of common samples

$$\forall k \in \{s, t\}, \quad \left\{(x_k^i, y_k^i)\right\}_{i \in \mathcal{I}_s \cap \mathcal{I}_t} \subset S_k.$$

  These act as anchor points for transferring information between agents. Transfer partitions often involve significant data imbalance, where $|\mathcal{I}_s| \gg |\mathcal{I}_t|$, consistent with the idea of "transferring"

substantial knowledge from $s$ to $t$. This setting frequently arises when organizations employ different data acquisition modalities but share overlapping entities. For example, a hospital with a large archive of traditional diagnostic records may support another institution, using a small set of shared patient samples.

### 2.2.2 Statistical Imbalance

In the native model, local datasets $S_k$ are assumed to be of the same size and sampled from a shared distribution $\mathcal{D}$ on $\mathcal{X} \times \mathcal{Y}$. However, in practice, the statistical properties of local datasets can vary significantly from one agent to another, and may evolve over time as agents join or leave the system, leading to different modes of imbalance (Kairouz et al., 2021; Paulik et al., 2021).

**Quantity imbalance.**   The most obvious phenomenon arises from *quantity imbalance* ($|\mathcal{S}_k| \gg |\mathcal{S}_l|$), where the number of samples collected locally varies significantly across agents. For example, a small clinic might manage thousands of patient records, while a multinational institution handles millions. This imbalance represents a natural phenomenon observed across many fields (Nisonger, 2008): most of the world's data is concentrated among a minority of data holders. The significant imbalance has given rise to different regimes known as *cross-device*, where millions of agents usually corresponding to users devices (e.g., a phone) collaborate, and *cross-silo*, where a few agents holding vast silos of data (e.g. an hospital) collaborate.

**Distributional shift.**   In addition, statistical imbalance in local datasets can arise from the distribution itself, where the datasets are drawn. Factors like population bias, dynamic environments, multiple modalities, and varying labeling standards can influence a shift in the marginal distribution $\mathcal{D}_k(x, y)$ from one agent to another, known as *distributional shift* (Moreno-Torres et al., 2012; Quiñonero-Candela et al., 2022; Kairouz et al., 2021; Zhu et al., 2021). We distinguish:

- *Covariate shift* occurs when the feature $\mathcal{D}_k(x)$ or label distribution $\mathcal{D}_k(y)$ observed locally, differ across agents but the concept $\mathcal{D}(x \mid y)$ or $\mathcal{D}(y \mid x)$ remains common. Covariate shift is a ubiquitous phenomenon in statistics dating back to early statistical studies (Snow, 1855) and is the most studied case for FL (Kairouz et al., 2021; Zhu et al., 2021; McMahan et al., 2017; Hsieh et al., 2020; Li et al., 2022). This often arises from population selection bias or non-stationary environments (Moreno-Torres et al., 2012). This distributional shift can be observed from two different perspectives. Firstly, differences in observed label distribution $\mathcal{D}_k(y)$, known as *label shift*, lead to to differences in observed features distribution $\mathcal{D}_k(x) = \mathcal{D}(x \mid y).\mathcal{D}_k(y)$. For example, a hospital located in a polluted area records a higher frequency of respiratory disease diagnoses compared to one in a clean environment. Similarly, a difference in observed feature distributions $D_k(x)$, known as *feature shift*, leads to a difference in label distribution $\mathcal{D}_k(x) = \mathcal{D}(y \mid x).\mathcal{D}_k(x)$. For example, one hospital focuses on diagnosing a young population while another focuses on an older population, leading to differences in features collected, yet the diagnostic knowledge remains the same.

- *Concept shift* occurs when the when the conditional distribution $\mathcal{D}_k(y \mid x)$ (or $\mathcal{D}_k(x \mid y)$) varies across agents. This corresponds to cases where the understanding of the concept varies from one agent to another. For example, the standards used for diagnosing hypertension from blood pressure differ between hospitals. This scenario is rather rare (Moreno-Torres et al., 2012), and challenging for most applications that assume shared mapping $F \in \mathcal{F}$ from features to labels (Huang et al., 2021).

### 2.2.3 System Heterogeneity

The native forms of FL, such as *FedAvg* in Algorithm 1, work best if the capabilities and tasks of the agents are similar. In practice, real-world systems include a variety of devices, networks, and tasks, and these system heterogeneities affect the learning and inference performance. We briefly characterize the main types of system heterogeneity that impact collaborative training and inference.

**Device and network constraints.** Collaborative learning systems must accommodate agents with vastly different hardware and network capabilities. These constraints manifest in two primary dimensions: computational resources and communication infrastructure.

- *Computational heterogeneity.* Agent computations are performed on diverse physical machines ranging from IoT sensors and mobile devices to telecom base stations, cloud containers, powerful servers, or even HPC clusters. This diversity creates order-of-magnitude differences in the time required to calculate local model updates (Xie et al., 2020; Even et al., 2024), and in the model sizes the devices can hold. Computing capabilities may also fluctuate over time due to parallel workloads, varying energy availability, or *quantity shifts*.

- *Communication heterogeneity.* Agents employ different network technologies such as fiber, WiFi, 5G, and low-power IoT protocols. These technologies determine both the *communication topology* (typically a star configuration with a central base station or a mesh with direct peer-to-peer links) and the *achievable bitrate*, which directly affects transmission delays (Hellström et al., 2022; Wang et al., 2020). For wireless communication, interference and noise introduce temporal variations that degrade link reliability. These network characteristics evolve dynamically as agents join, leave, or move through the environment.

These computational and communication heterogeneities create challenges at multiple levels. At the individual level, capability variations affect resource consumption such as energy cost, computation time, and communication overhead during collaborative learning (Wang et al., 2022; Meng et al., 2025). At the system level, slower agents, known as *stragglers*, can become bottlenecks that delay the entire learning process or introduce biases affecting all participants (Li et al., 2022). Dynamic populations present additional complications: when agents join or leave during training, the system must initialize newcomers and address potential distributional shifts and novel label classes.

**Task heterogeneity.** The native model assumes that all agents share the same task, represented as a common mapping from the feature domain $\mathcal{X}$ to the label domain $\mathcal{Y}$. In practice, however, agents may use their data to perform different tasks, thereby optimizing distinct mappings (Tan et al., 2022). For example, different hospitals may use the same radiography images to diagnose different diseases, resulting in heterogeneous local losses $\mathcal{L}_k$. Another case of task heterogeneity emerges when models are shared among applications with compound ML models, as exemplified by certain Android implementations (Huang et al., 2021). Furthermore, agents may employ different model architectures $\mathcal{F}$ due to limitations in memory or computational resources (Park & Joe-Wong, 2024).

### 2.2.4 Privacy Vulnerabilities

Collaborative learning techniques were initially motivated by their potential to enable privacy-preserving training between agents (McMahan et al., 2017; Bonawitz et al., 2017), making them well-suited for applications where local data cannot be pooled for regulatory reasons (Yang et al., 2018; European Union, 2016). The design of the original *FedAvg* algorithm (Algorithm 1) supports this motivation, since it ensures minimal data collection and anonymization (Bonawitz et al., 2022; Daly et al., 2024): local datasets $S_k$ remain with the agents while only model parameters $\theta_k(t)$ or gradients are communicated (line 10), limiting what the server learns about the underlying data; and communicated parameters are immediately aggregated into $\bar{\theta}(t)$ (line 11) before broadcast (line 2), preventing other agents from directly accessing individual contributions. Despite these design choices, collaborative learning does not provide inherent privacy guarantees. Even in the presence of *honest-but-curious* agents or server, that is, entities that follow the protocol correctly while attempting to infer sensitive information, collaborative learning in its native form suffers from information leakage at multiple levels:

- *The fully trained model parameters $\theta(T)$* are subject to *model inversion attacks* that infer training labels (Fredrikson et al., 2015) or determine membership of specific samples in the training data (Shokri et al., 2017).

- *Local model updates $(\theta_k(t) - \theta(t-1))$,* or equivalently gradients, sent to the server (Phong et al., 2017) are vulnerable to *gradient inversion attacks* that infer sample properties (Melis et al., 2019) or achieve pixel-perfect data reconstruction through gradient matching (Zhu et al., 2019; Geiping et al., 2020; Zhao et al., 2020). While other agents can also attempt gradient inversion upon receiving the aggregated model $\bar{\theta}(t)$, these attacks degrade significantly with the number of agents and batch size due to aggregation.

- Observing *the evolution of the aggregated model* $\{\bar{\theta}(t)\}_t$ over training enables attacks beyond single-round inference. Passive observation reveals temporal information, such as when samples with specific properties first appear in the training data (Melis et al., 2019). More advanced threat scenarios involve agents actively manipulating their contributions to extract information about others' training data (Melis et al., 2019; Hitaj et al., 2017).

**The Collaborative Learning Trilemma**

Maintaining effective and efficient learning under the privacy requirements of collaborative learning (Section 2.2.4), while simultaneously accommodating data and system heterogeneities (Sections 2.2.1–2.2.3), poses significant challenges. Specifically, collaborative learning must balance three competing objectives: *effective learning* (accurate model predictions), *efficient learning* (constrained communication and computation), and *privacy preservation* (anonymizing agents data). These objectives create fundamental tensions: effective learning requires extensive information exchange and complex computations that challenge efficiency, while privacy preservation requires cryptographic techniques and noise injection that decrease efficiency and effectiveness, respectively. These trade-offs form the *trilemma* illustrated in Figure 7, which serves as the foundation for analyzing the design choices in the following subsections.

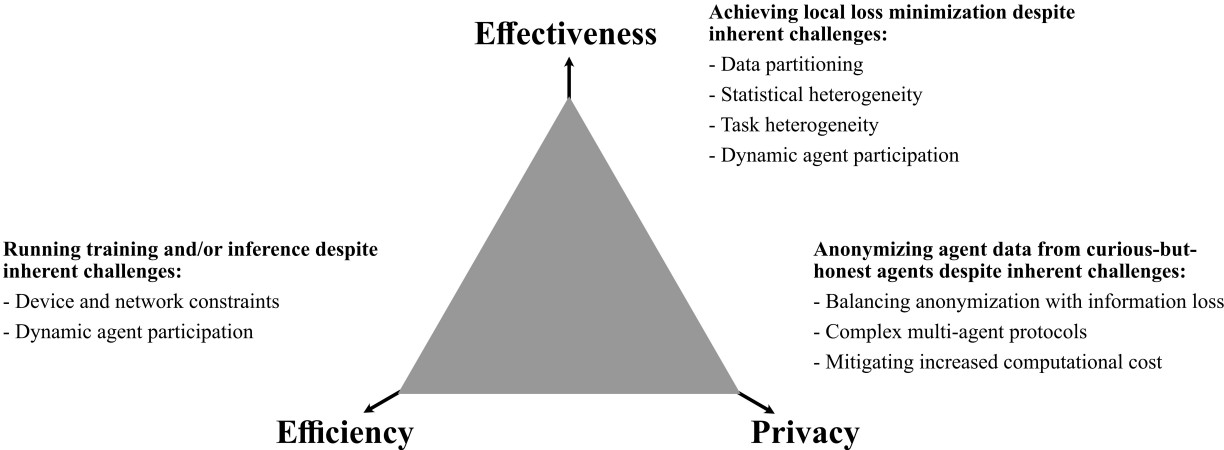

Figure 7: **The trilemma of collaborative learning:** balancing effectiveness (Section 2.3), efficiency (Section 2.3) and privacy (Section 2.5).

## 2.3 Improving Learning Effectiveness

Learning algorithms are effective if the generated models manage to minimize the learning loss of the agents, according to (1). The native proposition of collaborative learning (Algorithm 1) offers strong theoretical guarantees (Yu et al., 2019; Stich, 2019) to convergence to the solution of (5), particularly when the local datasets $\mathcal{D}_{k \in \mathcal{K}}$ are i.i.d. However, real-world deployments often diverge from this ideal case, as outlined in Section 2.1.

In such circumstances, the commonly used *FedAvg* algorithm becomes suboptimal to achieve adequate minimization of local objectives. This motivates the exploration of alternative strategies aimed at **improving**

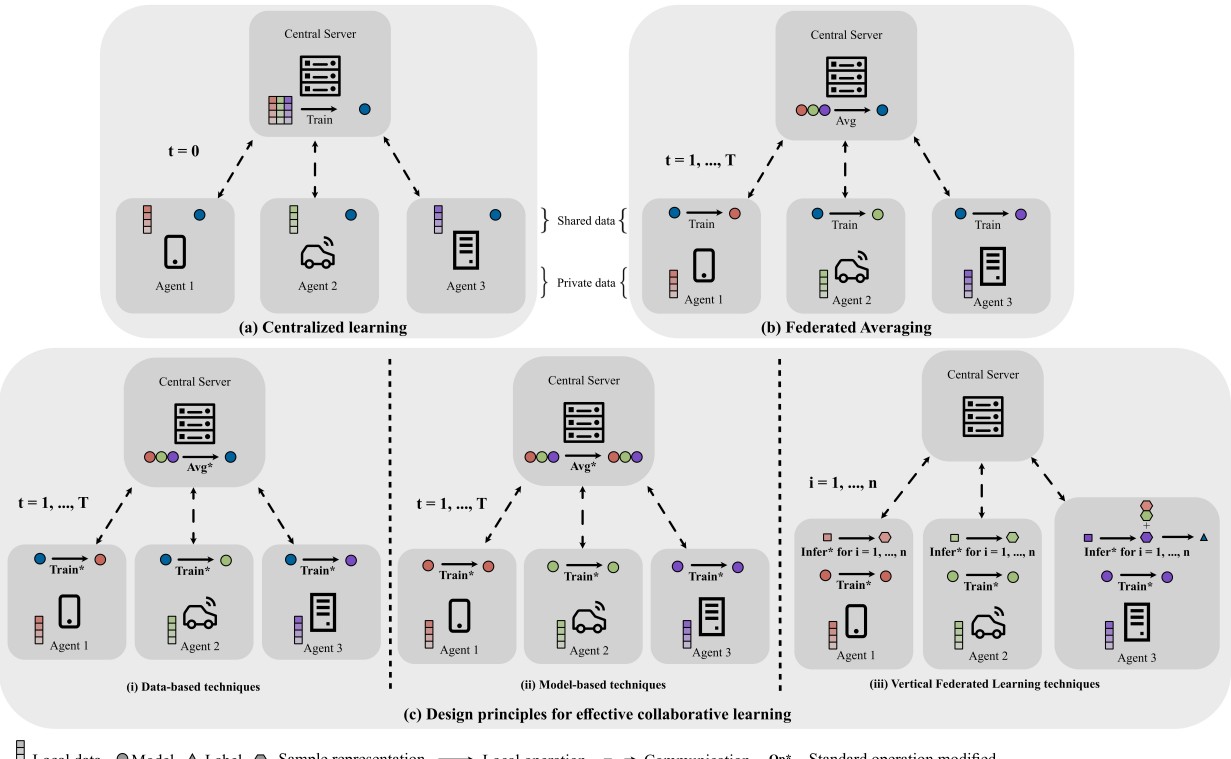

Figure 8: **Variations of federated learning that aim at improving the learning effectiveness.** In contrast to *centralized learning* (a), where data is collected from agents and training occurs on a central server, (b) *Federated Averaging* represents the baseline approach for collaborative learning systems. Methods to improve effectiveness (c) focus on *data-based* (i) and *model-based* (ii) techniques, typically applied to horizontal partitions. An alternative to the baseline is *Vertical Federated Learning (VFL) (iii)* where the inference becomes *collaborative*.

the effectiveness of the models used by agents to minimize their respective local objectives, that is, **data-based**, **model-based technique** and **learning under vertical partition**, as discussed in the following sections and outlined in Figure 8.

### 2.3.1 Data-based Techniques

Federated learning employs a **shared global model**, computed by the server as an average of the models over some participants in (see step (11)), to compute an optimal solution $\theta^*$ of the optimization problem (5). This approach performs effectively in its native application of horizontal cross-device scenarios with small, balanced, and similar datasets (Yang et al., 2018; Hard et al., 2018). However, in horizontal settings, under statistical data heterogeneity (seen in Section 2.2), or unbalanced agent participation (seen in Section 2.2.3), the returned local models $\theta_k(t)$ show high variance, which famously leads the average model $\bar{\theta}(t)$ to drift away from $\theta^*$, the solution of (5), a phenomenon well-known in FL as *client drift* (Karimireddy et al., 2020; Li et al., 2022). To mitigate this client drift, a variety of methods have been developed that retain a shared global model paradigm but modify the *FedAvg* Algorithm 1 to achieve tighter convergence to an optimal solution $\theta^*$. These methods, collectively known as *data-based solutions*, range from regularizing the local models to influencing the underlying data distribution through data augmentation and client sampling.

**Local regularization.** Regularization approaches aim to minimize the discrepancy between the local models $\{\theta_k(t)\}_{k \in \mathcal{K}_t}$ returned to the server by modifying the local updates (7). Bounding the variance of local gradient updates in (7) across agents has shown theoretical guarantees (Li et al., 2020) as well as

empirical evidence (Li et al., 2020) for *FedAvg* to converge to the optimal solution $\theta^*$, also in non-i.i.d. settings. Local regularization methods can be broadly categorized into three approaches (Kairouz et al., 2021):

- *Gradient control methods* reduce the gradient variance across agents and within local iterations by incorporating variance correction terms into the local updates (7). These terms are usually computed from the previous iterations of the local $\theta_k(t-1)$ and global $\theta(t-1)$ models. The main approaches rely on adaptive algorithms (Karimireddy et al., 2020; Khanduri et al., 2021; Kim et al., 2024) or dual-based optimization (Fan et al., 2022; Gong et al., 2022; Zhou & Li, 2023).

- *Selective update strategies* reduce noise by refining which model components are updated in (7), either by restricting updates to the least noisy layers (Park & Joe-Wong, 2025) or by using local optimizers that ensure smoother gradients (Wang et al., 2020).

- *Objective modification approaches* modify the local empirical objective (2) by penalizing divergence from the global model $\theta(t-1)$ (Li et al., 2020; Yao & Sun, 2020; Li et al., 2021) or, conversely, encouraging deviation from the previous local model $\theta_k(t-1)$ (Li et al., 2021). It can also penalize divergence between predictions of the global and the local model (Lin et al., 2020), typically using contrastive losses (Lee et al., 2022). Alternatively, second-order extensions to SGD provide smoother local updates (Fallah et al., 2020; T Dinh et al., 2020; Luo et al., 2025).

- *Prototype-based regularization* regularize the embedding space, that is, the hidden output of the model $F$ before its prediction head, rather than the model space. It is achieved using "prototypes", that is, representative embeddings for each label (Snell et al., 2017). Agents construct local prototypes by aggregating embeddings of the same-label samples of $S_k$, which are aggregated at the server into global prototypes. Local training then incorporates regularization terms to align embeddings with these prototypes (Tan et al., 2022a) or contrastive losses (Tan et al., 2022b).

**Data augmentation.** Data augmentation addresses distributional shifts, including quantity, feature, and label shifts, as well as missing labels, by modifying the local datasets $\mathcal{S}_k$ used for local training (see (7) in Algorithm 1). Augmentation strategies in federated settings can be categorized as follows:

- *Server-instructed augmentation.* A central server may augment local datasets by redistributing samples between agents to balance distributional shifts (Zhao et al., 2018), by instructing selective down-sampling of overrepresented classes in agents local datasets (Duan et al., 2020), or by generating synthetic samples and communicating them to the agents (Jeong et al., 2018).

- *Independent local augmentation.* Agents may augment their local datasets independently. Techniques include traditional transformations (Zhang et al., 2018) or learned generative models (Wu et al., 2020). When labels are missing, contrastive losses can be used to train on the unlabeled local dataset, augmented with negative samples (Li et al., 2022; He et al., 2025).

- *Collaborative local data augmentation.* Agents may augment their local datasets by collaborating with other agents. This is particularly salient in vertical partitioning, where common samples have to be aligned between agents (Yang et al., 2022; Kang et al., 2022). When not all local samples of the local datasets can be aligned, the pseudo features or labels can be estimated with generative models (Kang et al., 2022), or by matching features across agents (Yang et al., 2022).

**Client selection.** Client selection techniques tackle the statistical imbalance seen in horizontal partition settings by modifying the server strategy for sampling agents in line 3 of Algorithm 1. There are two approaches:

- *Total availability.* Some works assume total availability of agents. Strategies are designed to select underperforming clients (Li et al., 2021; Cho et al., 2020), group clients with similar performance in the same round (Chai et al., 2020; Yang et al., 2021), or prioritize clients that contribute to

improving the global accuracy (Wang et al., 2020). These strategies can be further optimized using reinforcement learning techniques on the server side, such as Q-learning (Wang et al., 2020) or multi-armed bandits (Yang et al., 2021).

- *Partial availability.* Other works assume that not all agents are available at the same time. Several participation patterns are studied, ranging from cyclical patterns (Cho et al., 2023) to temporal participation modeled by stationary Markov chains (Wang & Ji, 2022; Rodio et al., 2023), or a combination of both using R-separated Markov chains (Sun et al., 2025; Xiang et al., 2024). Some works look at spatio-temporal correlation using Markov chains (Rodio et al., 2023) or assuming independent agent unavailability (Wang & Ji, 2024) or handle agent turnover with departing and newly arriving agents (Ruan et al., 2021). The techniques involve maintaining estimate of the participation statistics to modify the sampling strategy (line 2 in Algorithm 1) (Cho et al., 2020; Wang & Ji, 2022; Ruan et al., 2021; Ribero et al., 2022), the aggregation (line 11 in Algorithm 1) (Ruan et al., 2021; Ribero et al., 2022; Wang & Ji, 2024), and the local update (line 19 in Algorithm 1) (Ruan et al., 2021) accordingly to guarantee a biased convergence to $\theta^*$. Theoretical results on correlated participation (Rodio et al., 2023; Cho et al., 2023) establish the trade-off of fast and tight convergence to $\theta^*$.

---

**Data-based techniques** address client drift under statistical heterogeneity while maintaining the shared global model. They follow three complementary approaches: local regularization, data augmentation, and client selection. These techniques intervene at different stages of the learning pipeline: regularization constrains local gradient updates during training, data augmentation modifies the effective data distribution before training, and client selection filters which distributions reach aggregation.

**Trilemma.** Each intervention point trades effectiveness through variance reduction against resource efficiency. Regularization requires computing correction terms or maintaining dual variables, data augmentation demands generative models or sample redistribution, and adaptive client selection needs participation tracking and optimized aggregation schemes. The privacy implications vary by technique: while regularization methods preserve privacy, techniques requiring data sharing (server-instructed augmentation, collaborative augmentation, prototype aggregation) expose local information and necessitate additional protection mechanisms.

---

### 2.3.2 Model-based Techniques

In the first place, FL aims to minimize the local objectives of (1) among all agents. However, the optimal solutions $F^*$ of the different local objectives in (1) might be different between agents due to factors like distributional shifts (Section 2.2) or task heterogeneity (Section 2.2.3). This implies that the convenient approach of looking for a model solution $F_\theta^* \in \mathcal{F}$, common to all agents, is suboptimal. Therefore, some efforts have been devoted to allow **model solutions that are personalized to each agent**, while retaining the main steps of the collaborative learning framework depicted in Section 2.1. With personalization, the optimization problem becomes

$$\min_{\theta_1 \in \Theta_1, \ldots, \theta_K \in \Theta_K} \frac{1}{K} \sum_{k=1}^{K} \mathcal{L}_k(F_{\theta_k}). \tag{6}$$

Personalization is a well-established direction in federated learning (Tan et al., 2022; Sabah et al., 2024). It can be achieved using two approaches: keeping a common architecture while personalizing the model parameters $\theta_k \in \Theta$, or personalizing the model architectures $\{\Theta_k\}_{k \in \mathcal{K}}$ themselves.

**Parameter personalization.** $\forall l \neq k, \theta_k \neq \theta_l; \Theta_k = \Theta_l$ Parameter personalization maintains a common domain $\Theta$ for the personalized models but allows models to converge to distinct parameters. These techniques apply exclusively to horizontal partitions, where all agents maintain similar model architectures. We distinguish:

- *Local adaptation.* Local adaptation extends Algorithm 1 by allowing each agent to modify the received global model parameters $\theta(t)$ into a personalized version $\theta_k(t + \frac{1}{2})$ through additional steps in the local training round. This adaptation can occur after the federated learning process, as in *meta-learning*, where $\theta(T)$ is rapidly fine-tuned on local data (Finn et al., 2017; Jiang et al., 2019). Alternatively, adaptation can be performed in every round $t$ by modifying the local model initialization step (line 5 in Algorithm 1), e.g., by interpolating between $\theta(t)$ and $\theta_k(t - 1)$ (Deng et al., 2020), using learnable local masks (Li et al., 2021; Dai et al., 2022; Huang et al., 2022), or applying low-rank adaption methods to $\theta(t)$ (Sun et al., 2024; Guo et al., 2025).

- *Hypernetwork-based.* A hypernetwork (Klein et al., 2015), that is, a network that generates the weights of a target model, can be used to produce personalized parameters for each agent. Given an agent representation vector $c_k$, the server-side hypernetwork $\mathcal{H}_\alpha$ with trainable parameters $\alpha$ outputs the personalized model $\theta_k(t) = \mathcal{H}_\alpha(c_k)$. Instead of maintaining a shared model $F_{\theta(t)}$, all agents share a *common* hypernetwork with learnable weights $\alpha$ jointly trained across agents by minimizing (Shamsian et al., 2021)

$$\frac{1}{K} \sum_{k=1}^K \hat{\mathcal{L}}_k\big(F_{\mathcal{H}_\alpha(c_k)}\big). \tag{7}$$

  Training is performed on the server, where agents transmit $\nabla_\alpha \hat{\mathcal{L}}_k$ and the server updates $\alpha$ via gradient steps. The representation $c_k$ can be fixed (e.g., encoding an agent type) or updated jointly with $\alpha$.

- *Cluster-based.* Similar agents are divided into clusters, and learn cluster-specific models. Assignments can be static (Duan et al., 2021), where clusters operate independently, or dynamic, allowing clustering and learning to occur simultaneously. A straightforward approach leverages similarities for clustering. These methods typically alternate between clustering and local updates, framing the problem as an optimization task (Long et al., 2023). Clustering optimization aims to minimize inter-cluster similarities (Briggs et al., 2020; Werner et al., 2024) or maximize intra-cluster similarities (Sattler et al., 2020; Briggs et al., 2020; Long et al., 2023; Werner et al., 2024). Similarities are computed between clients (Briggs et al., 2020), clients and clusters (Ghosh et al., 2020; Mansour et al., 2020; Werner et al., 2024), or clients and the global model (Sattler et al., 2020), using losses (Ghosh et al., 2020; Sattler et al., 2020) or gradients (Werner et al., 2024). Techniques include greedy assignment (Ghosh et al., 2020; Mansour et al., 2020), recursive bi-partitioning (Sattler et al., 2020), recursive fusion (Briggs et al., 2020), and threshold-based clustering robust to Byzantine settings (Werner et al., 2024). Assignments can be performed server-side (Sattler et al., 2020; Long et al., 2023) or client-side, although this requires sending all cluster models (Ghosh et al., 2020). Additionally, some studies also address the assignment of new entrants to clusters (Duan et al., 2021).

- *Similarities between agents* can be used to favor aggregation between the most similar agents even without hard division into clusters. A similarity matrix $\Omega = [\omega_{k,l}] \in \mathbb{R}^{K^2}$, measuring the proximity between local objectives, is maintained by the server to build personalized models. These similarities can be computed explicitly using metrics on local models $\theta_k(t)_{k \in \mathcal{K}}$ (Huang et al., 2021; Chen & Zhang, 2022) or by evaluating local empirical objectives on models of other agents $\hat{f}_k(\theta_l)_{(k,l) \in \mathcal{K}^2}$ (Onoszko et al., 2021; Zec et al., 2022). Given $\Omega$, models can be personalized by averaging the most similar ones (Onoszko et al., 2021) or using weighted averages (Zec et al., 2022; Huang et al., 2021). Alternatively, $\Omega$ can be learned as a problem parameter by adding a regularization term $\mathcal{R}(\Omega, [\theta_{k,l}]_{(k,l) \in \mathcal{K}^2})$ to (6): (Smith et al., 2017; Huang et al., 2021; Shoham et al., 2019; Chen & Zhang, 2022; Chen et al., 2022). This requires alternating optimization, updating $\Omega$ and model parameters in separate steps (Smith et al., 2017; Huang et al., 2021; Chen et al., 2022). While learning $\Omega$ introduces significant communication overhead, decentralized protocols (Zec et al., 2022) and smart sampling (Zec et al., 2024) help mitigate scaling issues.

**Architecture personalization.** $\forall l \neq k, \theta_k \neq \theta_l; \Theta_k \neq \Theta_l$ Agents can utilize different model architectures, for example, neural networks of different sizes. The need for architecture personalization is driven both by

concept statistical imbalance (seen in Section 2.2.2) but also by system heterogeneity constraints like device heterogeneity and application heterogeneity (seen in Section 2.2.3). We distinguish:

- *Model decoupling.* Local models, $\theta_k$, are divided into a shared component, $\theta_k^c = \bar{\theta}$, trained using the *FedAvg* algorithm, and a private component, $\theta_k^p$, which may vary in architecture across agents. In neural networks, this split is typically *layer-wise* (depth) (Wang et al., 2016; 2023; Liang et al., 2020; Arivazhagan et al., 2019; Bui et al., 2019), but may also occur *width-wise* (Diao et al., 2021) or in a hybrid manner (Kang et al., 2025). The structure of the split directly influences communication costs. Some strategies reduce these costs by replicating shared layers to private components, especially in CNNs (Park & Joe-Wong, 2024), or by focusing on the most impactful layers (Park & Joe-Wong, 2025). The level of decoupling impacts model performance: greater device heterogeneity benefits from more private neurons, while noisy local updates favor a larger shared component (Liang et al., 2020; Collins et al., 2021). The optimal split can be determined through *adaptive* (Vahidian et al., 2021) or even *learnable* (Deng et al., 2022) strategies. However, incomplete aggregation schemes introduce security risks (Park & Joe-Wong, 2024).

  *Multi-modal personalization.* The server maintains $M$ global models, $\theta = \cup_{m=1,...,M} \theta_m^c$, where model $\theta_m^c$ corresponds to a data domain $m$. Agents selectively combine these models based on their local data, addressing feature shifts. Identifying the modality can rely on prior knowledge (Chen & Zhang, 2022) or statistical estimation (Huang et al., 2019). Model combinations vary: agents may assign distinct models per sample (Huang et al., 2019) or aggregate outputs from multiple models when tasks require multi-modal fusion (Chen & Zhang, 2022). Training often leverages *parameter personalization*, using clustering (Huang et al., 2019) or agent similarity (Smith et al., 2017; Chen & Zhang, 2022) to train these models jointly.

- *Knowledge distillation.* Instead of sharing full model parameters, agents can exchange sample-like information, such as predicted labels or embeddings, which can be shared across agents with potentially different architectures. This information may come from a globally shared unlabeled dataset (Li & Wang, 2019; Bistritz et al., 2020; He et al., 2020) or, for more communication-efficient approaches, from prototypes, i.e., a fixed set of representatives of the same label samples built collaboratively (Tan et al., 2022b; Kim et al., 2023). The shared sample information must be aligned across agents, typically using knowledge distillation losses (Hinton et al., 2014), which can be further improved with contrastive losses (Tan et al., 2022b). Exchanging these compact data instead of full models can significantly reduce communication and storage costs (Tan et al., 2022b). Even when a global model is not strictly required, maintaining one can be useful: it can distill knowledge to agents with smaller architectures, rapidly initialize new agents' models from a pre-trained model (Li & Wang, 2019; Kim et al., 2023), or continuously provide shared knowledge through server-side training (He et al., 2020; Lin et al., 2020; Zhu et al., 2021).

---

**Model personalization techniques** address the fundamental limitation that a single shared model cannot optimally serve all agents under heterogeneous conditions. They follow two complementary strategies: parameter personalization and architecture personalization. Parameter personalization maintains common architectures while allowing the model parameters to diverge through local adaptation, hypernetworks, clustering, or similarity-based weighting. Architecture personalization accommodates structural heterogeneity through model decoupling (shared-private splits), maintaining multiple global models, or the alignment of representations via distillation and prototypes.

**Trilemma.** Agent-specific adaptation improves local performance but decreases efficiency due to increased coordination complexity (similarity computation, cluster maintenance, hypernetwork training) and communication overhead (multiple model transfers, prototype exchange). The techniques that operate purely on model parameters (local adaptation, layer decoupling) limit information leakage, while those requiring explicit agent comparison or data sharing (similarity estimation, clustering, distillation, prototype aggregation) expose agent-specific information.

---

### 2.3.3   Training and Inference under Vertical Partitioning

The techniques presented in Sections 2.3.1 and 2.3.2 consider *horizontal partitioning* of the data (Section 2.2.1). In many domains, however, the data are instead *vertically partitioned*, with agents observing complementary feature domains of the same samples (Section 2.2.1). In this case, effective learning requires combining the information from these partitions. There are two main lines of solutions for training under vertical partitioning, training with *collaborative* inference, or for *isolated* inference.

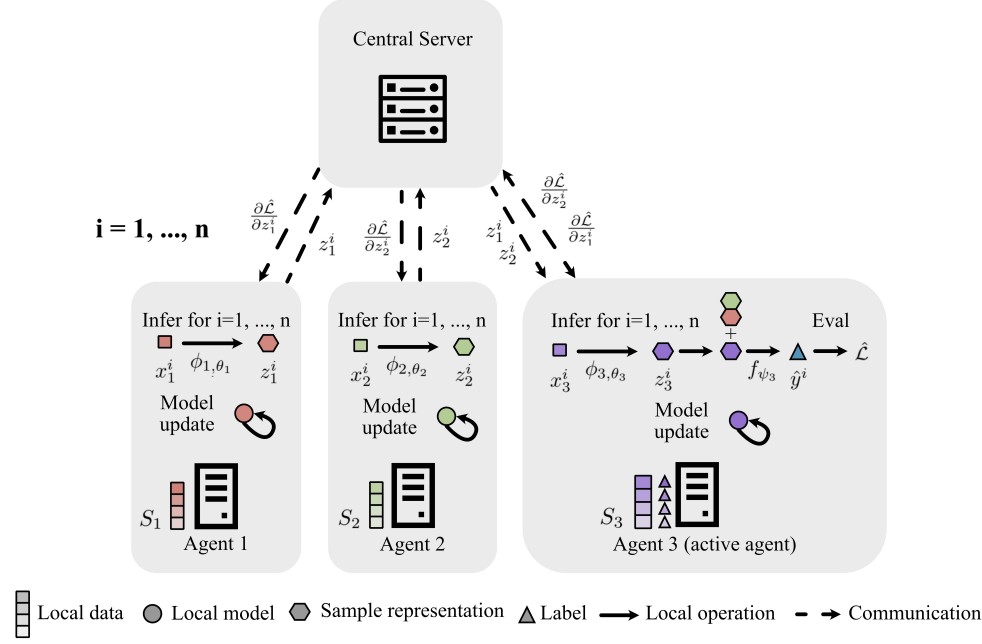

Figure 9: **Training under vertical partitioning with three agents.** During inference, the computations of the global model $F_{\theta,\psi}$ are distributed across agents: to produce a prediction $\hat{y}^i$ for input $x^i$, each agent $k$ communicates its partial representation $z_i^k = \phi_{k,\theta_k}(x_k^i)$ of the sample. Only one or a subset of the agents (here, agent 3) possesses the true label, which can evaluate the training loss $\hat{\mathcal{L}}$. During training, the gradients with respect to the learnable parameters $\theta_1, \theta_2, \theta_3, \psi_3$ must be communicated across agents in the form of $\frac{\partial \hat{\mathcal{L}}}{\partial z_k^i}$.

**Training with collaborative inference.**   Models can be trained under vertical data partitions by collaboratively optimizing a collaborative inference loss. In previous Sections 2.3.2 and 2.3.1, each agent trains a local model $\theta_k$ used to perform inference *in isolation* on its local data. Collaboration occurs solely during training when agents periodically share their local model iterate $\theta_k(t)$. However, when agents know about the same sample, as seen in the example of *vertical data partition* (Section 2.2.1), agents might benefit from collaborating during inference as well, for building a common prediction. This collaborative inference imposes a reevaluation of the problem definition of standard FL in (5), and a reevaluation of the training protocol. Under vertical partition, input features of a sample $(x^i, y^i)$ are split across the agents $k \in \mathcal{K}$. Each agent $k$ holds $x_k^i$, a subset of the features $x^i$ such as $x^i = \cup_{k \in \mathcal{K}} x_k^i$, forming the local dataset $\mathcal{S}_k = \{x_k^i \in \mathcal{X}_k\}_{i \in \mathcal{I}}$. For simplicity, we assume that a single agent, agent $k^*$ has access to the labels, hence $\mathcal{S}_{k^*} = \{(x_{k^*}^i, y^i) \in \mathcal{X} \times \mathcal{Y}\}_{i \in \mathcal{I}}$. Since the features of each sample are split across agents, the global model $F \in \mathcal{F}$, used to infer on the centrally available sample $(x^i, y^i)$, can be naturally decomposed to process the features locally [2]. Consider that each agent $k$ holds a local model component $\phi_{k,\theta_k}$ mapping from $\mathcal{X}_k$ to latent space $\mathcal{Z}_k$, parameterized

---

[2]A related setting, known as *split learning* (Gupta & Raskar, 2018; Vepakomma et al., 2019), partitions the inference model across agents for computational offloading, where typically one agent holds the data while others provide model layers. While this can be viewed as an extreme case of vertical partition (where one agent holds all features), we focus on scenarios where multiple agents each possess complementary private data.

by $\theta_k \in \Theta_k$ while a final predictor $f_{\psi_{k^*}}$, parameterized by $\psi_{k^*} \in \Psi_{k^*}$, is held by agent $k^*$. Letting $F$'s model parameters be $\theta = (\theta_1, \ldots, \theta_K, \psi_{k^*})$ with parameter space $\Theta = \Theta_1 \times \cdots \times \Theta_K \times \Psi_{k^*}$, the inference of $F_\theta$ on a sample indexed by $i$ can be expressed as

$$\hat{y}^i = F_\theta(x^i) = F_\theta((x_1^i, \ldots, x_K^i)) = f_{\psi_{k^*}} \left( \phi_{1,\theta_1}(x_1^i), \ldots, \phi_{K,\theta_K}(x_K^i) \right). \tag{8}$$

Therefore, the partition of the global model $F$ results in a *collaborative inference* described in Figure 9: for performing a prediction on $x^i$, each agent $k$ communicates its partial representation $\phi_{k,\theta_k}(x_k^i)$ of the sample. Since collaborative inference requires agents to exchange intermediate representations and gradients, which may expose sensitive information, practical deployments typically incorporate privacy protection mechanisms to mitigate such leakage (Liu et al., 2024), detailed in Section 2.5.

The population loss can be defined for all agents as

$$\mathcal{L}_{VFL}(F_{\theta_1, \ldots, \theta_K, \psi_{k^*}}) := \mathbb{E}_{((x_1, \ldots, x_K), y) \sim \mathcal{D}} \ell \left( F_{\theta_1, \ldots, \theta_K, \psi_{k^*}} ((x_1, \ldots, x_K)), y \right). \tag{9}$$

Since the data distribution $\mathcal{D}$ is unknown, the loss (9) is approximated by an empirical counterpart defined over $S_k$

$$\hat{\mathcal{L}}_{VFL}(F_{\theta_1, \ldots, \theta_K, \psi_{k^*}}) := \frac{1}{|\mathcal{I}|} \sum_{i \in \mathcal{I}} \ell \left( F_{\theta_1, \ldots, \theta_K, \psi_{k^*}} (x_1^i, \ldots, x_K^i), y^i \right). \tag{10}$$

Notably, both (9) and (10) are common to all agents, allowing the global loss (3) to be equivalently expressed in this collaborative setting as

$$\frac{1}{K} \sum_{k=1}^{K} \mathcal{L}_{VFL}(F) = \mathcal{L}_{VFL}(F). \tag{11}$$

However, unlike in (3), where local losses can be computed independently by each agent, the structure of (9) requires collaborative inference to evaluate the loss. As a result, computing the objective requires inter-agent communication, and we refer (9) and (10) to as *collaborative inference loss*. Despite this difference, the learning goal remains the same: to approximate a minimizer of the population loss (9) by minimizing its empirical surrogate (10) over a parametrized hypothesis class. This leads to the following optimization problem of *Vertical Federated Learning*, in direct analogy with (5):

$$\min_{\theta_1 \in \Theta_1, \ldots, \theta_K \in \Theta_K, \psi_{k^*} \in \Psi_{k^*}} \hat{\mathcal{L}}_{VFL}(F_{\theta_1, \ldots, \theta_K, \psi_{k^*}}). \tag{12}$$

The typical training pipeline follows two steps (Liu et al., 2024):

- *Privacy-preserving sample alignment.* The formulation in (10) assumes a shared understanding of sample indices $\mathcal{I}$, their features $(x^i)$, and labels $(y^i)$ across agents. However, in practice, each agent only sees its local view, making the computation of (10) infeasible without preliminary coordination. To address this problem, the first step is to identify the common index set $\mathcal{I}$ across the agents. It is usually achieved through the *Private Set Intersection (PSI)* protocols (Liang & Chawathe, 2004; Pinkas et al., 2018). PSI is extended to multiparty setups in (Zhou et al., 2021; Lu & Ding, 2020).

- *Collaborative training.* The optimization problem in (12) is solved collaboratively between agents $\mathcal{K} = \{1, \ldots, K\}$, as summarized in Algorithm 2. It proceeds in $T$ rounds, where each round consists of two phases:

  - *Collaborative Inference (lines 3–9):* Each agent $k$ computes intermediate representations $z_k^i = \phi_{k,\theta_k}(x_k^i)$ (line 3) for each sample and sends them to the active agent $k^*$ (line 4), which aggregates them and computes the empirical loss $\hat{\mathcal{L}}(F_\theta)$ on the mini-batch (line 7), updates $\psi_{k^*}$ (line 8), and sends partial gradients $\frac{\partial \hat{\mathcal{L}}}{\partial z_k^i}$ to each agent (line 9).

  - *Collaborative Model Update (lines 11–12):* Each agent applies the chain rule to compute $\nabla_{\theta_k} \hat{\mathcal{L}}$ (line 11) and updates its local parameters (line 12).

---

**Algorithm 2** Vertical Federated Learning algorithm (VFL) adapted from (Liu et al., 2024)

---

**Require:** Set of agents $\mathcal{K} = \{1, \ldots, K\}$, shared sample index set $\mathcal{I}$, learning rates $\eta_1, \eta_2$, local parameters $\theta(0) = \{\theta_k(0)\}_{k=1}^K \cup \{\psi_{k^*}(0)\}$, number of communication rounds $T$, mini-batch size $B$

1: **for** each round $t = 1, \ldots, T$ **do**

    **Collaborative Inference:**

2:     **for** each agent $k \in \mathcal{K}$ **in parallel do**

3:         Compute local representations: $z_k^i = \phi_{k,\theta_k(t-1)}(x_k^i)$ for all[3] $i \in \mathcal{I}$

4:         Send $\{z_k^i\}_{i \in \mathcal{I}}$ to the active agent $k^*$

5:     **end for**

    Active agent $k^*$:

6:     Aggregate $\{z_k^i\}$ and compute empirical predictions for each $i \in \mathcal{I}$: $\{\hat{y}^i\}_{i \in \mathcal{I}} = \left\{ f_{\psi_{k^*}}(z_1^i, \ldots, z_K^i) \right\}_{i \in \mathcal{I}}$

7:     Compute partial empirical loss on samples of index $\mathcal{I}$, $\hat{\mathcal{L}}_{VFL}\left(F_{\psi_{k^*},\theta_1,\ldots,\theta_K}\right) = \frac{1}{|\mathcal{I}|} \sum_{i \in \mathcal{I}} \ell\left(\hat{y}^i, y^i\right)$

8:     Update predictor model: $\psi_{k^*}(t) \leftarrow \psi_{k^*}(t-1) - \eta_1 \nabla_{\psi_{k^*}} \hat{\mathcal{L}}_{VFL}$

9:     Compute and send partial derivatives to each agent $k$: $\left\{ \frac{\partial \hat{\mathcal{L}}}{\partial z_k^i} \right\}_{i \in \mathcal{B}_t}$

    **Collaborative Model Update:**

10:     **for** each agent $k \in \mathcal{K}$ **in parallel do**

11:         Compute gradients via chain rule: $\nabla_{\theta_k} \hat{\mathcal{L}} = \sum_{i \in \mathcal{I}} \frac{\partial \hat{\mathcal{L}}_{VFL}}{\partial z_k^i} \cdot \frac{\partial z_k^i}{\partial \theta_k}$

12:         Update local model: $\theta_k(t) \leftarrow \theta_k(t-1) - \eta_2 \nabla_{\theta_k} \hat{\mathcal{L}}_{VFL}$

13:     **end for**

14: **end for**

15: **return** Final parameters $\theta = (\theta_1(T), \ldots, \theta_K(T), \psi_{k^*}(T))$ and sample predictions $\{\hat{y}^i\}_{i \in \mathcal{I}}$

---

Algorithm 2 can be extended in several ways to adapt to more realistic settings with:

- *Pre-training.* To decrease the level of communication required by the collaboration, agents may pre-train their local models on private data prior to collaborative training (Feng et al., 2022; Shen et al., 2025). This approach leads to scalable VFL, but may introduce bias from local data distributions (Shen et al., 2025).

- *Model averaging.* The VFL protocol in Algorithm 2 assumes that each agent $k \in \mathcal{K}$ maintains a private local model $\phi_{k,\theta_k}$ with domain parameters $\Theta_k$ which are individually updated based on the collaborative model update phase (lines 9–12). However, in practice, certain agents might use the same model architecture, leading to the same domain space $\Theta_k = \Theta_l$. This can occur, for example, when agents (e.g., hospitals) collect the same features related to the same samples (e.g., patients). In such cases, it can be beneficial to introduce a periodic averaging of the models, $\theta_k(t) = \theta_l(t) = \frac{\theta_k(t) + \theta_l(t)}{2}$, in the collaborative model update steps (Fu et al., 2024).

- *Unlabeled data.* The formulation in (12) assumes full access to supervision through the labels $y^i$. However, in practical scenarios, a significant portion of the data may be unlabeled. To handle such cases, the empirical loss in (10) can be extended to include self-supervised objectives, such as contrastive losses over unlabeled samples (Li et al., 2022; He et al., 2025). Additionally, the local inference modules $\phi_{k,\theta_k}$ can be instantiated as autoencoders, enabling self-supervised training via reconstruction losses (Feng, 2022). These extensions apply primarily to the collaborative inference phase (lines 3–4) of Algorithm 2, and they allow agents to learn useful representations even in the absence of labels.

**Training for isolated inference.** Transfer techniques aim to train models capable of performing isolated inference on vertically partitioned data while achieving effectiveness comparable to collaborative inference. This addresses the primary limitation of collaborative inference, which is the communication overhead at

---

[3]Note that, in practice, this operation can be efficiently parallelized using batching over the samples, voluntarily eluded here for clarity.

inference time, while still exploiting the knowledge contained in vertically partitioned datasets. Consider a transfer partition as defined in Section 2.2.1, involving a *source* agent $s$ and a *target* agent $t$, each aware of their common sample set $\mathcal{I}_s \cap \mathcal{I}_t$, obtained, for example, via *PSI* protocols. Two main approaches are commonly used depending on the structure of the partition (Liu et al., 2024):

- *Knowledge distillation.* When the feature spaces of the agents exhibit significant overlap, knowledge distillation (KD) techniques, also discussed for horizontal partitioning in Section 2.3.2, can be adapted to vertical settings. The objective is to transfer the knowledge of the collaborative inference model $(\phi_{s,\theta_s}, \phi_{t,\theta_t}, f_{\psi_s})$, pre-trained on domains $\mathcal{X}_s$ and $\mathcal{X}_t$ by agents $s$ and $t$ according to (12), to a student model $(\phi_{\theta_t^t}, f_{\psi_t^t})$ for agent $t$, that performs *isolated inference* on domain $\mathcal{X}_t$. To this end, agent $t$ trains its model on a distillation loss that aligns the representations of $\phi_{\theta_t^t}$, $\phi_{s,\theta_s}$, and $\phi_{t,\theta_t}$ in the shared latent space $\mathcal{Z}_s \times \mathcal{Z}_t$, using aligned unlabeled samples (Huang et al., 2023; Ren et al., 2022).

- *Transfer learning.* When the number of aligned samples is limited, and only the source agent $s$ has sufficient data to train an independent model, transfer learning techniques (Liu et al., 2020) can be employed. The goal is to adapt a model pre-trained by agents $s$ on its dataset $S_s$ and domain $\mathcal{X}_s$ to the target domain $\mathcal{X}_t$, using a shared embedding space $\mathcal{Z}$. The core assumption in this setting is that unaligned samples from $s$ and $t$ still yield semantically similar representations in $\mathcal{Z}$. The process involves two steps. First, agent $s$ trains a local encoder $\phi_{s,\theta_s} : \mathcal{X}_s \to \mathcal{Z}$ and a predictor $f_{\psi_s}$ on $S_s$, yielding representations $Z_s = \{z_s^i, i \in \mathcal{I}_s\} = \{\phi_{s,\theta_s}(x_s^i), i \in \mathcal{I}_s\}$. Then, agent $t$ initializes its own encoder $\phi_{t,\theta_t} : \mathcal{X}_t \to \mathcal{Z}$ and predictor $f_{\psi_t}$. The training of parameters $\psi_t$ and $\theta_t$ proceeds by aligning $Z_t = \{z_t^i, i \in \mathcal{I}_t\} = \{\phi_{t,\theta_t}(x_t^i), i \in \mathcal{I}_t\}$ with $Z_s$ in the latent space $\mathcal{Z}$ using a transfer loss, while keeping $\psi_s, \theta_s$ frozen. Since $Z_s$ and $Z_t$ may differ in size, advanced transfer losses are typically required, such as feature matching (Yang et al., 2022), pseudo-label matching (Feng et al., 2022), or adversarial methods (Kang et al., 2022).

> **Learning under vertical partitioning methods** improves learning effectiveness when agents hold complementary features of shared samples. They follow two paradigms based on inference requirements: *collaborative inference* and *isolated inference*. Learning with collaborative inference enables joint optimization over the full feature space, while isolated inference trains models for autonomous deployment through knowledge distillation (when feature spaces overlap) or transfer learning (when source agents have richer data).
>
> **Trilemma.** Collaborative inference decreases efficiency due to persistent coordination overhead (representation exchange, gradient communication) but achieves accurate models. Isolated inference avoids the communication overhead at inference, but needs complex representation alignment. Privacy exposure is severe in both approaches compared to horizontal methods: collaborative inference continuously shares feature-derived embeddings during training and inference, while isolated inference requires sample alignment through PSI protocols and representation sharing during knowledge transfer, necessitating explicit privacy-preserving mechanisms throughout.

## 2.4 Improving Learning Efficiency

While Section 2.3 explored techniques to enhance the effectiveness of the learning process, these approaches often overlook the constraints of the communication and computing resources introduced in Section 2.2.3. This section examines strategies for *efficient* learning, that is, ways to achieve the fast convergence of the learning models despite the constraints given by the network topology, the heterogeneity of the devices and network connections, and the often costly and unreliable communication links. We discuss **model aggregation** strategies, and solutions for **asynchronous aggregation**, and for increased **communication efficiency**.

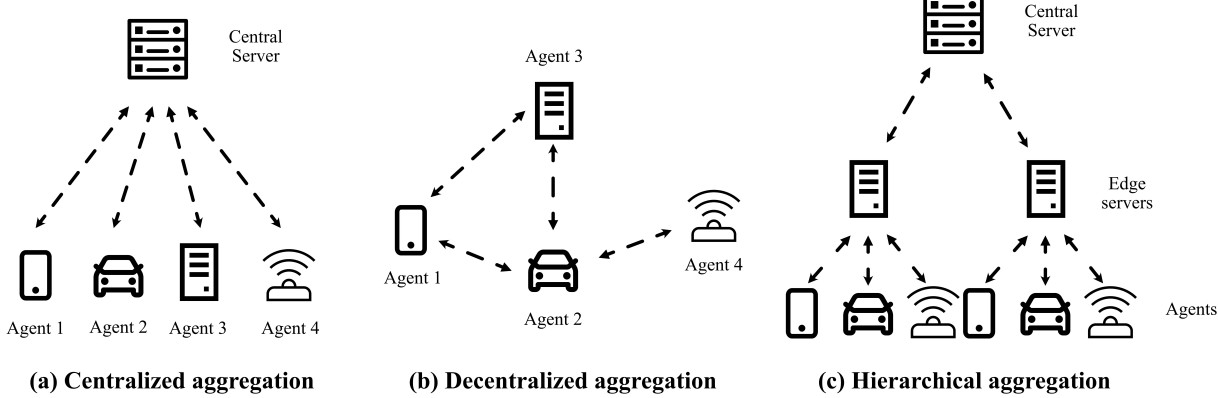

Figure 10: **Different aggregation patterns for collaborative learning.** *Centralized aggregation* (a) uses a central server that collects and combines model updates from all agents. *Decentralized aggregation* (b) allows agents to exchange and aggregate updates over a network with restricted communication links. *Hierarchical aggregation* (c) relies on a multi-level structure where a central server coordinates with edge servers, which in turn oversee the aggregation of updates of their associated agents.

### 2.4.1 Model Aggregation

The efficiency of collaborative learning depends primarily on the model aggregation pattern, defined by the topology and the timing of the model parameter exchange and aggregation. This section discusses aggregation schemes in three aggregation topologies: centralized, decentralized, and hierarchical, as shown in Figure 10.

**Centralized aggregation.** Centralized aggregation is the standard strategy for combining local models, as employed in Algorithm 1. In this setting, a central server coordinates training: at each round $t = 1, \ldots, T$, it collects local model parameters $\theta_k(t)$, each obtained after training for $E$ local epochs, aggregates them, and redistributes the result to all agents.

The motivation for centralized aggregation originates in distributed computing, where distributing stochastic gradient computations can theoretically reduce training time by a factor proportional to the number of agents $K$ (Zinkevich et al., 2009; McDonald et al., 2010). However, naive parallelization with aggregation at every epoch ($E = 1$) incurs significant communication overhead, which can offset the computational benefits depending on system cost constraints described in Section 2.2.3.

When communication overhead outweighs the gains from faster convergence, a simple workaround is *one-shot aggregation*, where models are aggregated once after local training completes ($T = 1$)(Mcdonald et al., 2009; Zinkevich et al., 2010). While highly communication-efficient (Zhang et al., 2012), this strategy relies on strong assumptions, such as statistical similarity across agents datasets, that become increasingly difficult to satisfy as $K$ increases (Li et al., 2020; Khaled et al., 2020).

A more balanced alternative is to let agents perform multiple local updates ($E > 1$) before periodically aggregating their models. This approach, known as *local SGD*, leverages the robustness of SGD to noise and relaxed synchronization (Dean et al., 2012; Bottou et al., 2018). It has proven successful both in practice and theory, achieving near-linear speedup under mild i.i.d. assumptions (Stich, 2019; Yu et al., 2019; Li et al., 2020).

Further refinements include hybrid schemes combining one-shot and local SGD (Hou et al., 2022), the use of server-side optimizers (Reddi et al., 2021), and dual-based methods such as ADMM (Boyd et al., 2011; Zhou & Li, 2023).

**Decentralized aggregation.** Decentralized aggregation is the opposite of centralized aggregation: agents aggregate models directly without a central coordinator. It is often proposed as a more robust alternative, mitigating the single point of failure inherent to centralized schemes. In some cases, it is also necessitated by system constraints or by governance, trust, and incentive considerations that make a central coordinator undesirable or infeasible. Unlike centralized approaches, both training and aggregation occur over a mesh network, introducing additional challenges related to network topology.

Distributed aggregation builds on the early results of *distributed averaging* (DeGroot, 1974; Tsitsiklis et al., 1986; Bertsekas & Tsitsiklis, 2015). For simplicity, consider agents connected according to a mesh topology modeled as a graph, and holding initial value vectors $\pi_i(0)$, with an average of $\bar{\pi}(0)$. At each step $t$, the updates follow a linear rule: $\forall k \in \mathcal{K}, \pi_k(t+1) = w_{kk}\pi_k(t) + \sum_{l \in \mathcal{N}(k)} w_{kl}(t)\pi_l(t)$, where $\mathcal{N}(k)$ denotes the neighbors of agent $k$. These updates can be expressed with a Markov chain as $\Pi(t+1) = W(t)\Pi(t)$ where $W(t) = [w_{kl}(t)]_{(k,l)\in\mathcal{K}^2}$ and $\Pi(t) = [\pi_{kl}(t)]$. Convergence to a consensus state occurs when all nodes reach the same value $\forall k \in \mathcal{K}, \lim_{t\to\infty} \pi_k(t) = \pi^*$. Average consensus is achieved if, in addition, $\pi^* = \bar{\pi}(0)$. In the literature, we find two algorithmic approaches for distributed averaging (Denantes et al., 2008):

- *Average consensus* algorithms assume that all nodes communicate and update their values simultaneously. The problem is formulated by minimizing the *Dirichlet energy*, expressed as

$$\frac{1}{2}\Pi(t)^\top L\Pi(t) = \sum_{i\in\mathcal{K}}\sum_{j\in\mathcal{N}(i)} \left(\pi_i(t) - \pi_j(t)\right)^2, \tag{13}$$

  which quantifies the quadratic disagreement between neighboring nodes, where $L$ is the graph Laplacian (Mohar et al., 1991) and $\mathcal{N}(i)$ denotes the neighbors of node $i$ in the mesh network. In *continuous time*, updates towards a minimum of (13) can be expressed as a diffusion equation as $\dot{\pi}(t) = -L\pi(t)$ in (Cortés, 2006; Jiang & Wang, 2009; Olfati-Saber et al., 2007). In *discrete time*, the update takes a Markov chain form $\Pi(t+1) = W(t)\Pi(t)$ in (Blondel et al., 2005; DeGroot, 1974; Nedic et al., 2009; Olshevsky & Tsitsiklis, 2009). For example $W(t)$ is derived directly from (13) with the update rule $W(t) = I - \epsilon L$, with step size $\epsilon > 0$ and $I$ the identity matrix in (Saber & Murray, 2003; Olfati-Saber & Murray, 2004; Olfati-Saber et al., 2007). For static graphs, convergence is guaranteed when $W$ is doubly stochastic and connected. The convergence rate is exponential, determined by the topology of the graph (Fiedler, 1973), through the second-largest eigenvalue of the symmetric Laplacian $\lambda_2(L_s)$ (Fax & Murray, 2004; Olfati-Saber & Murray, 2004). Average consensus algorithms have been widely applied in multi-agent systems (Vicsek et al., 1995; Jadbabaie et al., 2003; Blondel et al., 2005; Fax & Murray, 2004) and adapted to address issues like dynamic topologies (Jadbabaie et al., 2003; Olshevsky & Tsitsiklis, 2009; Moreau, 2005) and delays (Blondel et al., 2005; Tsitsiklis et al., 1986).

- *Gossip algorithms* perform pairwise communication (Karp et al., 2000; Bawa et al., 2003). Upon a tick of its internal clock, node $k$ communicates with node $l$ with probability $p_{kl}$, and they update their values as $\pi_k(t+1) = \pi_l(t+1) = \frac{\pi_k(t)+\pi_l(t)}{2}$ (Boyd et al., 2005; 2006). The probability matrix $P$ mirrors the role of adjacency matrices in synchronous algorithms. Gossip protocols are communication-efficient, and due to their inherent randomness, they are robust to asynchronous execution and partial observability (Jelasity et al., 2005; Denantes et al., 2008). Variations include geographic routing (Dimakis et al., 2008; Bénézit et al., 2010), weighted updates (Bénézit et al., 2010), and memory-augmented protocols (Cao et al., 2006; Liu et al., 2013).

Distributed averaging combined with local learning methods leads to decentralized aggregation. Compared to centralized aggregation, the challenge is that both the local learning and the model aggregation processes require multiple steps (as represented by respective indíces $e$ and $t$). Early methods combine local gradient updates with a single averaging step (Tsitsiklis et al., 1986; Nedic & Ozdaglar, 2009). More advanced techniques incorporate multiple propagation steps (Johansson et al., 2008), handle link failures (Lobel & Ozdaglar, 2008), enforce constraints via projections (Nedic et al., 2010), or consider non-i.i.d. data (Wang et al., 2022). Gossip-based methods (Ram et al., 2009; Ormándi et al., 2013; Blot et al., 2016) enhance robustness to asynchronous participation and dynamically changing agent population (Hegedűs et al., 2019).

Dual-based techniques, such as ADMM (Boyd et al., 2011), also support decentralized learning, where dual variable averaging is performed synchronously (Nesterov, 2009; Wei & Ozdaglar, 2012) or via gossip (Vanhaesebrouck et al., 2017).

The methods above rely on *distributed averaging*, where each agent's value $\pi_k(t)$ converges to a global consensus $\pi^* = \bar{\pi}(0)$. However, when the network topology reflects data heterogeneity, aggregation can instead balance global and local consensus. To preserve local information, the objective (13) can be modified to combine global agreement with local smoothness, as in $\frac{1}{2}\Pi(t)^T L \Pi(t) + \mu\|\Pi(t) - \Pi(0)\|_2^2$, as proposed in (Zhou et al., 2003). If this formulation admits a closed-form solution, it can be computed in a decentralized way using a broad class of *diffusion* (also known as *propagation* or *smoothing*) algorithms (Zhu & Ghahramani, 2002; Zhou et al., 2003). These mechanisms integrate naturally with local learning by initializing $\pi_k(t) = \theta_k(t)$, as in (Vanhaesebrouck et al., 2017).

**Hierarchical aggregation.** Both centralized and distributed aggregation schemes have limitations in scalability in terms of the number of agents and in terms of the size of the geographic area the agents are spread over. Hierarchical architectures are proposed to overcome these limitations by introducing multiple layers of aggregation. The lower layers typically follow a star topology with local aggregators. In wireless networks, these are often implemented at the base stations or edge servers. Aggregation at the higher layers may use a central server (Luo et al., 2020; Gupta et al., 2016; Wu et al., 2023; Azimi-Abarghouyi & Fischione, 2025), or be performed over a mesh structure (Castiglia et al., 2021; Zhang et al., 2022).

Hierarchical architectures are often motivated by retransmission constraints, where the wide geographic distribution of agents requires the model parameters to be transmitted over long transmission paths (Gupta et al., 2016). In wireless networks, the hierarchical structure improves the transmission quality over wireless links (Wen et al., 2022) and increases the efficiency of the learning process by localizing certain operations, which conserves communication resources and reduces time (Gupta et al., 2016). Clustering of agents helps to manage the heterogeneity of the system, supports personalization (Castiglia et al., 2021; Wu et al., 2023), and increases privacy by limiting data traffic to specific administrative regions or social groups (Zhou et al., 2023).

> **Model aggregation techniques** determine how and when agents exchange and combine their models, directly impacting communication efficiency and convergence behavior. *Centralized* aggregation offers simple coordination, but creates communication bottlenecks and single points of failure. *Decentralized* aggregation eliminates central coordination through distributed synchronous or asynchronous consensus protocols, naturally handling dynamic topologies. *Hierarchical* aggregation scales to larger systems by introducing multiple aggregation layers that localize communication.
>
> **Trilemma.** Centralized aggregation allows effective learning with accurate models and fast convergence. Under decentralized aggregation, both the convergence and the model accuracy depend on the aggregation topology, and hierarchical aggregation needs complex coordination to ensure efficient learning under non-i.i.d. data. Beyond topology, aggregation timing trades communication efficiency against convergence speed. The privacy exposure depends both on the number of aggregating nodes and the required model updates: fewer aggregators reduce exposure but require trust, while frequent local updates increase data leakage.

### 2.4.2 Asynchronous Aggregation

The model aggregation approaches discussed above generally assume that all clients are fully synchronized in the learning process, that is, they are all aware of and, if selected, participate in a given global iteration $t$. This is often an unrealistic approach. First, due to the high number of agents, which makes it is costly to manage a shared view in the entire system. Second, due to the presence of stragglers, that is, agents with limited computational capacity or slow network connectivity, which can result in the server receiving the updated model $\theta_k(t)$ from the agents (line 11) at different times. The common concept to describe stragglers is *staleness*, defined for the server as the difference between the index of the current round global model

update (line 12 of Algorithm 1), and the index $t_k \leq t$ of the local model $\theta_k(t_k)$ received by a straggler $k$. The issue of staleness is considered both for centralized aggregation and for decentralized learning, with the usual assumption of bounded staleness.

**Asynchronous centralized aggregation.**   In the centralized aggregation scheme of (Xie et al., 2020), the global model is updated (line 11) at each asynchronously arriving client update, with weights that consider the staleness of the client. Under a non-i.i.d. data distribution, the models of straggling agents may still hold important updates for the global learning process. Therefore, (Li et al., 2022) suggests that late model updates from stragglers should also be weighted according to the importance of their update for the learning process. However, updating the global model after individual client updates has two disadvantages. It increases the communication requirements and contradicts the efforts for privacy-preserving FL (Nguyen et al., 2022). Therefore, to allow efficient model aggregation despite stragglers, (Chai et al., 2021) introduces a hierarchical scheme, where clients are dynamically clustered according to their update speed, synchronous learning is performed within the clusters, and asynchronous updates across the clusters. The same challenge is addressed in (Wei et al., 2015; Nguyen et al., 2022; Huba et al., 2022), suggesting delaying the global model update until a given number of asynchronous client updates arrive at the central parameter server.

**Asynchronous aggregation for decentralized learning.**   Asynchronous updates with stragglers are also addressed for decentralized learning over graph topologies. The convergence of the learning process is proved in (Lian et al., 2018), assuming bounded staleness, and an averaging process that does not take this staleness into account. Instead, the weight matrix for the averaging is optimized according to the expected staleness of the neighboring nodes in (Bornstein et al., 2023). The comprehensive work (Even et al., 2024) suggests update strategies and proves convergence, considering both staleness due to computation delays and the effect of unreliable communication.

The convergence results of the asynchronous distributed learning schemes show that under bounded staleness, the convergence rate and the accuracy of synchronous systems are preserved, though the level of staleness and delayed aggregated updates may decrease the actual convergence speed.

> **Asynchronous aggregation techniques** address the presence of stragglers, that is, agents with delayed updates due to computational or communication constraints, by relaxing synchronization requirements in both centralized and decentralized settings. Two aggregation strategies emerge based on update timing: immediate aggregation processes each arriving update independently, maximizing responsiveness but increasing communication overhead, while batched aggregation waits for multiple updates before aggregating, reducing communication costs. These strategies can be enhanced through staleness-aware weighting (accounting for update age), hierarchical clustering (grouping agents by speed for mixed synchronous-asynchronous operation), or optimized consensus matrices in decentralized settings (adapting weights to expected neighbor delays). Theoretical analysis under bounded staleness assumptions shows preserved convergence guarantees with degraded convergence speed proportional to staleness levels, establishing the fundamental tradeoff between synchronization flexibility and optimization tightness.
>
> **Trilemma.**   Immediate aggregation leads to effective learning, but compromises resource efficiency and leads to privacy exposure due to the individual updates and the frequent model exchanges. Batched aggregation at the same time increases the convergence times, and potentially discarding valuable straggler information may decrease model accuracy.

### 2.4.3   Communication Efficiency

For collaborative learning, model parameters or gradients must be communicated to and from the parameter server (line 3 and 10 of Algorithm 1) or among the agents in a peer-to-peer or mesh architecture. Since the communication links between the agents and the central server or among the agents may be heterogeneous, unreliable, and costly (as described in Section 2.2.3), the transmission of the information needs to be organized by taking the state of the communication links and the cost of communication into account. The

communication requirements of collaborative learning differ from traditional data transmission, as exact, error-free information is usually not required for learning, and the model aggregation, as (11), needs the function of the inputs, rather than the individual values. These motivate the design of novel, FL-specific communication schemes, including the efficient transmission of information, the organization of medium access in wireless networks, and client scheduling (Hellström et al., 2022).

**Transmission efficiency.** Transmission efficiency measures the importance of the transmitted information versus the invested resources, such as bandwidth and transmission power. Bandwidth usage and learning efficiency in distributed learning can be balanced through quantization, that is, the number of bits representing a model parameter, and model sparsification, that is, the number of model parameters transmitted (Oh et al., 2024; Shlezinger et al., 2021; Xie et al., 2015). Similarly, in wireless networks, power control can achieve a tradeoff between energy efficiency and learning rate (Wang et al., 2020). As randomization can help the learning process, noise and wireless interference are proposed to be tuned dynamically to accelerate learning in (Zhang et al., 2022). To ensure that important model updates are transmitted successfully, (Liu et al., 2019) suggests importance-aware power and retransmission control.

**Wireless communication for model aggregation.** FL often assumes that agents communicate through wireless channels. Several works propose to utilize the superposition property of the wireless multiple access channel to allow all nodes in a wireless cell upload the model parameters at the same time, performing the averaging step of (11) over the air in the analog (Liu et al., 2020; Hellström et al., 2022) or in the digital (Razavikia et al., 2024) domain, including hierarchical federated learning (Azimi-Abarghouyi & Fodor, 2024). Over-the-air computation ensures that the communication costs are independent of the number of nodes. In fully decentralized learning, the broadcast nature of the medium can be used for efficient model exchange among neighbors (Pérez Herrera et al., 2025).

**Client scheduling.** To limit wireless resource usage while ensuring learning accuracy, (Nishio & Yonetani, 2019; Yang et al., 2019) suggests scheduling clients with good channel quality for each global update (line 3 in Algorithm 1). Recognizing that some gradient or model updates have higher importance for the learning process, combined importance and channel-aware client selection is proposed in several works, with different importance metrics (Goetz et al., 2019; Ren et al., 2020; Liu et al., 2021; Leng et al., 2022). Typically, client selection methods aim to maximize the learning accuracy in one global round, under a time constraint. Instead, client selection for minimizing the total learning time is proposed in (Chen et al., 2020; Nishio & Yonetani, 2019), balancing the time of a global round and the improvements achieved.

**Summary:** Communication efficiency techniques exploit the unique properties of collaborative learning, the tolerance to noise, and the aggregation-oriented communication, to reduce transmission costs. They cover three complementary mechanisms: transmission efficiency, wireless-specific aggregation, and client scheduling. These mechanisms target different bottlenecks: transmission efficiency reduces information volume through quantization and sparsification, wireless aggregation exploits channel superposition to perform computation during transmission, while client scheduling selectively activates agents based on communication conditions.

**Trilemma.** Improved efficiency typically trades off learning accuracy (e.g., quantization error, gradient sparsity), system complexity (e.g., synchronized wireless access, channel state information), or fairness (e.g., bias toward agents with better connectivity). Wireless over-the-air computation achieves communication costs independent of agent count but requires careful noise management and synchronization.

### 2.5 Privacy-preserving Collaborative Learning

As we have seen in Section 2.2.4, collaborative learning in its native form does not provide formal privacy guarantees and is thus subject to privacy vulnerabilities. To address these vulnerabilities, the literature has developed both a rigorous framework for quantifying information leakage through *differential privacy (DP)* and privacy-preserving mechanisms that operate at different stages of the collaborative learning process. Considering *honest-but-curious* agents and servers, the two main lines of privacy-preserving mechanisms are *blind computations*, which apply cryptographic techniques that prevent access to intermediate computations, and DP for collaborative learning, which provides statistical guarantees that limit what can be inferred from the evolution of model parameters.

**Blind computations.** If training data is not shared, Algorithms 1 and 2 still require agents to exchange intermediate computations, which may need to be protected from information leakage. These intermediate values, such as local models, gradients, or intermediate representations, still carry information on $S_k$ and can potentially be exploited to infer private information (Fredrikson et al., 2015; Phong et al., 2017; Chai et al., 2020). VFL scenarios are more subject to this information leakage since they require the exchange of intermediate representations of samples, whereas horizontal FL scenarios only aggregate models or gradients (Liu et al., 2024). One approach to prevent such an attack is to run the algorithm using *blind computation* techniques. They provide cryptographic guarantees that the operations on the private variables of the agents are performed without the possibility of reading them. Notable techniques in the literature (Kairouz et al., 2021; Liu et al., 2024) include Trusted Execution Environments (TEEs) (Mo et al., 2021), Homomorphic Encryption (HE) (Chai et al., 2020), and Secure Multi-Party Computation (SMPC) (Liu et al., 2024).

**Differential privacy (DP).** Even without access to raw data or intermediate computations, agents can observe the evolution of the global model parameters $\theta(t)$, which may reveal information about $S_k$. Prior knowledge on the training algorithm can be used to extract private information on $S_k$, for example, by querying the final model $F_{\theta(t)}$ (Fredrikson et al., 2015), or querying the collaborative learning algorithm (Melis et al., 2019; Zhu et al., 2019). To quantify and address these risks, the concept of *differential privacy (DP)* (Dwork et al., 2006; Dwork, 2006) can be applied. Formally, a randomized mechanism $\mathcal{M}$ satisfies $(\varepsilon, \delta)$-DP if for all neighboring sample sets $S$ and $S'$ differing in one sample and all measurable outcome sets $U$, we have:

$$\mathbb{P}[\mathcal{M}(S) \in U] \leq e^{\varepsilon}\mathbb{P}[\mathcal{M}(S') \in U] + \delta \tag{14}$$

where $\varepsilon$ represents the privacy budget and $\delta$ the failure probability. In other words, DP bounds the ability of an adversary to distinguish whether a specific sample was included in $R$, with smaller $\varepsilon$ providing stronger guarantees and $\delta$ allowing for rare violations of this bound. Originally proposed for centralized databases where $S$ represents the database records and $\mathcal{M}$ a query mechanism, DP offers rigorous formal guarantees against information leakage (Dwork & Roth, 2014). Since DP is a general-purpose privacy framework, it can be effectively adapted to any operations that involve the release of aggregate statistics (Duchi et al., 2018; Liu et al., 2022). In machine learning, DP is widely used to protect the training process, where $S$ represents the training dataset and $\mathcal{M}$ the training algorithm (e.g., SGD). Repeated queries, such as the stochastic gradient computations, are privatized using *Differentially Private Stochastic Gradient Descent (DP-SGD)* and its variants (Song et al., 2013; Bassily et al., 2014; Abadi et al., 2016). These methods ensure that an adversary observing the final model parameters $\theta(T)$ cannot reliably infer whether any specific sample $(x, y)$ was included in the training dataset $S$, through a sequence of randomized operations:

- subsampling and shuffling of training data (Kasiviswanathan et al., 2011; Bassily et al., 2014; Abadi et al., 2016)

- clipping or quantization per-sample of the gradients to bound sensitivity (Abadi et al., 2016; Andrew et al., 2021)

- adding calibrated noise scaled to the privacy budget $(\varepsilon, \delta)$ before model update (Dwork et al., 2006; Abadi et al., 2016)

**Differential privacy for collaborative learning.** Applying DP to collaborative learning introduces new challenges due to the distributed nature of the data and the system. In collaborative settings, the global dataset $S$ is partitioned across agents as $S = \cup_{k \in \mathcal{K}} S_k$, making it natural to protect each agent's data as a whole rather than individual samples. Consequently, *agent-level DP* (McMahan et al., 2018; Kairouz et al., 2021), that is, where neighboring datasets differ by one agent's entire local dataset $S_k$ rather than a single sample (Dwork et al., 2006), is typically considered. In this setting, the randomized mechanism $\mathcal{M}$ corresponds to the full collaborative learning algorithm (e.g., Algorithm 1). To achieve this protection, privacy-preserving primitives can be integrated at different levels depending on the privacy threat model:

- *Local DP (LDP).* A first approach involves applying DP techniques directly to the local training loop (lines 6– 9 of Algorithm 1) to privatize the gradient or model parameters $\theta_k(t)$ transmitted to the aggregator, for example, using DP-SGD and its variants. These techniques are designed for the case when participants aim to protect their data from an "honest-but-curious" central aggregator. While LDP provides strong privacy guarantees, it has high communication (Smith et al., 2017; Duchi & Rogers, 2019) and computation demands (Kasiviswanathan et al., 2011; Choi et al., 2018), and leads to the degradation of model estimates (Duchi et al., 2013; 2018; Bhowmick et al., 2018). This limited the practical adoption of LDP techniques (Bonawitz et al., 2022).

- *Distributed DP.* A second approach to protecting the privacy of local data from the central coordinator is to use distributed DP techniques. These methods combine mild LDP mechanisms with server-side primitives for aggregating local statistics, which amplify this privacy level (Evfimievski et al., 2003). *Privacy amplification* is typically achieved through randomized shuffling and subsampling of the received LDP records (Bittau et al., 2017; Erlingsson et al., 2019; Cheu et al., 2019; Balle et al., 2020).

  Adapting these distributed DP principles to collaborative learning presents specific challenges. The goal is to perform distributed training while achieving agent-level DP. However, naively distributing DP-SGD is not possible, as the server lacks sampling control over local training data and local model updates. Therefore, DP variants of *FedAvg* (Algorithm 1) have been proposed that balance effective learning, privacy, and communication efficiency (McMahan et al., 2018; Agarwal et al., 2018). These variants modify model aggregation (lines 11– 12) by incorporating server-side primitives: clipping (Andrew et al., 2021), scaling (Geyer et al., 2017), shuffling (Geyer et al., 2017; Pihur et al., 2022), and subsampling (Geyer et al., 2017; Pihur et al., 2022; McMahan et al., 2018; Talwar et al., 2024) of the received gradients or model parameters. These server-side primitives are coupled with mild local DP mechanisms, including additive noise and gradient quantization (McMahan et al., 2018). The current state-of-the-art technique, *Differential Privacy-Follow The Regularized Leader (DP-FTRL)* (Kairouz et al., 2021), proposes a different approach. It avoids sampling and shuffling by combining tree aggregation (Dwork et al., 2010; Chan et al., 2011) with correlated noise through online estimate regularization (Duchi et al., 2011).

  Distributed DP techniques have been demonstrated for large-scale federated learning deployments at Google (Ramaswamy et al., 2020; Xu et al., 2023), Apple (Apple, 2023), and Microsoft (Ding et al., 2017).

While most existing work focuses on privacy for centralized model aggregation, (El Mrini et al., 2024; Cyffers et al., 2024; Biswas et al., 2025) extend privacy mechanisms to decentralized architectures with gossip learning. To enable DP in real systems, recent research proposes DP solutions with transparent model aggregation (Talwar et al., 2024; Daly et al., 2024) and inference (Shumailov et al., 2025), and verifiable privacy mechanisms (Daly et al., 2024).

> **Privacy-preserving techniques** address the gap between keeping data local and achieving true privacy. They represent two complementary paradigms: *blind computations* and *differential privacy (DP)*. These paradigms address distinct threat models: blind computation uses cryptographic primitives (TEE, HE, SMPC) to prevent adversaries from accessing intermediate values during computation, while DP performs calibrated randomization and provides formal statistical bounds on what can be inferred from observing the model itself, even under full observability of the training process. Distributed DP techniques achieve practical agent-level protection through server-side amplification mechanisms that strengthen mild local protections while maintaining efficiency.
>
> **Trilemma.** Privacy protection trades off learning effectiveness (e.g., noise-induced accuracy degradation, bias from gradient clipping, reduced statistical efficiency due to subsampling). It also trades off system efficiency, by increasing computational overhead (e.g., cryptographic operations, agent- and server-side processing), communication cost (e.g., verbose or repeated noisy updates under local mechanisms), and reducing robustness to asynchronicity (e.g., complex multi-agent protocols).

## 3 Collaborative Learning on Graph-structured Data

We have reviewed the main design choices of collaborative learning on Euclidean data in Section 2. We now extend these concepts to the case of *graph-structured data*. Since graph-structured data introduces unique characteristics and challenges, we first provide an overview of state-of-the-art solutions for *learning on graph-structured data* in Section 3.1. Building on this foundation, we apply the same analytical structure used for *Euclidean data* to the *graph domain*. We begin by formulating the *collaborative learning problem* for graph-structured data in Section 3.2, then examine the different types of *heterogeneity* that emerge in Section 3.3. Subsequently, we structure our analysis of design choices along the same *trilemma* identified in Section 2 and formalized in Figure 7: *learning effectiveness* in Section 3.4, *efficiency* in Section 3.5, and *privacy preservation* in Section 3.6. Table 2 summarizes the papers surveyed along these three lines.

### 3.1 Machine Learning on Graphs and GNNs

**Problem definition.** Machine Learning on graphs aims to develop models that operate on graph-structured data to solve tasks defined on graphs and generalize to unseen graph instances. Graphs are a fundamental data structure for modeling relationships between entities, which arise naturally in numerous domains such as social networks, molecular chemistry, and networked systems. Unlike traditional data residing in Euclidean spaces (e.g., images, text sequences), graphs exhibit irregular, non-Euclidean structure requiring specialized methods to process them effectively.

Formally, a graph is defined as $\mathcal{G} = (\mathcal{V}, \mathcal{E})$, where $\mathcal{V}$ is the set of *vertices*, also called *nodes*, and $\mathcal{E} \subseteq \mathcal{V} \times \mathcal{V}$ the set of *edges*, also called *links*. Given a fixed indexing of the nodes $\mathcal{V} = \{v_1 \ldots, v_N\}$, the graph structure can be conveniently encoded in an equivalent[4] matrix form as an adjacency matrix $A \in \mathbb{R}^{N \times N}$, where $a_{ij} = 1 \iff (v_i, v_j) \in \mathcal{E}$.

In practice, graphs often include auxiliary information. Most commonly, nodes are associated with features $\{x_v \in \mathcal{X}\}_{v \in \mathcal{V}}$, organized into a feature matrix $X \in \mathbb{R}^{N \times d_x}$, where $d_x$ denotes the feature dimension. Depending on the task, label information may also be available: node-level labels for classification (e.g., user characteristics), edge-level labels for link prediction (e.g., friend recommendation), and graph-level labels for graph classification (e.g., molecule property).

A *graph sample* can then be expressed as $(G, y)$ where $G = (A, X)$ represents the *graph data* in its matrix form and $y$ the label associated with the task. When learning on graph data, we consider that we observe $n$ graph data samples drawn from an unknown distribution $\mathcal{D}$, which together form a dataset $S = \{G^i, y^i\}_{i=1}^n$.

---

[4]A graph $\mathcal{G} = (\mathcal{V}, \mathcal{E})$ corresponds to an equivalence class of adjacency matrices $A \in \mathbb{R}^{N \times N}$, where two matrices are equivalent if they are related by a permutation of vertex indices; that is, for any permutation matrix on $\Pi \in \mathbb{R}^N$, we have $\Pi A \Pi^\top \sim A$.

| Case | Subgraph | | | | | | | | Multiple Graph Instances | |
|---|---|---|---|---|---|---|---|---|---|---|
| Graph type | Homogeneous | | | | Heterogeneous | | | | Homogeneous or heterogeneous | Heterogeneous |
| Feature partition | Horiz. | | Vert. | | Horiz. | | | | Horiz. | Horiz. |
| Type partition | Not applicable | | Not applicable | | No | | Yes | | Possible | Yes |
| Topology partition | Horiz. | Horiz. | Vert. | Vert. | Horiz. | Horiz. | Horiz. | Horiz. | No partitioning | Vert. |
| Granularity | Subgraph | Ego | edge-full | edge-free | Subgraph | Ego | Subgraph | Ego | Not applicable | Not applicable |
| Resulting graph cut | Edge | Edge | Vertex | Vertex | Vertex | Vertex | Vertex | Vertex | Not applicable | Not applicable |
| Representative work | **436** | **382** | **283** | **67** | **158** | **400** | **304** | **355** | **162** | **89** |
| **Improving Learning and Inference Effectiveness (Section 3.4)** | | | | | | | | | | |
| **Data-based techniques (Section 3.4.1)** | | | | | | | | | | |
| Data Augmentation | 436; 307; 303 | 235; 138 | | | | **4; 54**; 63; 249 315; 54 | 258 | | 148; 438; 129; 362 | |
| Local Regularization | 364 | | 126 | | | | 64 | | **362; 207**; 11; 258 48 | |
| Subsampling | 56; 244 | 296; 91 | | | | | 304 | | 48; 363; 340 | |
| **Model-based techniques (Section 3.4.2)** | | | | | | | | | | |
| Personalization of embeddings or parameters | 232 | | | | 158 | 63; 4; 391; 316 | | | **11; 438**; 376; 410 113; 340 | |
| Architecture Personalization | | 269; 211 | **126**; 146; 258 | | | | | | **340**; 162; 438; 113 | |
| **Training and Inference under aligned partitions (Section 3.4.3)** | | | | | | | | | | |
| Aggregating node representations | 217 | 445 | 283; 64 | 67 | 158 | **400**; 63; 399; 435 315; 316; 4 391 | | | 67 | |
| Aggregating graph representations | | | | | | | | | | 89 |
| Transfer techniques | | | **258**; 258 | | | | | | | |
| **Improving Communication and Computation Efficiency (Section 2.4)** | | | | | | | | | | |
| **Aggregation patterns (Section 2.4.1)** | | | | | | | | | | |
| Centralized (Models) | 422 | | 53 | | | **400**; **4**; 249; 315 54; 304; 399; 400 244; 414; 391; 316 158; 435; 63 | 64; 258; 304 | | **340**; **410**; **11**; 376 438; 162; 113 129; 48; 363; 148 67 | |
| Centralized (Embeddings/Features) | **422** | 91 | 53; 283; 61 | | | **63**; 304; 399; 400 244; 54; 4; 414 391 | | | | |
| Decentralized Aggregation (Models) | **335**; **128**; 244; 445 393; 150 | 138; 268; 382; 292 | | | | | | | 162 | |
| Decentralized (Embeddings) | **429**; **28**; 429; 217 316; 393; 150 | **296**; **277**; 132; 447 295; 394; 138; 216 349; 131; 130; 214 211; 133; 292; 268 382 | | | | | | 355 | | |
| Hierarchical (Models) | **150** | | | | | | | | | |
| Hierarchical (Embeddings) | | **277**; 296; 277 | | | | | | | | |
| **Asynchronicity (Section 3.5.2)** | | | | | | | | | | |
| Stale model aggregation | | | | | | | | | | |
| Stale Message Passing | | **349**; **109**; 261; 128 394 | | | | | | | | |
| **Communication (Section 3.5.3)** | | | | | | | | | | |
| Transmission Efficiency | | **214**; 132; 382 | | | | | | | | |
| Wireless communication for aggregation | 28 | **131**; 394; 130; 216 123 | | | | | | | | |
| Client scheduling and topology management | 244; 429; 277; 335 | 382 | | | | | | | | |
| **Privacy (Section 3.6)** | | | | | | | | | | |
| For model parameters | 422 | | | | | 249; 400; 414 | | | | |
| For exchanged embeddings | 303 | 235; 296; 216 | 53; 283; 61; 62 | | | **400**; 399; 315; 414 445; 249 | | | 438 | |
| For topology | 56; 150; 163 | 235; 296; 303 | 283 | | | **414**; **4**; **315**; 399 400; 63; 158; 249 435; 431; 64 | | | 438 | |

Table 2: **The landscape of collaborative learning techniques on graph data organized by graph partition.** The table presents the design choices and techniques employed by different approaches across various graph partitioning schemes. Each row represents a specific technique or design choice, while columns correspond to different graph partitioning strategies categorized by graph type, feature partition, and granularity. Citations are placed in columns according to the partitioning approach used by each work, and representative works are marked in bold.

The goal is then to learn a model $F$ among our class of models $\mathcal{F}$ which minimizes the expected loss

$$\mathcal{L}(F) := \mathbb{E}_{(G,y)\sim\mathcal{D}}\ell(F(G), y). \tag{15}$$

**Graph representation learning.** Unlike traditional data residing in Euclidean spaces (e.g., images, text sequences), graphs exhibit irregular, *non-Euclidean* structure where nodes have variable numbers of neighbors and no canonical ordering, requiring permutation equivariant (or invariant) representations (Bronstein et al., 2021). *Graph representation learning* addresses this by learning mappings that embed graph data into fixed-dimensional Euclidean vector spaces $\mathbb{R}^{d_z}$ while preserving both topological structure $A$ and node attributes $X$. Specifically, the goal is to learn a mapping that embeds vertices, edges, or full graphs into $\mathbb{R}^{d_z}$ such that the embeddings preserve both topological structure and node attributes (see Figure 11). This mapping typically consists of a composition of two functions. First, a learnable node encoder $\phi_{node}$ maps each node $v$ in graph $G$ to an embedding $z_v := \phi_{node}(G, v) \in \mathbb{R}^{d_z}$ that captures both structural and feature information about node $v$. These node embeddings $\{z_v; v \in \mathcal{V}\}$ then serve as building blocks for edge and graph embeddings, depending on the task:

- The edge embeddings can be computed by an encoder $\phi_{link}(z_u, z_v)$ (e.g., a simple concatenation), which can also be learnable (e.g., via an MLP).

- The graph embedding can be computed by an encoder on the multiset of all node representations $\phi_{graph}(\{\{z_v, v \in \mathcal{G}\}\})$ (e.g., a mean), which can be learnable (e.g., via an attention layer).

The complete encoder model $\phi$ thus depends on the task at stake. In most models, only the node encoder is learnable, and $\phi_{node}$ is parametrized by these learnable weights $\theta \in \Theta$, expressed by the notation $\phi_\theta$. When labels are available, a predictor $f_\psi$, parameterized by $\psi \in \Psi$, is added to the output of the encoder such that $F_{\psi,\theta}$ is the complete model. The problem is then to identify $F_{\psi^*,\theta^*}$ which minimizes

$$\mathcal{L}(F_{\psi,\theta}) := \mathbb{E}_{(G,y)\sim\mathcal{D}}\ell(F_{\psi,\theta}(G), y). \tag{16}$$

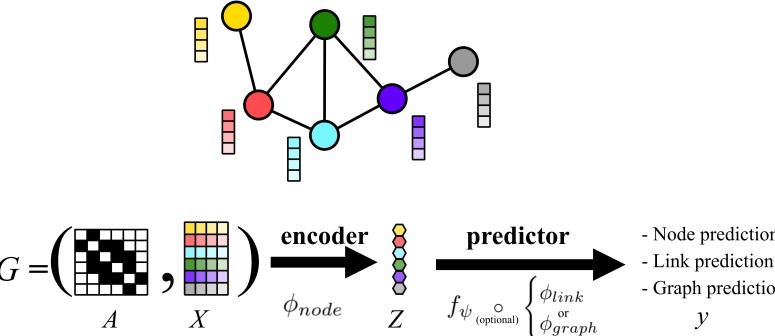

Figure 11: **Graph representation learning.** The objective is to learn a node encoder $\phi_{node}$ that embed the graph data $G = (A, X)$, consisting of the adjacency matrix $A \in \mathbb{R}^{N \times N}$ and the feature matrix $X \in \mathbb{R}^{N \times d_x}$, into a node representation matrix $Z \in \mathbb{R}^{N \times d_z}$. These node representations are then used for predictive tasks at the node, link, or graph level, with the use of a predictor $f_\psi$ and, depending on the task, a link $\phi_{link}$ or graph $\phi_{graph}$ encoder.

**Graph Neural Networks (GNNs).** Graph Neural Networks represent the dominant node encoder $\phi_{node}$ architecture. Two architectural paradigms are commonly distinguished (Wu et al., 2020):

- *Spectral GNNs* apply spectral filters to the graph. They operate on the full spatial domain and a truncated spectral domain (Hammond et al., 2011).

- *Spatial GNNs* apply spatial filters to the graph. They operate on the truncated spatial domain and the full spectral domain (Kipf & Welling, 2017).

Modern GNN architectures predominantly adopt the *Message Passing Neural Network (MPNN)* framework (Gilmer et al., 2017) (see Figure 12), which is a spatial approach to define $\phi_{node}$. To encode a node $v \in \mathcal{V}$ in a representation $z_v$, an MPNN follows an iterative process that aggregates the information of $v$'s $L$-hop neighborhood. This neighborhood, denoted as $G^v$, represents the so-called $L$-hop *ego-graph* centered on node $v$. The iterative process is as follows:

- *Feature encoding:*
$$h_v^{(0)} \leftarrow f_{Enc}(x_v) \tag{17}$$

- *Message Passing:* for each layer $l = 0, \ldots, L - 1$,

$$m_{v,u}^{(l+1)} = f_{Mes}^{(l+1)}(h_v^{(l)}, h_u^{(l)}) \qquad \text{(message computation)} \tag{18}$$

$$m_v^{(l+1)} = f_{Agg}^{(l+1)}(\{\{m_{v,u}^{(l+1)}, u \in \mathcal{N}(v)\}\}) \qquad \text{(message aggregation)} \tag{19}$$

$$h_v^{(l+1)} = f_{Up}^{(l+1)}(h_v^{(l)}, m_v^{(l+1)}) \qquad \text{(state update)} \tag{20}$$

where $\mathcal{N}(v)$ denotes the set of 1-hop neighbours of $v$ in $\mathcal{G}$, $f_{Enc}, f_{Mes}^{(l+1)}, f_{Up}^{(l+1)}$ are neural functions (e.g., MLPs), and $f_{Agg}^{(l+1)}$ is a permutation-invariant operator on multisets (e.g., sum, mean, attention).

- *Final Embedding:*
$$z_v := h_v^{(L)} = \phi_{node}(G^v, v). \tag{21}$$

The full encoder $\phi_\theta$ can be considered to be parametrized by $\theta$, representing all learnable weights across message, aggregation, and update functions and across all layers $l = 0, \ldots, L$. The framework is modular: different architecture choices for $f_{Mes}$, $f_{Agg}$ and $f_{Up}$ define the most famous GNN architectures (e.g., GCN (Kipf & Welling, 2017), GraphSAGE (Hamilton et al., 2017), GATs (Veličković et al., 2018; Brody et al., 2022)). The resulting node embeddings $\{z_v\}_{v \in \mathcal{V}}$ can then be fed to the predictor for solving downstream tasks.

While the MPNN framework described above represents the modern approach to GNNs, *spectral GNNs* were the historical approach, motivating the design of architectures from spectral graph theory (Bruna et al., 2013). Given the spectral decomposition of the symmetric normalized graph Laplacian $\tilde{L} = I - D^{-\frac{1}{2}}AD^{-\frac{1}{2}}$ as $\tilde{L} = U\Lambda U^\top$, where $U \in \mathbb{R}^{N \times N}$ is the orthogonal matrix of eigenbasis and $\Lambda = \text{diag}(\lambda_1, \ldots, \lambda_N)$ is the diagonal matrix of eigenvalues, spectral GNNs define convolution through a spectral filtering operation $*$ given by

$$g * X = U g(\Lambda) U^\top X, \tag{22}$$

where $g : [0, 2] \rightarrow \mathbb{R}$ is a spectral filter (Hammond et al., 2011). However, this requires expensive eigendecomposition and filtering operations, and the decomposition must be recomputed for graph modifications. To address these limitations, practical spectral GNNs approximate spectral filters using learnable polynomial functions of the normalized Laplacian $\tilde{L}$,

$$g(\tilde{L}) = \sum_{l=0}^{L} \alpha_l \tilde{L}^l, \qquad Z = g(\tilde{L})H^{(0)}, \tag{23}$$

with learnable coefficients $\{\alpha_l\}_{l=0}^{L}$. This polynomial formulation induces iterative information propagation, with each power $\tilde{L}^l$ aggregating features from $l$-hop neighborhoods, thereby reducing to spatial-based Message Passing Neural Networks (MPNNs). Various polynomial bases have been explored (e.g., Cheb-Nets (Defferrard et al., 2016; He et al., 2022), BernNet (He et al., 2021), APPNP (Gasteiger et al., 2019), JacobiNet (Wang & Zhang, 2022), and GPRGNN (Chien et al., 2021)), each exploiting the recurrence relations of its polynomial basis to efficiently compute $g(\tilde{L})$ through 1-hop message passing operations.

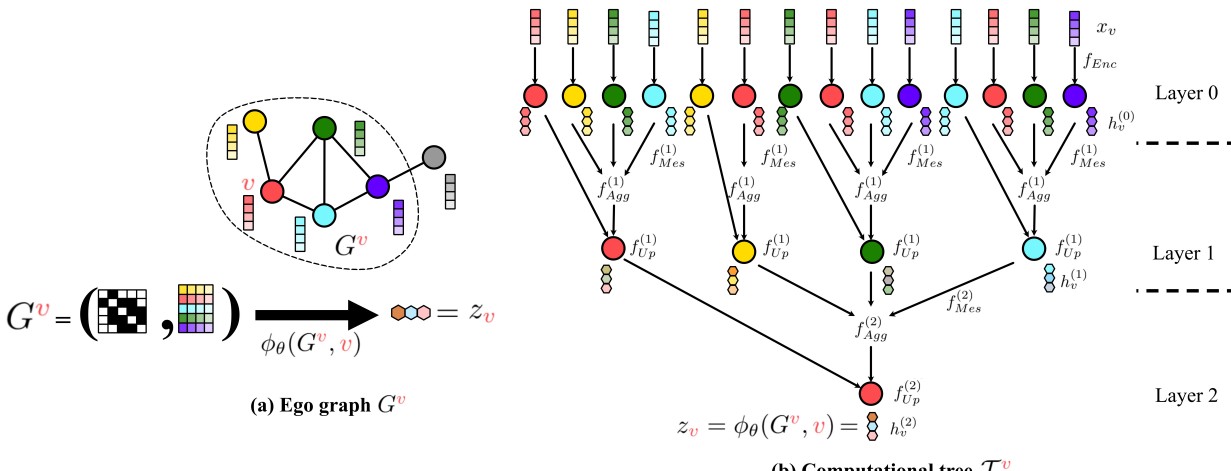

$G^v = \left(\begin{array}{c}\blacksquare\end{array}, \begin{array}{c}\end{array}\right) \xrightarrow{\phi_\theta(G^v, v)} \hexagon\hexagon\hexagon = z_v$

**(a) Ego graph** $G^v$

**(b) Computational tree** $\mathcal{T}^v$

Figure 12: **Computations in a Message Passing Neural Network (MPNN)** $\phi_\theta$. To infer the representation $z_v$ of a node $v$, the MPNN takes as input the ego-graph $G^v$ centered on $v$, illustrated in (a) for the case of a 2-layer MPNN. The inference process consists of a sequence of hierarchical steps, given by (17),(18),(19), and (20), which together form the computational tree $\mathcal{T}^v$ shown in (b). In practice, these operations can be efficiently parallelized to infer the representations $z_v$ of a batch of nodes.

**GNN training.** GNNs can be trained to approximate the solution of (16) on empirical samples by minimizing the loss

$$\hat{\mathcal{L}}(F_{\psi,\theta}) = \frac{1}{|S|} \sum_{(G,y) \in S} \ell(F_{\psi,\theta}(G), y). \tag{24}$$

Minimizing (24) is commonly done using the SGD algorithm and neural backpropagation techniques (Rumelhart et al., 1986; Hamilton et al., 2017), often augmented with self-supervised objectives such as link prediction (Hamilton et al., 2017) or with contrastive learning (Veličković et al., 2019; You et al., 2020). In practice, most approaches minimize the empirical risk (24) as a tractable proxy for the true risk (16), assuming that the resulting model generalizes well, i.e., that the generalization gap is small (Jegelka, 2022).

**Interest for distributed systems** A notable characteristic of the MPNN architecture is that both the model parameters and the computed gradients need to be propagated along the edges of the graph. MPNNs share strong similarities with distributed message passing algorithms (Papp & Wattenhofer, 2022), which iteratively aggregate messages from their neighbors, update their state, and then pass on the new states to their neighbors again. This suggests that MPNNs are a natural fit for Machine Learning tasks in collaborative systems.

### 3.2 Collaborative Learning on Graph-structured Data Problem Definition

As pointed out for Euclidean data in Section 2, the term distributed ML often encapsulates two techniques: parallel training on distributed machines and training on private distributed data. Our settings focus on the latter, where the graph data is collected locally by different agents. For the parallel training case, we refer the reader to (Shao et al., 2024).

We now extend the collaborative learning framework introduced in Section 2.1 to learning on graph data. Two main problem settings are distinguished: *learning from multiple graph instances* and *learning from subgraphs*, as shown in Figure 13.

**Learning from multiple graph instances.** We consider a set of agents $\mathcal{K} = \{1, \ldots, K\}$, where each agent $k$ holds a dataset $S_k$ containing $n_k$ graphs sampled from an unknown local distribution $\mathcal{D}_k$. For

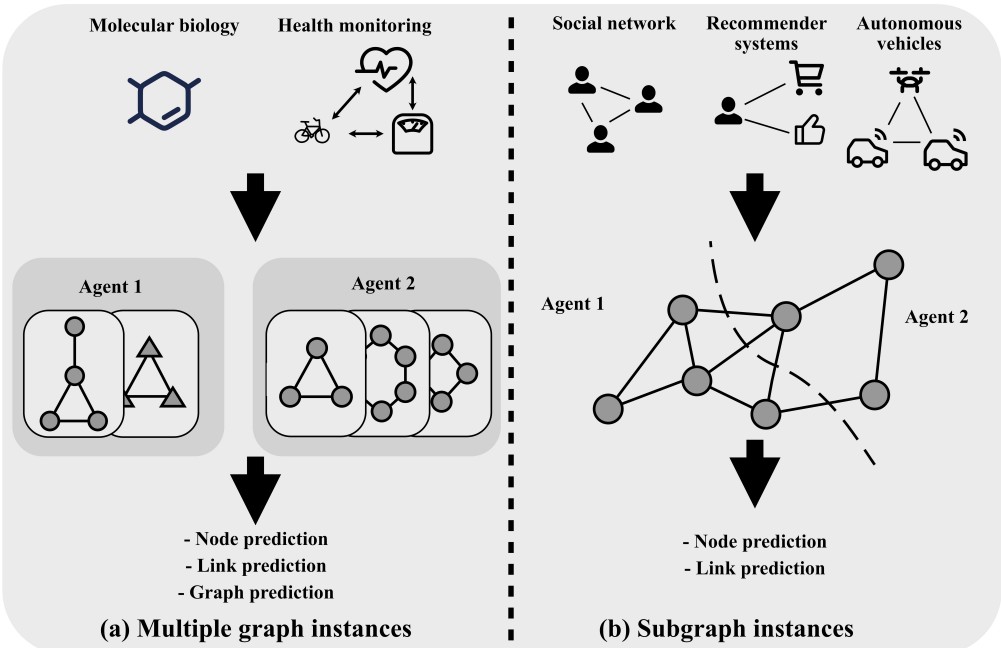

Figure 13: **Two main settings of Collaborative Learning on Graph Data.** Agents hold either (a) multiple independent graph instances or (b) disjoint subgraphs of a single global graph.

example, the dataset can be made of numerous molecule graphs known by different labs. Usually, graph-level tasks are considered for this case, for example, predicting a property of a molecule graph (Xie et al., 2021). Each agent aims to minimize the local objective

$$\mathcal{L}_k(F_{\psi,\theta}) = \mathbb{E}_{(G,y)\sim\mathcal{D}_k}\ell(F_{\psi,\theta}(G), y). \tag{25}$$

Agents can compute local estimates of the local objectives

$$\hat{\mathcal{L}}_k(F_{\psi,\theta}) = \frac{1}{n_k}\sum_{(G,y)\in S_k}\ell(F_{\psi,\theta}(G), y). \tag{26}$$

One can notice that the local objectives (25) (26) are formulated in the same form as for Euclidean data in (1) (2), with the difference here that the data distribution $\mathcal{D}_k$ operates on graph data. As all the other aspects are the same as in Euclidean settings, if the distributions $\mathcal{D}_k$ are i.i.d., then the *FedAvg* algorithm can be used to effectively identify the optimal GNN parameters, the solution of (3).

**Learning from subgraphs.** We now consider the subgraph case. We consider a global graph $\mathcal{G} = (\mathcal{V}, \mathcal{E})$ where each agents $k \in \mathcal{K}$ observes a set of $N_k$ nodes and the edges connecting these nodes $\mathcal{E}_k = \{(u,v) \in \mathcal{E} | u, v \in \mathcal{V}_k\}$ $\mathcal{V}_k \subset \mathcal{V}$, forming a subgraph topology $\mathcal{G}_k = (\mathcal{V}_k, \mathcal{E}_k)$. Each agent collects subgraph data $G_k$ related to its observed nodes $\mathcal{V}_k$, and some labels $y$ associated with the task. Learning for recommender systems, large knowledge graphs, social networks, or large-scale infrastructures, but also learning in groups of interactive objects, falls into this category. The learning tasks are typically on the node, link, or subgraph level.

Recall that the GNN requires, for each node $v$, the data from the $L$-hop neighborhood of $v$ as formulated in (21). The local empirical objective for agent $k$ is then expressed not on the graph data $G_k$, but on the graph data supported on $\mathcal{G}_k$ expanded by an $L$-hop neighborhood within the global graph. We denote this expanded topology by $\tilde{\mathcal{G}}_k$, and the corresponding data supported on it by $\tilde{G}_k$

$$\hat{\mathcal{L}}_k(F_{\psi,\theta}) := \ell(F_{\psi,\theta}(\tilde{G}_k), y). \tag{27}$$

If agent $k$ does not observe $\tilde{G}_k$, then it does not have the necessary data to minimize its local loss (27). This makes the collaboration of the agents that hold parts of $\tilde{G}_k$ necessary, a phenomenon illustrated in Figure 14.

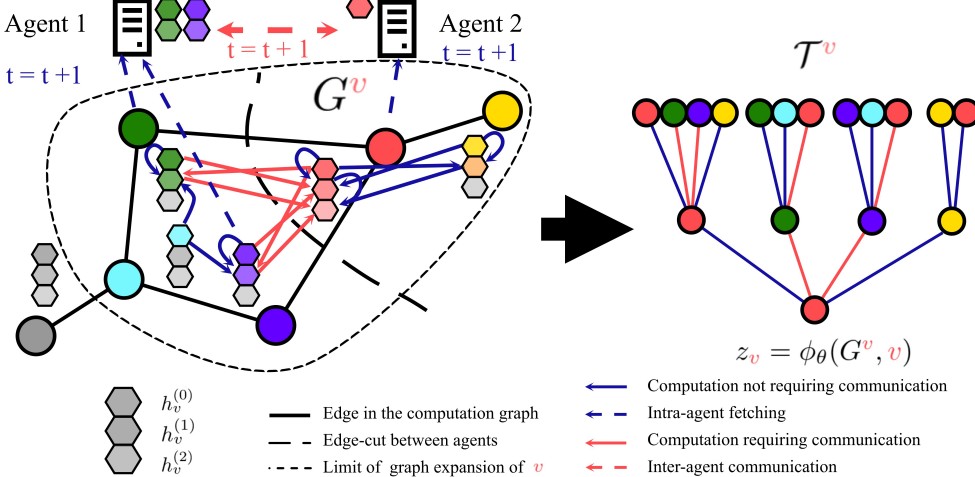

Figure 14: **Collaborative inference in a subgraph.** In the subgraph setting, inference becomes collaborative as described in Algorithm 3, illustrated here for the inference of the red node $v$'s representation based on its $L$-hop ego-graph $G^v$. The computational tree $\mathcal{T}^v$ shows how computations can be distributed across multiple agents, requiring the exchange of intermediate results between them, hence the term *collaborative inference.*

This need for collaborative inference resembles the setting of vertical partitions described in Section 2.3.3 and leads to the following observation: *collaborative learning on subgraph data borrows from the principles of Vertical Federated Learning.* If we assume that $(\tilde{G}_k, y)$ are sampled from an unknown distribution $\mathcal{D}_k$, the true local objective can be formulated in a collaborative form

$$\mathcal{L}_k(F_{\psi,\theta}) := \mathbb{E}_{(\tilde{G}_k y) \sim \mathcal{D}_k} \ell \left( F_{\psi,\theta}(\tilde{G}_k), y \right). \tag{28}$$

An effective solution to minimize the local objectives (9) is to collaboratively train a common GNN $F_{\psi,\theta}$ to minimize the average of local objectives across agents

$$\frac{1}{K} \sum_{k \in \mathcal{K}} \mathcal{L}_k(F_{\psi,\theta}). \tag{29}$$

This formulation is analogous to the Horizontal FL problem formulation (2) with the difference that the local losses $\mathcal{L}_k$ are collaborative, and thus *collaborative learning on subgraph data borrows from the principles of Horizontal Federated Learning.*

A GNN can be trained to minimize (29) in collaborative settings by following two algorithmic directions:

- With *isolated inference*, inference is performed on the local subgraph only. This case borrows directly from the Horizontal FL Algorithm 1.

- With *collaborative inference*, inference requires the information exchange between the agents. This case borrows directly from the Vertical FL Algorithm 2.

A possible unified implementation of federated training from subgraph data is described in Algorithm 3, following the concepts from (Chen et al., 2021; Wu et al., 2022; Du & Wu, 2022; Chen et al., 2022; Yao et al., 2023; Han et al., 2024). It is a round-based algorithm which outputs both the model parameters $\psi(T), \theta(T)$,

and the node representations $\{z_v(T)\}_{v \in \mathcal{V}}$ (line 25) at the end of the training round. The algorithm builds on the centralized aggregation of Algorithm 1, where model parameters iterate $\psi(t)$ and $\theta_k(t)$ are aggregated by a central server (line 22). In the subgraph setting, however, the server also assumes additional responsibilities: it stores the node representation iterates $z_v(t)_{v \in \mathcal{V}}$ (line 23) and allows the exchange of node embeddings between agents during collaborative inference (lines 11 and 12).

In the algorithm, for agents $k \in \mathcal{K}$, $G_k$ is the local subgraph data defined on $\mathcal{V}_k$, the set of nodes held by the agent $k$. $G_k^v$ is the part of the ego-graph data held by agent $k$ and related to $v$. We also need to define boundary nodes, that is, nodes that are connected to nodes held by different agents. For each agent $k$, the set of foreign boundary nodes is the set of nodes held by a different agent and connected to a node seen by $k$: $\mathcal{V}_{f,k} = \{v; \exists u \in \mathcal{V}_k \mid (u,v) \in \mathcal{E}, v \in \mathcal{V} \setminus \mathcal{V}_k\}$. The set of private boundary nodes is the set of nodes private to $k$ that are foreign nodes of other agents: $\mathcal{V}_{p,k} = \bigcup_{l \in \mathcal{K} \setminus k} \{v \in \mathcal{V}_k \mid v \in \mathcal{V}_{f,l}\}$.

A round of Algorithm 3 begins with the broadcast of common model parameters to all agents (line 2), which then initialize their local model parameters accordingly (line 4). Two inference modes are considered:

- *Isolated inference:* each agent performs inference solely on its local subgraph data $G_k$ (line 6).

- *Collaborative inference:* agents collectively replicate the inference process of a message-passing neural network (MPNN) in a distributed fashion. This is executed iteratively over the $L$ layers of the MPNN. At each iteration $l$, agents communicate (line 11), and request (line 12) the $l$-th layer embeddings of their foreign boundary nodes $\mathcal{V}_{f,k}$ through the server, then compute the $(l+1)$-th embeddings using the MPNN inference equations (18), (19), and (20).

The remaining steps are common to both inference modes. Node representations are sent to the server for aggregation (lines 17–23). Local model parameters are updated using gradient descent based on the computed loss (line 19), and subsequently sent to the server for aggregation (line 11).

### 3.3 Graph Data Partition, Heterogeneity and Privacy Challenges

The homogeneity of graph data collected across agents is central to the problem setup in (25)–(28). However, data heterogeneity, which is well-studied for Euclidean data in Section 2.2, also challenges collaborative learning in the graph setting. We now describe graph-specific heterogeneity from two angles: data partitioning and statistical heterogeneity.

### 3.3.1 Graph Partition Taxonomy

A key feature of distributed learning is how the data is partitioned across agents. We subsequently propose a characterization of the different partitions for graph data appearing in the literature. To structure our analysis, we build on the taxonomy for Euclidean data partitions (Section 2.2.1) and adapt it to graph data, following prior efforts in (Zhang et al., 2021; He et al., 2021; Liu et al., 2025). We continue to discuss the two main cases introduced in Section 3.2: (i) agents holding multiple graph instances, and (ii) agents holding a subgraph as described in Figure 13.

**Multiple graph instances.** We examine the case where agents observe full graph instances as introduced in Section 3.2. To formalize, we assume the existence of a global dataset $S$ made of full graph instances $S = \{(G^i, y^i)\}_{i \in \mathcal{I}}$. We distinguish two ways in which graph instances are typically distributed across agents, illustrated in Figure 15:

- *Horizontal topology partition* analogous to the taxonomy of *inter-graph FL* (Xie et al., 2021) or *graph level FL* (He et al., 2021). Similar to horizontal partitioning seen in Section 2.2.1, the index $\mathcal{I}$ of $S$ is partitioned across agents into local indexes $\mathcal{I}_k \subset \mathcal{I}$ such as $\cup_{k \in \mathcal{K}} \mathcal{I}_k = \mathcal{I}$, yielding datasets for agent $k$:
$$S_k = \left\{((A^i, X^i), y^i)\right\}_{i \in \mathcal{I}_k}.$$

---

[5]In practice, this step can be efficiently computed in parallel using vectorization libraries for GNN such as PyTorch Geometric (Fey et al., 2025).

---

**Algorithm 3** Federated training of subgraph GNN

---

**Require:** $\mathcal{K} = \{1, \dots, K\}$ set of agents , $F_{\psi,\theta}$: a GNN parametrized by $\psi \in \Psi, \theta \in \Theta$, global graph topology $\mathcal{G} = (\mathcal{V}, \mathcal{E})$, $\{\mathcal{V}_k\}_{k \in \mathcal{K}}$ subset of nodes $\mathcal{V}$ held by each agent, foreign boundary nodes $\{\mathcal{V}_{p,k}\}_{k \in \mathcal{K}}$ known by each agent, private boundary nodes $\{\mathcal{V}_{f,k}\}_{k \in \mathcal{K}}$ known by each agent, boundary edges $\{\mathcal{E}_k\}_{k \in \mathcal{K}}$ known by each agent, learning rate $\eta$, initial model parameters $\{\psi(0), \theta(0)\}$ and initial node embeddings $\{z_v(0)\}_{v \in \mathcal{V}}$, local graph data $\{G_k\}_{k \in \mathcal{K}}$ held by each agent, number of training rounds $T$, boolean *isolated inference, ollaborative inference*

      — **Training loop** —

1: **for** each round $t = 1, \dots, T$ **do**

    Server:

2:     Broadcast current model parameters $\theta(t-1), \psi(t-1)$ to all agents

3:     **for** each agent $k \in \mathcal{K}$ in parallel **do**

        — **Local Model Initialization** —

4:         Initialize $F_{\psi_k(t-\frac{1}{2}),\theta_k(t-\frac{1}{2}),}$ such as $\theta_k(t-\frac{1}{2}) \leftarrow \theta(t-1)$, $\psi_k(t-\frac{1}{2}) \leftarrow \psi(t-1)$

        — **Inference Phase** —

5:         **if** *isolated inference* **then**

            — **Local Inference on Isolated Subgraph** —

6:             Decompose $G_k$ into $\{G_k^v\}_{v \in \mathcal{V}_k}$ and compute $z_v(t - \frac{1}{2}) = \phi_{\theta_k(t-\frac{1}{2})}(G_k^v, v)$ for each node[5]$v \in \mathcal{V}_k$ as in (21)

7:         **end if**

8:         **if** *collaborative inference* **then**

            — **Collaborative Inference** —

9:             Compute $\{h_v^{(0)}\}_{v \in \mathcal{V}_k}$ as in (17) and set $\forall v \in \mathcal{V}_k, h_v^{(0)}(t) \leftarrow h_v^{(0)}$

10:           **for** $l = 0, \dots, L-1$ **do**

11:             Send $\{h_v^{(l)}(t)\}_{v \in \mathcal{V}_{p,k}}$ to the server

12:             Request $\{h_v^{(l)}(t-1)\}_{v \in \mathcal{V}_{f,k}}$ from the server

13:             Upon reception, compute (18) (19) (20) for each[5] $v \in \mathcal{V}_k$

14:           **end for**

15:           $z_v(t - \frac{1}{2}) \leftarrow h_v^{(L)}$ for all $v \in \mathcal{V}_k$

16:         **end if**

17:         Send $\{z_v(t - \frac{1}{2})\}_{v \in \mathcal{V}_k}$ to the server

        — **Loss Computation** —

18:         Compute $\hat{\mathcal{L}}_k \left( F_{\psi_k(t-\frac{1}{2}),\theta_k(t-\frac{1}{2})} \right)$

        — **Local Model Update** —

19:         $\theta_k(t) \leftarrow \theta_k(t - \frac{1}{2}) - \eta \cdot \frac{\partial \hat{\mathcal{L}}_k}{\partial \theta_k^{(l)}(t-\frac{1}{2})}, \quad \psi_k(t) \leftarrow \psi_k(t - \frac{1}{2}) - \eta \cdot \frac{\partial \hat{\mathcal{L}}_k}{\partial \psi_k^{(l)}(t-\frac{1}{2})}$

20:         Send $\theta_k(t), \psi_k(t)$ to the server

21:     **end for**

    — **Server Aggregation** —

    Server:

22:     $\theta(t) \leftarrow \sum_{k \in \mathcal{K}} \frac{N_k}{N} \theta_k(t), \psi(t) \leftarrow \sum_{k \in \mathcal{K}_t} \frac{N_k}{N} \psi_k(t)$

23:     $\{z_v(t)\}_{v \in \mathcal{V}} \leftarrow \bigcup_{k \in \mathcal{K}} \{z_v(t - \frac{1}{2})\}_{v \in \mathcal{V}_k}$

24: **end for**

    — **Final Output** —

25: **return** $\psi(T), \theta(T), \{z_v(T)\}_{v \in \mathcal{V}}$

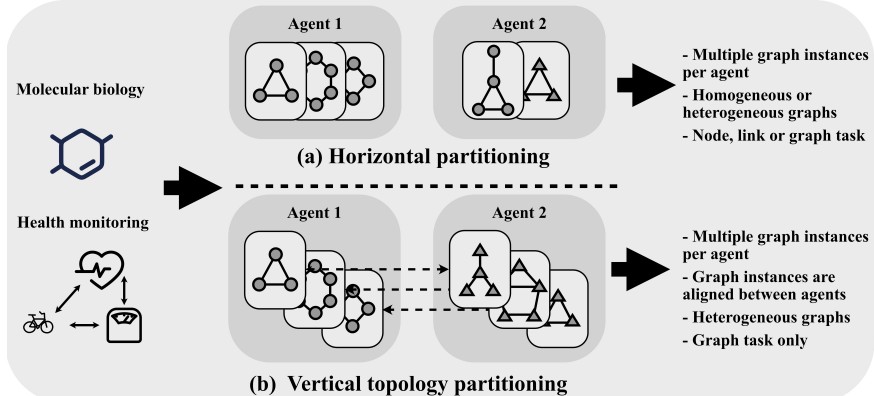

Figure 15: **Illustration of the different partitioning schemes for the case of multiple graph instances**. Multiple graph instances can be partitioned (a) horizontally, where agents hold unrelated graph samples, or (b) vertically, where graph instances associated with the same label are divided among agents.

That is, agents possess data about distinct graph samples indexed by $\mathcal{I}_k$, for example, laboratories observe distinct sets of protein graphs (Xie et al., 2021; He et al., 2022).

- *Vertical topology partition* For each sample of graph data $G^i$, the topology $\mathcal{G}^i$ is partitioned into $K$ disconnected components $\mathcal{G}_k^i$ held by different agents. The node features $X^i$ are split so that agent $k$ observes the features of the nodes $\mathcal{G}_k^i$. The label, usually a graph-label for these cases, is held by agent $k^*$, yielding datasets:

$$S_k = \{G_k^i\}_{i \in \mathcal{I}}; \quad S_{k^*} = \{G_k^i, y^i\}_{i \in \mathcal{I}}.$$

This partitioning is typical in multimodal scenarios where the graph $G_k^i$ represents one modality $k$ for the same sample $i$. These modalities are combined to solve a task related to the same sample $G^i$. An example is to diagnose patients based on the combination of different physiological observations, represented as graphs and held by different hospitals (Dong et al., 2021). This vertical topology partition is studied for centralized training (Qian et al., 2022), but has not yet been implemented in distributed training settings.

**Subgraphs of a global graph.** We now consider a setting where agents observe parts of a single, usually large-scale, global graph. In such scenarios, agents have only partial local views of the global graph, making learning over isolated graph data a central challenge.

In the most general case, the global, possibly heterogeneous graph, is partitioned across agents, such as each agent $k$ observes a local heterogeneous subgraph $\mathcal{G}_k = (\mathcal{V}_k, \mathcal{E}_k, \mathcal{O}_k, \mathcal{R}_k)$. That is, each agent observes a subset of nodes $\mathcal{V}_k \subseteq \mathcal{V}$ of a subset of types $\mathcal{O}_k \subseteq \mathcal{O}$, and links $\mathcal{E}_k \subseteq \mathcal{E}$ of subset of relation types $\mathcal{R}_k \subseteq \mathcal{R}$. These may induce a restricted type and relation mapping $\rho_k : \mathcal{V}_k \to \mathcal{O}_k$ and $\rho_k : \mathcal{E}_k \to \mathcal{R}_k$. For example, in a citation network, we might have object types $\mathcal{O} = \{\text{Author}, \text{Paper}, \text{Venue}\}$ and relation types $\mathcal{R} = \{\text{Writes}, \text{PublishedIn}, \text{Cites}\}$.

A graph is characterized by three main components: its topology, the types of nodes and relations along with their mappings, and the node features and potential labels. Accordingly, partitions can be defined along these dimensions, giving rise to the following taxonomy of graph partitions illustrated in Figures 16–17.

- *Topology partition.* The partition of the graph is done along the graph structure $\mathcal{H}_\mathcal{G} = (\mathcal{V}, \mathcal{E})$, without considering the node types, and is achieved by :
  - *Edge-cut:* Nodes are divided into (possibly overlapping) sets $\bigcup_{k \in \mathcal{K}} \mathcal{V}_k$, inducing local edge sets $\mathcal{E}_{\mathcal{V}_k \times \mathcal{V}_k} = \{(u, v) \in \mathcal{E} \mid (u, v) \in \mathcal{V}_k \times \mathcal{V}_k\}$. For example, consider a network of IoT devices related by a global graph. Telecom base stations monitor different geographical areas, resulting

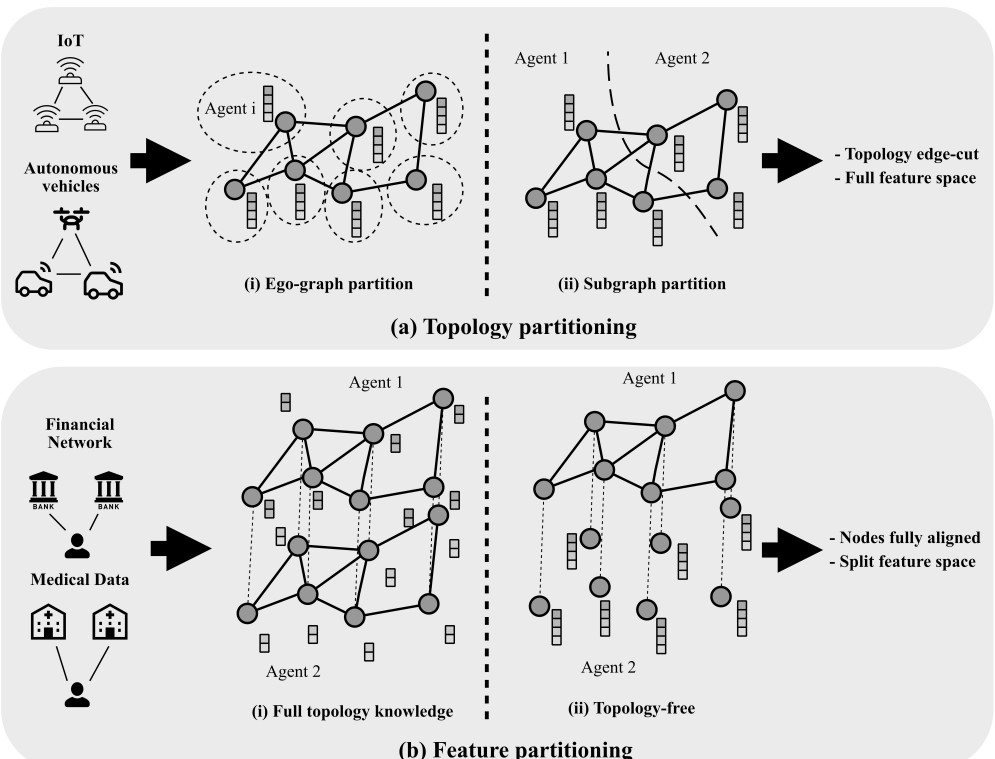

Figure 16: **Illustration of different subgraph partitioning schemes in a homogeneous graph.** In *topology partitioning* (a), the graph topology is divided across agents by edge cuts. Each agent observes the complete set of node features for the nodes assigned to it. Two cases can be distinguished based on the subgraph size: *ego-graph partition* (i), where an agent only observes a node together with its direct neighbors, thus forming an ego-graph; and *subgraph partition* (ii), where an agent holds a larger portion of the graph consisting of multiple, aligned nodes. In *feature partitioning* (b), agents share the same set of nodes but access different subsets of features. Two scenarios arise: in *full topology knowledge* (i), all agents know the complete set of graph links; whereas in the *topology-free* case (ii), only a subset of agents has access to link information.

in a different set of devices observed (Zeng et al., 2022). Usually, agents also know about their foreign nodes $\mathcal{V}_{f,k} = \{v; \exists u \in \mathcal{V}_k \mid (u,v) \in \mathcal{E}, v \in \mathcal{V} \setminus \mathcal{V}_k\}$ and the related edges cut $\mathcal{E}_{cut,k} := \mathcal{E}_{\mathcal{V}_k \times \mathcal{V}_{f,k}} = \{(u,v) \in \mathcal{E} \mid u \in \mathcal{V}_k \ v \in \mathcal{V}_{f,k}\}$ but not their features.

– *Vertex-cut:* Edges are divided into (possibly overlapping) sets $\bigcup_{k \in \mathcal{K}} \mathcal{E}_k$, inducing node sets $\mathcal{V}_k = \mathcal{V}_{|\mathcal{E}_k} = \{u \in \mathcal{V} \mid \exists v \in \mathcal{V}, (u,v) \in \mathcal{E}_k \text{ or } (v,u) \in \mathcal{E}_k\}$. For example, a device records the items browsed by a single user in a marketplace, forming an interaction graph private to the device (Wu et al., 2022; Han et al., 2024). We define the set of *vertex-cut*, vertices which are shared between agents: $\mathcal{V}_{cut,k} = \{v \in \mathcal{V}_k \mid \exists k' \neq k, \ v \in \mathcal{V}_{k'}\}$. The interactions represent the edges private to the agent, and some items represent the vertex cut between agents.

- *Feature partition.* Analogous to vertical partitioning for Euclidean data seen in Section 2.2.1, the feature space $\mathcal{X}$ is split among agents $\mathcal{X} = \oplus_{k \in \mathcal{K}} \mathcal{X}_k$, resulting in *vertical feature partitioning* of the features matrix $X$. Specifically, for each object type $o \in \mathcal{O}$, the feature matrices $X_o$ is partitioned across feature dimensions: for each node $v$ of object type $o$, $x_v = (x_{v,k})_{k \in \mathcal{K}}$, where $x_{v,k} \in \mathcal{X}_k$ is the part of vector $x_v$ observed by agent $k$.. Feature partition may result topology partition as well, which can be classified as a vertical topology partition, with a vertex cut.

- *Schema partition.* In a heterogeneous graph $\mathcal{G}$, partitioning can be defined along the network schema $\mathcal{T}_{\mathcal{G}} = (\mathcal{O}, \mathcal{R})$, by partitioning:

- *The object types:* Partitioning the set of object types $\mathcal{O}$ into disjoint subsets $\{\mathcal{O}_k\}_k$, i.e., $\mathcal{O} = \bigcup_k \mathcal{O}_k$, induces a corresponding partition of the node set in the original heterogeneous graph via the inverse mapping of the type function $\tau : \mathcal{V} \to \mathcal{O}$, namely $\mathcal{V}_k = \mathcal{O}_k^{-1} := \tau^{-1}(\mathcal{O}_k) = \{v \in \mathcal{V} \mid \tau(v) \in \mathcal{O}_k\}$. Object type based partition often leads to edge cuts.
- *The relation types:* Similarly, partitioning the set of relation types $\mathcal{R}$ into disjoint subsets $\{\mathcal{R}_k\}_k$, i.e., $\mathcal{R} = \bigcup_k \mathcal{R}_k$, induces a partition of the edge set via the inverse mapping of the relation function $\rho : \mathcal{E} \to \mathcal{R}$, yielding $\mathcal{E}_k = \mathcal{R}_k^{-1} := \rho^{-1}(\mathcal{R}_k) = \{e \in \mathcal{E} \mid \rho(e) \in \mathcal{R}_k\}$. Relation type based partition often leads to vertex cut.

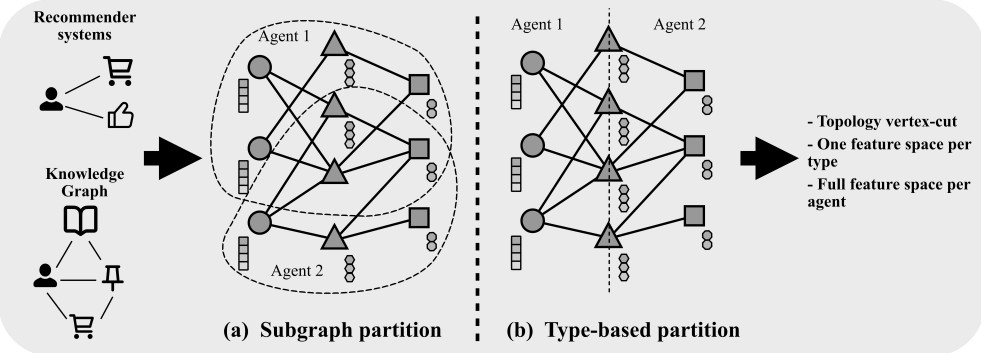

Figure 17: **Illustration of different subgraph partitioning schemes in a heterogeneous graph**. The subgraph can be partitioned at two levels: either by dividing the node set $\mathcal{V}$ among agents, or by partitioning the schema graph, partitioning the object types $\mathcal{O}$ across agents.

Graph partition of topology, schema, or features can be combined freely, leading to a high number of possible combinations of data distribution across the agents. In Table 2 we list the combinations that are most prominent in the literature.

- Considering *homogeneous graphs*, *topology partition*, resulting horizontal feature partition, is the most frequently studied case. Agents may hold a *subgraph* of the global graph due to geographic limitations, for example, in mobile networks, or due to collaborating system providers in IoT systems (Zhang et al., 2021). The other typical case of topology partition is when agents hold a single node and perform inference through the *ego-graph*. This is the typical case of interacting autonomous vehicles, robots, or drones (Pan et al., 2023). Traffic flow forecasting (Zeng et al., 2022), location-based user recommendation (Zeng et al., 2022), vehicle motion forecasting (Casas et al., 2020), or swarm navigation (Zhou et al., 2021; Solodova et al., 2025; Gao et al., 2021; Blumenkamp et al., 2022) are prominent tasks.

- *Vertical feature partition* of *homogeneous graphs* arises when agents hold partial information about the nodes, for example, spending habits of customers are known partially by e-commerce sites (Ni et al., 2021; Chen et al., 2024; He et al., 2024), and hospitals have partial records of patients (Fu et al., 2024). In most scenarios, *all agents hold a graph* (Ni et al., 2021; Chen et al., 2024; He et al., 2024), but alternatively, some agents may be topology-free, holding only Euclidean data (Mei et al., 2019; Cheung et al., 2021; Fu et al., 2024).

- Key examples of *heterogeneous graphs* are knowledge graphs (Chen et al., 2022; Wang et al., 2024) and recommender systems (Agrawal et al., 2024; Han et al., 2024; Liu et al., 2022; Qiu et al., 2022; Yan et al., 2024). Heterogeneous graphs can be distributed without considering the type of nodes, resulting in *topology partition*. Within that, *subgraph partition* emerges when companies observe the ego-graphs of a group of customers (Han et al., 2024). The requirement to preserve node privacy, on the other hand, may lead to *ego-graph* partition, where one agent knows about the ego graph of a single customer (Agrawal et al., 2024).

- *Type-based partition* in *heterogeneous graphs* is typical in knowledge graphs, for example, when one agent holds a patient-symptom database, and another one a symptom-disease database (Peng et al., 2021), with a common goal of training a patient-to-disease predictor, leading to agents holding *type-based subgraphs*. Privacy preservation under collaborating agents, for example, in health applications, can require a combination of *type-based* and *ego-graph* partition (Sun et al., 2020).

### 3.3.2 Statistical Heterogeneity in Graphs

Section 2.2.2 outlines the main forms of statistical imbalance in Euclidean data, *quantity skew* and *distributional shift*, which in turn include *covariate shift* and *concept shift*. In the Euclidean setting, data is typically drawn from a joint distribution $\mathcal{D}(x, y)$. In contrast, for graph data, the feature matrix $X$ is structured through an adjacency matrix, which together forms a graph sample $G = (A, X)$. The coupling between topology, features, and labels is captured by the joint distribution $\mathcal{D}(G, y)$, and brings new dimensions in the possible forms of statistical imbalance in the data, illustrated in Figure 18.

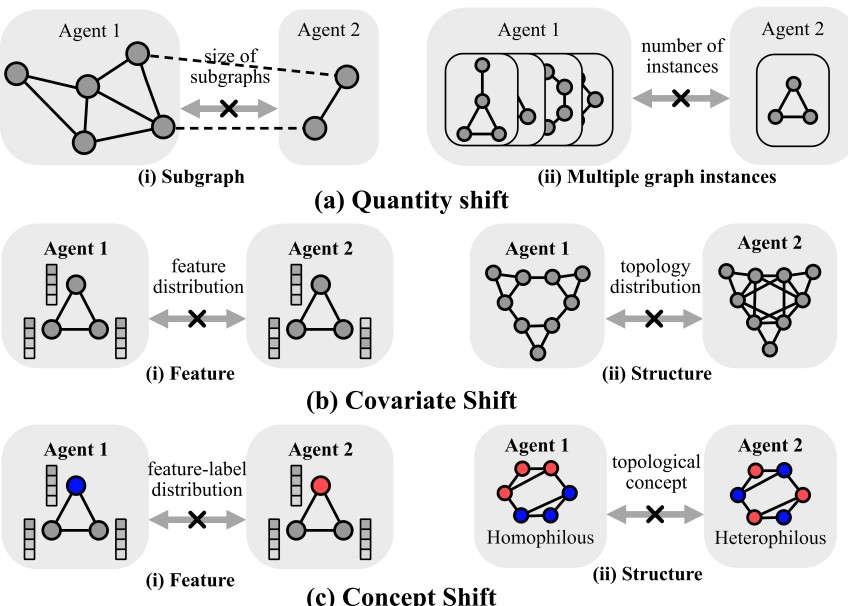

Figure 18: **Different types of statistical heterogeneity in partitioned graph data.** (a) *Quantity shift* refers to differences in dataset size across agents sampling from the same distribution, reflected in varying (i) subgraph sizes or (ii) numbers of graph instances. (b) *Covariate shift* refers to differences in input graph distributions, reflected in (i) node or edge features or (ii) graph structures. (c) *Concept shift* refers to differences in the mapping from input graphs to labels (colored on the figure), reflected in (i) *feature–concept* or (ii) *structure–concept* shifts.

*Quantity shift* can be observed in graph data when agents hold different amounts of graph information. In the subgraph case, this can be observed when the size of subgraphs, measured by the number of edges or nodes, differs across agents. In the full graph instances case, quantity shift can be observed when agents hold different numbers of graph samples.

*Distributional shifts* exhibit greater diversity for graph data, compared to the Euclidean case, due to the additional information of the graph structure $A$, as discussed in detail in (Zhang et al., 2024). Below, we characterize graph-specific distributional shifts using the global properties $(A, X, y)$. However, it is important to note that MPNNs operate on the collection $L$-hop ego graphs data $\{G^v\}_{v \in \mathcal{V}_k}$ centered at each node rather than the full graph data $G = (A, X)$, and thus experience these shifts in practice through changes in local neighborhood distributions $\mathcal{D}(G^v)$.

**Covariate shift.** Covariate shift refers to differences in the input graph distribution $\mathcal{D}(G)$ across agents, while the conditional label distribution $\mathcal{D}(y \mid G)$ remains unchanged. Alternatively, one can view it from the label perspective as discussed in Section 2.2.2, where $\mathcal{D}(y)$ and $\mathcal{D}(G \mid y)$ differ across agents. We adopt the input perspective, which is more common in the literature, and introduce graph-specific covariate shifts, *feature shift* and *structure shift* (Zhang et al., 2024).

- *Feature shift* for graph data is analogous to *feature shift* for Euclidean data seen in Section 2.2.1. It occurs when, given a graph topology $A$, the node features distribution $\mathcal{D}(X \mid A)$ differs (Zhang et al., 2024). For example, in superpixel graphs of natural scenes (Monti et al., 2017), one agent may use grayscale images while another uses color images from the same scene. In email networks (Bojchevski & Günnemann, 2018), changes in the language used across organizations can lead to different node features.

- *Structure shift* is specific to graph data. Real-world graphs often exhibit structural diversity across regions and nodes. In collaborative scnearios it means that the topology distribution $\mathcal{D}(A)$ may differ significantly across agents (Liu et al., 2023). The factors influencing structural shifts are many, often driven by hidden latent variables of the environment (Wu et al., 2018; Gui et al., 2022) or differences in data domains (Tan et al., 2024). These shifts can be present from the start or emerge when new agents or nodes with distinct structures or labels arrive. While structural shifts can be characterized through the spectrum of $A$ (Tan et al., 2024), three large categories are usually considered: *Size shifts* occur when graphs or subgraphs differ in the number of vertices, e.g., sentence graphs vary with text length, and it is the most discussed in the literature (Yehudai et al., 2021). *Density shifts* are present when the edge density, measured by different metrics (node degree, clustering coefficient), differs. This is a structure shift that could prove to be the most critical to GNN generalization capabilities (Bazhenov et al., 2023). *Motif shifts* occur when the base of the graph is common, but only a portion of the graph differs in terms of typical connection patterns (Wu et al., 2022).

**Concept shift.** Concept shift arises when the conditional distribution $\mathcal{D}(y \mid G)$ differs between agents.

- *Feature concept shift* in graph data is analog to *concept shift* seen in Section 2.2.2. It refers to changes in the concept distribution $\mathcal{D}(y \mid X)$, assuming the topology $A$ is fixed. For example, in the case of full graph instances, some superpixel graphs might have a tight correlation between color and class (e.g., blue for water), while others do not (Gui et al., 2022). In the subgraph case, an email network, agents holding emails from respectively older and younger users, may observe different vocabulary-class relations. These shifts are particularly challenging for GNNs since they assume stable feature-label correlations (Duong et al., 2019; Faber et al., 2021).

- *Structure concept shift* for graph refers to differences in $\mathcal{D}(y \mid A)$ (or $\mathcal{D}(A \mid y)$)(Liu et al., 2023). This structural concept has been historically modeled using using the *Stochastic Block Model* (Holland et al., 1983; Karrer & Newman, 2011; Abbe, 2018) that assumes $\mathcal{D}_u(A_{uv} \mid \{y_v, y_u\})$ uniform across the graph. More commonly, structure concept shift is connected to the well-studied principle of graph homophily (Zhu et al., 2020; Platonov et al., 2023;; Fuchsgruber et al., 2025), which states that nodes tend to connect to nodes of the same labels. Constant homophily level is critical in centralized training, as GNN tends to perform gradually worse on heterophilous graphs, referred to as *homophily bias* (Tan et al., 2025a). In the case of distributed graphs, differences in the graphs homophily level between agents, referred to as *homophily conflict* (Tan et al., 2025a), lead to decreased learning accuracy.

While the distinctions between feature shift, structure shift, covariate shift, and concept shift help understanding the challenges of learning, this categorization is limiting for *evolving* graphs, and requires further research. Since MPNNs operate on overlapping $L$-hop ego graphs, topology $A$ and features $X$ are inherently coupled: when nodes join or leave a network, both $\mathcal{D}(A)$ and $\mathcal{D}(X)$ change concurrently across multiple overlapping neighborhoods (Kim et al., 2025). In the subgraph case, when partition boundaries evolve (e.g., nodes reassigned across agents), agents observe different local distributions despite a unified global graph.

Moreover, these shifts have heterogeneous impacts across ego graphs, since not all nodes affect the distribution equally when joining or leaving the network. For example, highly connected nodes, nodes with rare labels, and nodes violating local homophily levels have greater effects.

### 3.3.3   System Heterogeneity in Collaborative Learning over Graph Data

Collaborative learning over graphs must address *device*, *network*, and *task* heterogeneity, as in the Euclidean case discussed in Section 2.2.3. However, for the inference process, both under and after learning, graph-structured data introduces new challenges that stem from the interdependence between the system constraints and the graph topology itself. We discuss these challenges below.

**Device and network constraints under collaborative inference.**   Unlike Euclidean federated learning, where inference is local (Algorithm 1), subgraph inference is inherently *collaborative* and requires agents to exchange node embeddings across subgraph boundaries during message passing (lines 11–12 of Algorithm 3). While collaborative inference also appears in vertical federated learning (Algorithm 2), graph-structured data differs in that the graph topology itself determines inference dependencies between nodes and communication patterns between agents (Figure 14), resulting in tightly coupled system constraints. Building on the device and network heterogeneities identified in Section 2.2.3, collaborative inference over graphs does not introduce entirely new aspects. Instead, the graph topology modulates and amplifies existing computational and communication heterogeneities:

- *Computational heterogeneity* is amplified by the structure of local subgraphs. Message passing cost scales with topology: *computation* grows with the number of edges in $\mathcal{G}_k$ times the number of layers $L$, while *memory* must store boundary embeddings $\{h_v^{(l)}\}_{v\in\mathcal{V}_{f,k},\, l=1,\ldots,L}$ (line 12). Consequently, dense subgraphs or highly connected agents induce higher computational and memory loads, exacerbating eventual device heterogeneity.

- *Communication heterogeneity* is further amplified by the topology of inter-agent boundaries. Incoming and outgoing communication volumes scale with the number of foreign and private boundary nodes, $\mathcal{V}_{f,k}$ and $\mathcal{V}_{p,k}$, which typically correlate with the centrality of boundary nodes. These unbalanced loads further compound the communication heterogeneities identified in Section 2.2.3, where unreliable links can delay or corrupt embedding exchanges. When subgraph boundaries span many agents, synchronizing boundary node embeddings incurs additional communication overhead and delays.

The joint effects of communication and computing heterogeneities induce a *partitioning tradeoff*. Minimizing edge cuts reduces cross-agent communication but may distribute load unevenly, amplifying device heterogeneity, whereas distributing computation more evenly increases cross-agent edges and consequently communication overhead. This trade-off becomes harder to manage as the number of agents grows and additional system constraints are introduced. The partitioning dilemma results in system-level bottlenecks. Stragglers caused by uneven computation or communication delays slow both parameter updates and node representation propagation (line 17). Late embeddings prevent message passing from completing, directly degrading inference speed and quality.

These challenges affect the management of dynamic graphs. When new nodes join, the inference topology changes, compounding existing system heterogeneity with the shifting local neighborhood distributions discussed in Section 3.3.2.

**Task heterogeneity.**   Task heterogeneity requires the learning and inference processes to handle multiple distinct tasks within the same GNN framework (Mai et al., 2024; Meng et al., 2021; Zhang et al., 2023). This heterogeneity can manifest across prediction levels (node, edge, or graph-level tasks) or across different application domains (e.g., citation networks and social networks) (Wang et al., 2025). The modular structure of graph representation learning, and MPNNs in particular, naturally supports this heterogeneity through separate encoding, message passing, and decoding components, as described in Section 3.1. This modularity allows agents to share common encoders while maintaining task-specific prediction heads.

### 3.3.4 Privacy Vulnerabilities in Collaborative Learning over Graphs

Collaborative learning over graphs inherits the privacy vulnerabilities identified in Section 2.2.4 for model aggregation. The fully trained parameters, local updates, and the evolution of aggregated models remain vulnerable to model and gradient inversion. Additionally, it inherits the privacy vulnerabilities of *collaborative inference* seen for Euclidean data in Section 2.3.3. Collaborative inference also presents additional privacy challenges, considering the information held by the topology itself, as discussed below.

**Privacy-preserving collaborative inference.** Unlike standard collaborative learning where inference is local (Algorithm 1), collaborative inference over graphs requires agents to exchange node embeddings across subgraph boundaries during message passing (lines 11–12 of Algorithm 3). These embedding exchanges create an attack surface as observing boundary node representations enables inference about the local subgraph structure and feature distributions held by neighboring agents. If intermediary embeddings $h_v^{(l)}$ and features $x_v$ during distributed message passing as in (18), (19), (20)) are not communicated, the final embeddings $z_v = h_v^{(L)}$ shared across agents can still leak sensitive information (Duddu et al., 2020). Adversaries can perform *inference attacks* that can extract private node features $x_v$ for nodes held by other agents (Duddu et al., 2020), reconstruct edge existence $(u, v) \in \mathcal{E}_j$ in the subgraphs of other agents $\mathcal{G}_j$ (Wu et al., 2022; He et al., 2021), or determine whether specific nodes (Olatunji et al., 2021; Zhang et al., 2021) or edges (He et al., 2021) participated in the training.

The vulnerabilities discussed above assume *honest-but-curious* agents that passively infer information from shared embeddings. Beyond passive observation, malicious agents could *actively* manipulate their local subgraph structure or features to attack inference outcomes (Zügner et al., 2020; Sun et al., 2022) or poison the training process (Dai et al., 2018; Sun et al., 2022). These malicious threat scenarios fall outside the scope of this study.

## 3.4 Improving Inference and Learning Effectiveness

As per Euclidean data, methods for improving the effectiveness of learning have been proposed for graph data. Table 2 shows that the proposed solutions spread over data-based and model-based techniques, as well as methods for vertically aligned data. Several solutions address learning for multiple graph instances, which is closely related to traditional FL.

Methods to increase learning effectiveness aim at two objectives: designing a tighter convergence to a global solution $\psi^*, \theta^*$ of the collaborative training objective (29), or tighter individual convergence to the optimal solution of the agent, $\psi^*, \theta^*$, according to the local objectives (25) (28). Effectiveness is challenged by the generalization gap (seen in Section 3.1), data isolation arising from the various graph data partitions (seen in Section 3.3.1), or by statistical heterogeneity (seen in Section 3.3.2). Subsequently, we classify and describe the main strategies that address effectiveness, following the classification seen for Euclidean data in Section 2.3, that is, **data-based**, **model-based** techniques, and **training and inference under aligned partitions**.

### 3.4.1 Data-based Techniques

Data-based techniques manipulate the training data to improve convergence toward the optimal parameters $\psi^*, \theta^*$ of a jointly trained GNN. For Euclidean data, these techniques typically operate on independent samples through data augmentation, controlled sampling, or local regularization to mitigate statistical heterogeneity (Section 2.3.1). However, graph data is more complex than sets of independent samples. First, topology couples nodes, features, and labels, creating high structural diversity: nodes vary by centrality and connectivity, subgraphs differ in density and spectral properties, exhibiting high statistical heterogeneity (Section 3.3.2), making stable learning challenging. Second, in subgraph partitions, receptive fields extend beyond local boundaries, creating severe data isolation as neighborhoods are cut across agent boundaries (Section 3.3.1). Furthermore, depending on the data partition setting and the learning tasks, graph-specific data-based techniques manipulate entities of different granularity, such as node features and labels, topology, or entire graph instances.

**Data augmentation.** The datasets $S_k$ held locally by agents can be augmented to address three challenges: addressing the generalization gap, statistical heterogeneity, and data isolation. We distinguish methods that augment a graph instance, or augment the dataset with new graph instances.

Methods that **augment a graph instance** may manipulate the graph topology:

- *L-hop recovering* techniques are applied both for edge and vertex cut, and exchange raw node and edge information between agents, so that they complete the ego-graphs of the nodes. This simplifies the collaborative inference phase of Algorithm 3; however requires the use of cryptographic primitives to preserve privacy (Qiu et al., 2022; Agrawal et al., 2024). Consequently, the $L$-hop expansion can prove to be expensive and can therefore be restricted to the 1-hop neighborhood (Agrawal et al., 2024).

- *Node rewiring* modifies the existing edge set $\mathcal{E}_k$ to yield a graph (or subgraph) with a connectivity that improves generalization. This is achieved by a collaboratively trained transformer encoder-decoder to modify the spectrum, and thus the connectivity, of each local graph instance (Tan et al., 2024), introducing new frequencies that counter covariate structure shift.

- *Node addition* techniques aim to extend the local graph by adding new virtual nodes and their associated edges using local graph data only. This can be done using generative models trained on the server-side and then communicated to agents to extend local graphs (Peng et al., 2022). Rewiring strategies using pairwise feature similarity have also been explored, allowing agents to infer plausible edges and labels. These are then integrated into a global graph on the server and redistributed to other agents (Chen et al., 2024).

Other ways of manipulating the graph instances are the augmentation of features and labels. These techniques are also specific to graph data, since their objective is to decrease statistical heterogeneity among neighbors or in ego-graphs:

- *Feature augmentation* techniques for graph data manipulate the feature matrix $X$. Averaging features of neighbors and synthetic feature generation address homophily bias and feature imbalance (Pei et al., 2023; Lin et al., 2022). Classical methods developed for Euclidean data are used to inject controlled noise (Gao et al., 2024).

- *Label augmentation* has been proposed for various learning tasks. For link prediction, one method uses a local link prediction model to identify plausible links, which are treated as virtual edges. These virtual edges can then be shared across agents to enhance their subgraphs in vertex-cut partitions (Liu et al., 2022). For node classification, label smoothing has been applied to mitigate the effects of homophily bias. In this setting, a single round of distributed averaging (as introduced in Section 2.4.1) is used to propagate label information across agents (Mai et al., 2024). An alternative approach uses contrastive learning, where negative node embeddings, chosen to be close to true node embeddings, are used to sharpen representation learning. These embeddings can be sampled from other agents, such as randomly selected agents in ego-graph datasets when the number of agents is large enough (Giaretta & Girdzijauskas, 2023).

Other methods **augment the dataset with new graph instances**. A graph generative model, trained on the server side, can be used to synthesize new graph instances on the agent side based on local graph data (Guo et al., 2023). The existing graph instances and their node labels can be combined (Wang et al., 2021) to generate new plausible graph and label instances. These techniques can be applied to full graph instances or to the collection of ego graphs and node label $(G_k^v, y_v)$ in the subgraph case (Zhang et al., 2023). Instead of generating new plausible graphs, one can focus on generating non-plausible graphs. This contrastive data augmentation is a frequent approach to provide good generalization capabilities to GNN (Veličković et al., 2019; You et al., 2020). Perturbations include node dropping, edge perturbation, attribute masking, random reindexing of the features, or subgraph sampling (Veličković et al., 2019; You et al., 2020). In collaborative settings, these techniques can be applied to the local graph data for better generalization as in (Chen et al., 2021; 2022; Mai et al., 2024).

**Local regularization.**    Local regularization techniques reduce discrepancies between agents in their model parameters or vertex-cut embeddings. For graph data, these methods exploit the topology by using graph perturbations for contrastive learning, regularizing based on structural properties (centrality, connectivity), or aligning shared vertex representations across partitions.

- *Regularizing local models* is ubiquitous for FL as described in Section 2.3.1. For graph data, it can be achieved by modifying the local loss (line 18) of the agents with an additional regularization term. This additional term can make use of the augmented data techniques described above, when integrated in a contrastive term against perturbed graphs (Mai et al., 2024), or a loss on trimmed subgraphs (Ceyani et al., 2025).

- *Prototype-based regularization* is challenging since nodes can exhibit high structural variety yet must be mapped to a limited set of prototypes. Solutions, therefore, leverage graph-specific metrics to construct structurally relevant prototypes. Prototypes can be stratified according to graph properties such as node centrality (Tan et al., 2025b; Kim et al., 2025) or label influence (Tan et al., 2025b) within subgraphs, thereby mitigating noise from low-connectivity nodes while amplifying information from rare labels. The locally constructed prototypes are then aligned with global prototypes through contrastive losses (Kim et al., 2025; Tan et al., 2025b). For full-graph instances, prototype variance can be reduced via accumulation across training iterations (Tan et al., 2024).

- *Regularizing the vertex cut representations* is specific to the subgraph case with vertex cut, and so far discussed for KG tasks only (Chen et al., 2022). Regularization terms can be added to the local loss (line 18) to decrease the variance between vertex cut representations, respectively on the agent-side (line 15) and server-side (line 23), or to decrease the similarity of the consecutive updates.

**Subsampling strategies.**    In contrast to data augmentation, which uses more data than available locally, subsampling techniques use less. This subsampling operates at multiple levels: agents are sampled in training rounds (line 2 of Algorithm 1 and line 3 of Algorithm 3) while nodes are sampled during local inference (lines 6 and 13 of Algorithm 3). For graph data, subsampling serves dual purposes: it can improve learning effectiveness through regularization or improve efficiency by reducing message-passing costs. The techniques include:

- *Agent selection* for graph data is inspired by agent selection techniques in traditional FL, seen in Section 2.3.1. It can be used to mitigate the *homophily conflict* through server-side weighted aggregation of the models, where the weight assigned to each agent depends on the homophily level of its local dataset. Learning effectiveness is increased if agents with high-homophily graphs receive higher aggregation weights (Tan et al., 2025a). As a form of agent selection for vertex-cut subgraphs, (Peng et al., 2021) suggests averaging $z_v(t)$ only if it leads to lower local loss.

- *Graph subsampling* has the potential to improve the model accuracy. For effective sampling in large graphs, (Ceyani et al., 2025) suggests the use of advanced generative models (Bengio et al., 2023). Training on graphs that are perturbed using subsampling techniques like random node dropping or subgraph sampling is a classical technique that allows GNN to better generalize (You et al., 2020). For collaborative learning, the same technique is applied to the local datasets in (Chen et al., 2021; 2022). Node and model parameter sampling is proposed in (Shi et al., 2025) to decrease the computational and communication burden under training. In the subgraph collaborative inference case, node subsampling can be applied to the local expanded graph $\tilde{G}_k$. For instance, (Liu et al., 2021) proposes to select foreign vertex boundary nodes only if their contribution improves the local loss (26). Subsampling strategies are also designed to balance efficiency and effectiveness, using greedy heuristics (Pan et al., 2023), reinforcement learning (Chen et al., 2021), or Markov chain methods (Pan et al., 2023).

**Open challenges of data-based techniques.**    In collaborative settings, graph-specific data properties create fundamental challenges for learning and inference effectiveness that extend beyond the Euclidean case. Some challenges stem from open problems existing also for centralized graph machine learning, while others

emerge specifically from the interaction between graph structure and collaborative dynamics during both training and inference. Below we identify key research directions where graph-specific data properties create unique obstacles for collaborative settings:

- *Towards collaborative graph rewiring.* In centralized settings, recent research has explored decoupling the observed input graph from the graph used for message passing (Wang et al., 2019; Fatemi et al., 2021; Kazi et al., 2022; Topping et al., 2022; Gutteridge et al., 2023). These *graph rewiring* approaches augment the graph structure to use neighbor information more effectively, improving model accuracy. Extending such rewiring to collaborative settings presents unique challenges: when a global graph is partitioned across agents, rewiring decisions have to take into account conflicting constraints. For instance, adding edges between nodes held by different agents increases inter-agent communication and thus adds communication costs and privacy vulnerabilities.

- *Towards capturing statistical heterogeneities.* Collaborative GNN learning is challenged by structural diversity inherent to graph domains. Unlike images or text, graphs span vastly different domains, from social networks to molecular structures, that create challenging statistical heterogeneity across and within agents datasets. At the same time, graph properties are typically characterized by classical network analytics metrics (e.g., centrality, degree), which do not integrate the computational scheme of MPNN (Figure 12), nor the feature information. Promising directions for more expressive characterization include adapting GNN-specific metrics from centralized training theory to collaborative settings. For example, moving tree distances (Chuang & Jegelka, 2022) have been tied to GNN generalization capabilities both empirically and theoretically (Maskey et al., 2025; Southern et al., 2025). This improved characterization could enable more informed data-sharing strategies, such as identifying which agents should collaborate based on structural similarity rather than simple topological statistics.

- *Towards graph-specific model optimization.* While global model optimization has been extensively studied for collaborative learning on Euclidean data (Section 2.3.1), ensuring convergence and generalization for GNNs remains poorly understood even in centralized settings (Jegelka, 2022; Morris et al., 2024). Collaborative scenarios exacerbate this challenge: agents with topologically distinct subgraphs (scale-free vs. regular networks, dense vs. sparse regions) may experience fundamentally different optimization landscapes. For instance, MPNNs exhibit representation collapse along high commute-time paths (Di Giovanni et al., 2023), causing sensitivity to vary with network structure, a problem amplified when agents cannot share data to compensate for these structural differences. Developing convergence guarantees and regularization strategies that account for topology-dependent optimization dynamics in collaborative settings is a promising research direction.

- *Towards collaborative learning on evolving graphs.* Most collaborative GNN work assumes static graphs, where topology and features remain fixed during training. However, real-world graphs evolve temporally: nodes and edges appear or disappear (e.g., users joining social networks, devices entering IoT systems), causing topology $\mathcal{D}(A)$ and features $\mathcal{D}(X)$ to shift concurrently (Section 3.3.2). This coupled evolution creates unique challenges for collaborative settings: when a node's neighborhood changes, its ego graph distribution $\mathcal{D}(G^v)$ shifts, but agents holding overlapping ego graphs may observe these changes asynchronously, creating temporal misalignment during collaborative training. Therefore, it is an essential open challenge to establish characterization methods for temporal evolution and benchmarks for dynamic collaborative graph learning (Kim et al., 2025).

> **Data-based techniques for graph-structured data** adapt the techniques seen for Euclidean data (local regularization, data augmentation, and subsampling) to address statistical heterogeneity and data isolation in partitioned graphs. However, in contrast to Euclidean data, many of the methods manipulate the graph itself, extending, regulizing or subsampling nodes, links, subgraphs, vertex-cuts or the agents themselves. Open challenges include the statistical characterization of static or dynamic subgraphs, as well the extension of emerging centralized models to collaborative settings.
>
> **Trilemma.** Efficiency trade-offs emerge as subsampling reduces message-passing costs but may discard informative boundary nodes, while topology augmentation increases communication overhead through cross-agent coordination. Privacy concerns arise as methods requiring structural coordination (L-hop recovery, label smoothing, prototype sharing) expose topology and embeddings.

### 3.4.2 Model-based Techniques

Model-based solutions with *parameter personalization* and *architecture personalization* are proposed to find agent-optimal GNN models despite data and system heterogeneity. Most model-based solutions seen for Euclidean data in Section 2.3.2 have been adapted for graph-structured data as seen in Table 2. However, graph-structured data and GNNs introduce distinct considerations for model personalization. Techniques for personalizing model parameters can address statistical heterogeneity on different levels of granularity (nodes, edges, subgraphs, entire graphs). Model personalization can utilize the layered message-passing architecture of GNNs to decouple model components.

**Parameter personalization.** Parameter personalization techniques seen in Section 2.3.2 have been adapted for graph data, using graph-specific quantities to personalize model parameters:

- *Local adaptation* strategies can be extended to GNNs. In (Zhang et al., 2023), the local model is formulated as a learnable interpolation between the global and the local GNN parameters, while (Baek et al., 2023; Fang et al., 2025; Han et al., 2024) suggest masking the weights of the global models using learnable parameters. A meta-learning framework is suggested for node-level tasks on graph data in (Wang et al., 2022; Han et al., 2024), where the support and query sets are defined as subgraph partitions.

- *Hypernetwork-based* techniques (Shamsian et al., 2021) are extended for graph data in (Liang et al., 2024), that defines the representation vector of agent $k$ as the mean of node representations within the local subgraph: $\frac{1}{N_k} \sum_{v \in \mathcal{V}_k} z_v(t)$.

- *Cluster-based* techniques can be used to cluster agents based on gradient similarities (Xie et al., 2021). If the agents observe a single ego-graph only, then they can be clustered based on their node embeddings. As suggested in (Qu et al., 2023), an effective way to support learning within clusters is to construct virtual local graphs.

- *Similarities-based* techniques are also adapted specifically to graph data. However, defining similarities between graph distributions is challenging. The similarity matrix $\Omega = [\omega_{kl}]_{k,l \in \mathcal{K}^2}$ can be interpreted as defining a latent *inter-agent graph*, distinct from the local graph data of each agent. Similarity scores, inspired by (Zec et al., 2022), are evaluated for different GNN models on randomly generated subgraphs in (Baek et al., 2023). (Liang et al., 2024) use subgraph representations $c_k$ to compute similarities between agents, which are then diffused over the inter-agent graph using learnable *sheaf diffusion* (Bodnar et al., 2022) algorithms trained on the server.

**Architecture personalization.** Architecture personalization modifies the model architecture used by each agent. While standard techniques from Section 2.3.2 remain applicable, GNNs compositional message-passing architecture is particularly well-suited for flexible architecture personalization:

- *Model decoupling.* As described in Section 3.1, a GNN $F_{\psi,\theta}$ can be decomposed into an encoder $\phi_\theta$ decomposed into sublayers $f_{Enc}, f_{Mes}^{(l)}, f_{Agg}^{(l)}, f_{Up}^{(l)}$ for l=1,...,L, and a predictor $f_\psi$. This layered architecture allows for splitting the model *layer-wise* into different components. The split means that one component is shared while another is personalized to the agents (Mai et al., 2024). More precisely, the shared component can either be trained on the agent side and is then aggregated periodically on the server, or it is trained on the server (Meng et al., 2021; Zhang et al., 2023) using alternative optimization techniques (Meng et al., 2021). Decoupling can occur between the encoder $\phi_\theta$ and predictor $f_\psi$ (Mai et al., 2024; Tan et al., 2024), or between the feature encoder $f_{Enc}$ and subsequent layers, where either the early layers are personalized, and later ones shared (Meng et al., 2021), or vice versa (Zhang et al., 2023).

  Local GNNs have also been pruned *width-wise* in (Shi et al., 2025), where each agent removes local learnable weights based on a layer-specific, learnable threshold. To mitigate severe performance degradation during aggressive pruning, a gradual rollback strategy is employed. Aggregating width-wise pruned models remains challenging and has been proposed to be performed at lower bit resolution to promote generalization across agents.

- *Multimodal personalization.* The server can maintain $M$ specialized GNNs parameters $\{\theta_m^c\}_{m=1}^M$ for each graph or subgraph type. The framework for multimodal personalization (Chen & Zhang, 2022; Smith et al., 2017) seen in Section 2.3.2 has been applied to full graph instances associated when one agent that can collect graphs of different modalities or tasks (He et al., 2022). In vertically partitioned subgraph settings, predictions from agents local models for the same nodes can be combined using a confidence-based aggregation scheme known as *knowledge voting* (Mai et al., 2024).

- *Knowledge distillation.* Instead of sharing knowledge through model parameters, agents can exchange node representations while remaining free to choose their local architectures. This approach leverages a key property of GNNs: they maintain a shared representation space $\mathbf{R}^{d_z}$ across layers, enabling meaningful comparison of node embeddings between agents. To achieve knowledge sharing, agents add an alignment term to their local loss function. This term measures the discrepancy between local and global representations, either for vertex-cut nodes (Fu et al., 2024) or for node prototypes (Tan et al., 2024). In vertical topology-free partitions, this embedding alignment enables a useful form of knowledge transfer: topology-aware agents can distill their structural knowledge to topology-agnostic agents (Fu et al., 2024). The topology-agnostic feature encoder $f_{Enc}$ thereby learns to integrate topological information indirectly, avoiding communication and synchronization overhead during inference.

- *Decoupling the message passing operations.* An architecture personalization approach specific to GNNs is the personalization of the message passing operation itself. In (Lei et al., 2023), the message computations in (18) are adapted based on edge type. Messages along edges cut $\mathcal{E}_{\text{cut}}$ carry only partial information, for instance, when connected to stale embeddings. To compensate for this limited information, these cross-boundary messages are processed using more expressive message and aggregation functions, $f_{\text{Mes}}$ and $f_{\text{Agg}}$, designed to better capture missing dependencies. In contrast, messages over local edges (i.e., those within $G_k$) rely on simpler functions (Wu et al., 2019), as they benefit from more complete and up-to-date local information.

**Open challenges of model-based techniques.** While collaborative GNN learning has adapted many techniques from Euclidean federated learning, recent advances in graph machine learning, from expressive architectures beyond standard MPNNs to foundation models for graphs, present new opportunities and challenges for collaborative settings. Key research directions include:

- *Towards more advanced topological structures.* Work in collaborative settings has primarily studied standard GNN architectures (e.g, GCN (Kipf & Welling, 2017), GAT (Veličković et al., 2018)). However, more expressive extensions of the MPNN framework have been introduced under the umbrella of *topological deep learning*(Hajij et al., 2022; Papamarkou et al., 2024) or *geometric deep learning* (Bronstein et al., 2021). These architectures introduce new collaborative challenges, since

partitioning higher-order structures across agents creates more complex boundary interactions than standard vertex cuts. Therefore, understanding how these structures integrate into the current collaborative learning solutions and physical network architectures is an exciting open research direction.

- *Towards effective layer pruning.* Apart from (Wu et al., 2019; Lei et al., 2023), most work assumes all GNN layers are equally important for collaborative learning. However, studies have revealed this is generally not true in deep architectures, with GNNs potentially benefiting from aggressive layer pruning strategies (Kummer et al., 2025). While this concept has been exploited to alleviate communication costs in CNN-based collaborative settings (Park & Joe-Wong, 2024), its application to GNNs remains unexplored. In partitioned graphs, layer importance may depend explicitly on graph topology and partitioning characteristics. Investigating which layers are essential for collaborative learning, building on (Lei et al., 2023)'s finding that boundary-processing layers require special treatment, could enable more efficient communication strategies.

- *Towards the utilization of foundation models.* Most works focus on training task-specific models from scratch, or maintain separate models for different domains. However, this use of personalized per-domain or per-task architectures might be challenged by the paradigm shift of *foundation models (FMs)* currently revolutionizing ML. These generalist pre-trained models, often with billions of parameters, can be effectively adapted to downstream tasks with a relatively small amount of new data, but the development of FMs on graphs remains an open problem (Wang et al., 2025). Their integration into collaborative settings, where billion-parameter models face computational, storage, communication, and privacy constraints, is a research avenue.

> **Model-based techniques for graph-structured data** adapt parameter personalization and architecture personalization strategies from Euclidean collaborative learning to address data and system heterogeneity. Parameter personalization, however, needs to consider statistical properties observed by an agent, which requires new solutions. GNNs layered message-passing architecture enables fine-grained model decoupling across layers, edge types, and message-passing operations, providing richer personalization opportunities than fully-connected networks. Challenging research questions include the adaptation of advanced topological structures, optimized layer pruning, and the general question of the adaptability of foundation models.
>
> **Trilemma.** Architecture decoupling improves effectiveness by allowing agents with heterogeneous model architectures to collaborate, but increases communication overhead through coordination of multiple model components. Knowledge distillation and embedding alignment methods expose intermediate node representations and topology information, necessitating privacy-protection mechanisms.

### 3.4.3 Training and Inference under Aligned Partitioning

The vertical and vertex-cut partitioning cases introduced in Section 3.3.1 create a distinct challenge, since multiple agents hold complementary views of the same graph entities (nodes or entire graphs). Unlike horizontal partitions, where each agent observes complete but disjoint samples, these partitions split information about the same entities across agents, similarly to vertical federated learning for Euclidean data (Section 2.3.3). The presence of aligned entities across agents induces the need for aggregation mechanisms that go beyond model parameter aggregation (Algorithm 1) or cross-boundary message passing (Algorithm 3), borrowing instead from aggregating cross-agent views (Algorithm 2).

Specifically, when multiple agents generate representations for the same node (vertex-cut partitions of homogeneous graphs in Figure 16 and heterogeneous graphs in Figure 17) or for the same graph instance (vertical partitions of multiple graph instances in Figure 15), these representations must be combined to support effective prediction. This section, therefore, organizes existing methods into three families:

techniques for aggregating node representations across agents, transfer learning strategies for cross-agent adaptation without inference-time alignment, and methods for combining graph-level representations.

**Aggregating node representations.** In the case of vertex-cut partitions, node representations for the same node are generated at different agents, and therefore need to be combined. Indeed, the server receives (line 17) a collection of node representation $\{z_{v,k}(t - \frac{1}{2})\}_{v \in \mathcal{V}_k}$ from each agent $k \in \mathcal{K}$. So for a node cut $v \in \mathcal{V}_{cut}$, the different representations $\{z_{v,k}(t - \frac{1}{2})\}_{k \in \mathcal{K}}$ have to be aggregated (line 23). Different choices exist for this aggregation step, which aim to increase the quality of the final node representation $\{z_v(T)\}_{v \in \mathcal{V}}$:

- *Averaging node representations.* The standard approach is to compute $\bar{z}_v(t)$, the average on $\mathcal{K}$ of the representations $\{z_{v,k}(t - \frac{1}{2}), k \in \mathcal{K}\}$ from the different agents to obtain more generalizable representations (Chen et al., 2021; 2022; Wu et al., 2022; 2021; Zhang et al., 2022; Qu et al., 2023; Han et al., 2024; Agrawal et al., 2024; Ni et al., 2021). The averaging can be restricted to agents with low local loss (Mai et al., 2024). Alternatively, the embeddings $z_v(t)$ can be updated by taking a small step toward $\bar{z}_v(t)$ using a gradient-based formulation, which provides more stable updates throughout the training (Wu et al., 2021; 2022; Zheng et al., 2023).

- *Personalizing node representations.* Personalization techniques return a different node representation $z_{v,k}(T)$ for each agent at the end of the training instead of a common one $z_v(T)$. The personalization techniques used for the model parameters $\theta_k(t - \frac{1}{2})$ in Section 2.3.2 can be applied to the node representations $z_{v,k}(t - \frac{1}{2})$ to address statistical shifts for graph data and task heterogeneity. $z_{v,k}(t)$ is constructed by combining $z_{v,k}(t - \frac{1}{2})$ and $\bar{z}_v(t)$ using MLP (Chen et al., 2021) or a learnable linear interpolation (Wang et al., 2024).

**Transfer learning.** Rather than aggregating the node representations $\{z_{v,k}(t - \frac{1}{2})\}_{k \in \mathcal{K}}$, their difference between agents can be used to design *transfer learning.* Typically, it is assumed that a model has been trained by "source" agents $s$ on a sufficient amount of vertically partitioned subgraph data, and this knowledge is transferred to a target agent $t$. Aligning these representations while $t$ trains its local model allows to adapt the frozen model of $s$ to $t$'s domain. This alignment can be achieved by training against an adversarial model (Shen et al., 2020), trained to distinguish between source and target domains (Guan et al., 2021), or using a knowledge distillation loss (Hinton et al., 2014; Mai et al., 2024).

**Aggregating graph representations.** Lastly, in the case of vertical partition of full graph instances (see Figure 15), multiple graphs held by different agents can characterize the same item or person. To increase the generalization capabilities of graph-level tasks, (Dong et al., 2021) suggests averaging the graph embeddings to increase prediction accuracy, considering the specific use case of health applications.

**Open challenges.** Most work on vertical graph partitions focuses on basic aggregation of aligned representations. However, practical implementations come with open challenges, as they often include multi-modal features on one side, and have significant communication or privacy constraints on the other side, leading:

- *Towards transfer learning for multi-modal aligned partitions.* Current work on vertical partitions primarily uses simple averaging of aligned node representations. However, cross-organizational collaboration typically involves multi-modal heterogeneity: entities align, but features and topology differ (e.g., banks and social platforms sharing customer data). While transfer learning (Frasca et al., 2024; Wang et al., 2025) and knowledge distillation (Tian et al., 2025) are well-studied for centralized GNNs, their adaptation to collaborative multi-modal settings remains underexplored. A key challenge is negative transfer: structural diversity across agents means transfer may not always be beneficial, requiring methods to identify when knowledge transfer is appropriate across heterogeneous collaborators.

- *Towards efficient and private aggregation for aligned entities.* Current work merges all aligned node representations uniformly, regardless of importance. However, vertex-cut partitions can involve hundreds of aligned nodes, creating substantial communication overhead and privacy exposure. For

graphs, structural heterogeneity means nodes vary significantly in their contribution to learning (centrality, boundary position). Additionally, aggregating node representations leaks inference data beyond model or gradient inversion attacks. Developing selective aggregation strategies that prioritize informative nodes while minimizing leakage remains an open challenge.

---

**Training and inference under aligned partitions** adapts aggregation and transfer learning strategies from vertical federated learning (Section 2.3.3) to merge complementary views of the same graph entities held across agents. Unlike horizontal partitions, where agents observe complete but disjoint samples, vertical and vertex-cut partitions split information about the same entities across agents. For effective learning and inference, local representations are aggregated, or personalized collaboratively. Transfer learning over alignment entities enables collaborative learning and isolated inference. Challenging open questions include multi-modal alignment and increasing efficiency and privacy through sampling.

**Trilemma.** Merging aligned node representations creates substantial communication overhead, while transfer learning reduces inference-time communication at the cost of potential negative transfer when structural diversity is high. Node representation aggregation exposes topology and inference data beyond model or gradient inversion attacks, with aligned entities creating greater attack surfaces that require privacy-potection. The process of aligning nodes or subgraphs themselves requires privacy guarantees.

---

### 3.5 Improving Inference and Learning Efficiency

While Section 2.4 reviewed general design principles to improve efficiency in collaborative learning under realistic resource constraints for Euclidean data, we now focus on how these principles translate to graph data specifically. The most important difference between the Euclidean and graph-based scenarios is that due to message passing, efficiency now needs to be considered both for training and for inference, and during training, model updates with gradient backpropagation may also use communication resources. We discuss **aggregation patterns**, **asynchronous aggregation**, and **communication efficiency**. Table 2 shows that existing work concentrates primarily on aggregation patterns, where solutions depend heavily on the graph partitioning.

### 3.5.1 Aggregation over a network

The centralized, decentralized, and hierarchical aggregation patterns, introduced in Section 2.4.1, are all applicable for learning with graph data. They can be employed to aggregate the local models $\theta_k(t)$ during *collaborative training* and to aggregate node embeddings in the message passing equation (19), during *collaborative inference.*

Under collaborative learning with Euclidean data, the communication graph $W$ for the model aggregation is determined mainly by the physical network topology, that is, by the possible communication links or paths among the agents. This is true as well for collaborative learning over multiple graph instances. However, in the subgraph or ego-graph partition cases, the computational graph $\mathcal{G}$ (or its matrix representation $A$) has to be taken into account as well, both for inference and for the local model update.

The coupling of the computation graph $\mathcal{G}$ and the communication graph given by $W$ typically follows one of two paradigms, as illustrated in Figure 19.

- *Communication graph derived from subgraph partitioning of $\mathcal{G}$.* In this case, the communication topology is consistent with the computational one, meaning that certain edges in $\mathcal{G}$ also represent communication links in $W$. For example, in a swarm of robots (Blumenkamp et al., 2022; Nazzal et al., 2024), $\mathcal{G}$ connects physically adjacent robots, thereby defining both computation and communication topologies. In ego-graph settings, the communication graph is identical to the computation graph. For edge-cut subgraph partitions, communication links correspond to edges cut between agents, as in IoT systems where agents monitor different subsets of nodes (Pan et al., 2023; Naz-

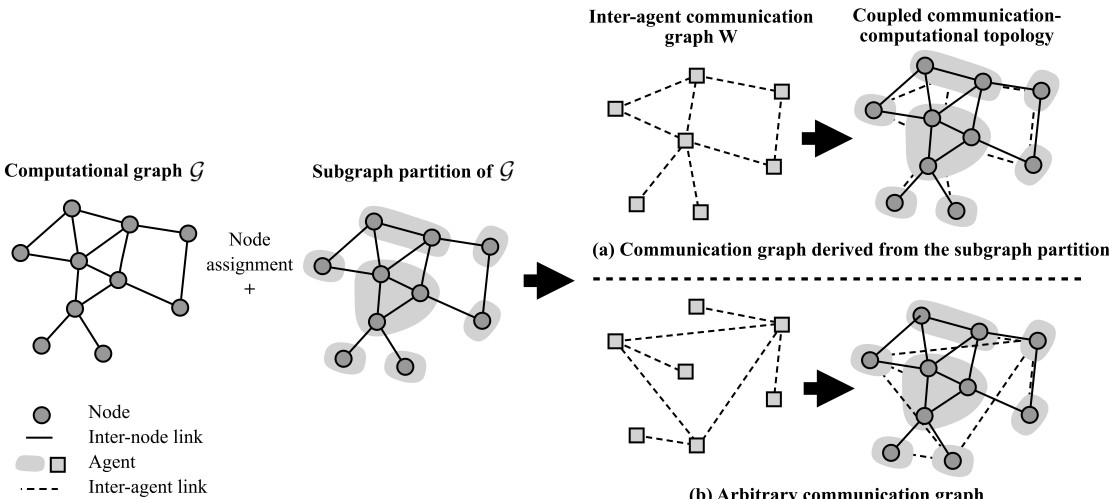

Figure 19: **llustration of the coupling between computation and communication topologies in the subgraph case.** The computational topology, represented by graph $\mathcal{G}$, and the communication topology, represented by matrix $W$, can be: (a) *derived from* the partitioning of $\mathcal{G}$ between agents, such as an edge-cut partition (see Section 3.3.1), or (b) *independent from* $\mathcal{G}$, representing an arbitrary communication network.

zal et al., 2024; Zeng et al., 2022). The node-to-agent assignments in $\mathcal{G}$ may be given as problem constraints, for example, determined by agent locations (Nazzal et al., 2024; Zeng et al., 2022), or optimized given the problem constraints (Pan et al., 2023; Zeng et al., 2022).

- *Communication and computational graph decoupled.* Here, computation and communication are decoupled. Typically, when peer-to-peer communications among all agents are allowed (Giaretta & Girdzijauskas, 2023; Qu et al., 2023) ($W$ forming a fully connected graph), the communications graph can be further refined adaptively to accommodate learning efficiency (Wang et al., 2025).

Typically, agents aggregate their model parameters and embeddings through the same topology. However, distinct communication topologies can be employed for model and embedding aggregation, allowing the topology to differ between training and inference (Lei et al., 2023; Giaretta & Girdzijauskas, 2023; Guo et al., 2024; Wang et al., 2025). In hierarchical architectures, centralized or decentralized aggregation patterns may occur for model updates and for inference at different levels of the hierarchy (Guo et al., 2024). Due to this increased flexibility, we discuss solutions for embedding aggregation and for model aggregation separately.

**Embedding aggregation.** In the case of subgraph partitions, embeddings $h_v^{(l)}$ generated by different agents must be aggregated either to perform message passing (line 13 of Algorithm 3) or for vertex-cut processing (see Section 3.4.3). Embedding aggregation techniques aim to adapt to constrained communication topologies while reducing both the communication cost of transmitting embeddings (line 12) and the computational cost of aggregating them (line 13) during collaborative inference in Algorithm 3. Solutions are proposed for centralized and for decentralized aggregation topologies:

- *Centralized communications* are the default approach inherited from FL algorithms and Algorithm 3. The server collects and redistributes embeddings and is sometimes responsible for maintaining them (Liu et al., 2022; Wu et al., 2021; 2022; Yan et al., 2024).

- *Decentralized communications,* leading to truly decentralized inference, means that embeddings are exchanged directly between the agents. Decentralized inference is the default for edge-cut ego-graph partitions, where the communication pattern naturally aligns with the computation pattern (Pan & Zhu, 2022; Solodova et al., 2025; Guo et al., 2024) However, decentralized inference also appears

in subgraph edge-cut partitions, for example, when agents represent institutions or edge servers monitoring a subset of nodes (Lei et al., 2023; Pan et al., 2023; Nazzal et al., 2024; Zeng et al., 2022; Wang et al., 2023). In such cases, the communication graph can be inferred from the cut of $\mathcal{G}$ (Zeng et al., 2022) or from the communication coverage area of the servers (Nazzal et al., 2024), as shown in Figure 19.

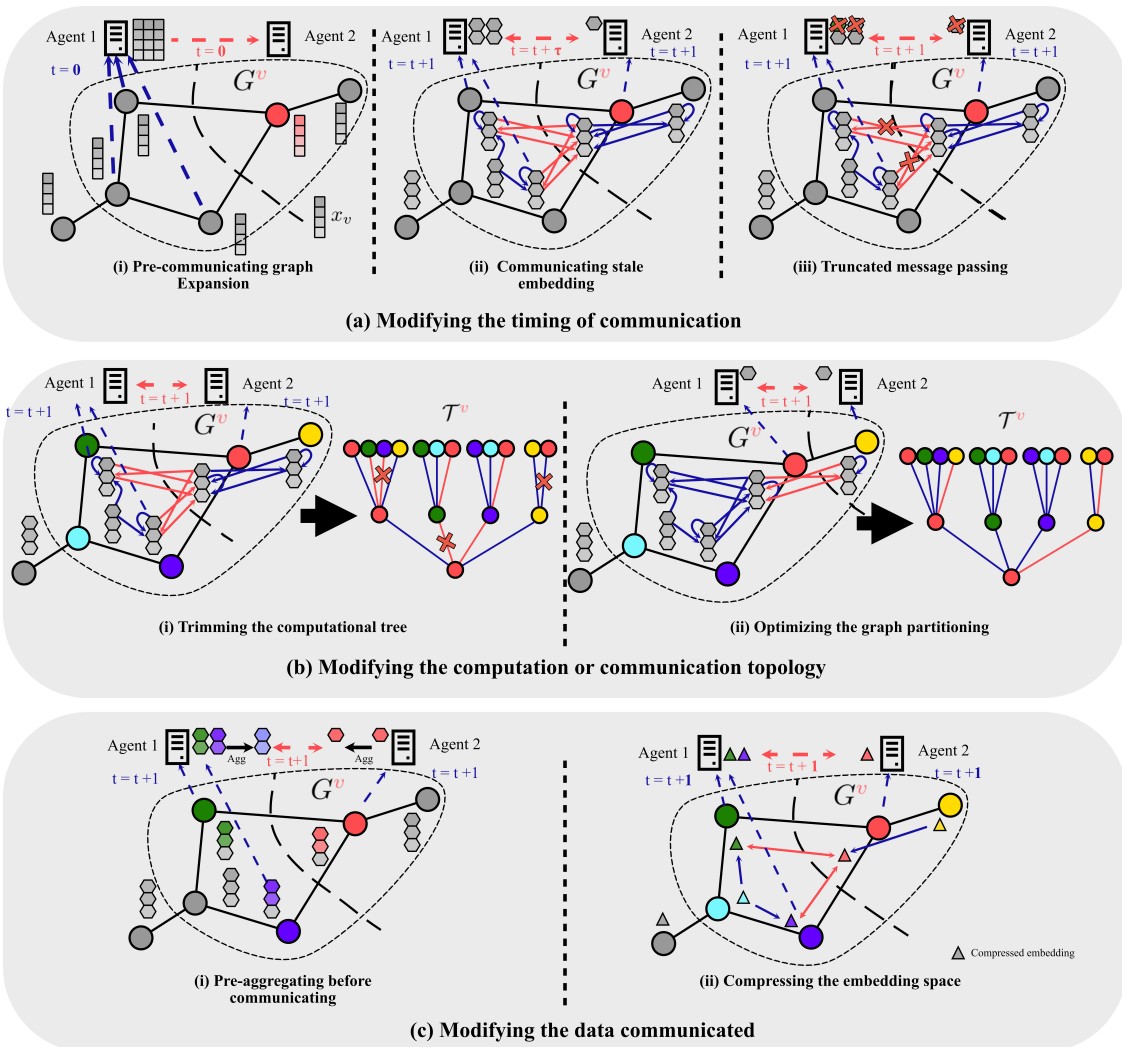

Figure 20: **Embedding aggregation techniques.** Methods focus on modifying: (a) the timing of communications, (b) the computation or communication topology, or (c) the data communicated

The embedding aggregation techniques for increased efficiency modify the time of the aggregation, the topology, or the content of the exchanged data, as summarized in Figure 20.

The first line of work focuses on modifying the *timing* of embedding aggregation to reduce the communication cost using:

- *Pre-communication.* Communicating sufficient information before the start of the training can avoid the need for a collaborative inference. Typically, the expanded subgraph data $\tilde{G}_k$ is communicated to relevant agents or the server prior to the first training round, so that agents perform isolated inference on $\tilde{G}_k$, proposed for centralized aggregation (Zhang et al., 2022; Chen et al., 2021; 2022) or through a decentralized protocol (Agrawal et al., 2024; Qiu et al., 2022). This uphill communication

cost can prove worth it if the algorithm takes many training rounds to converge in the collaborative inference, ultimately outweighing it.

- *Stale embeddings.* Letting node embeddings $h_v^{(l)}(t)$, requested normally in every round in the default Algorithm 3 (line 12), become stale reduces the communication cost of collaborative inference. Therefore, $h_v(t)$ is proposed to be requested periodically in (Du & Wu, 2022). To avoid affecting learning effectiveness due to this staleness, (Du & Wu, 2022) derives an optimal sampling rate w.r.t. the tradeoff between a communication budget and the convergence speed of training.

- *Truncated message passing.* Alternatively, to reduce the amount of information to be communicated in each round, (Liu et al., 2021) suggests communicating only a part of the embeddings, specifically only the later layers $l$, eliminating layers that have less contribution to the learning process.

The literature focuses on modifying the computation and communication topology to reduce communication overhead. During inference, the representation $z_v$ for node $v$ is obtained by propagating and aggregating messages along the edges of the input graph, following the procedures in (18), (19), and (20). This sequence of computations induces a *computational tree* $\mathcal{T}^v$ rooted in $v$, directly derived from its ego-graph $G^v$. In the case of subgraph partitioning, edge cuts in the input graph translate into inter-agent communication links in the computational tree. Reducing communication during collaborative inference thus often amounts to modifying $\mathcal{T}_v$ by:

- *Trimming the computational tree* Trimming the computational tree $\mathcal{T}^v$ can significantly reduce the communication needed. Tree-trimming strategies proposed in the literature (Liu et al., 2021; Chen et al., 2021; Pan & Zhu, 2022; Pan et al., 2023) remove selected branches of $\mathcal{T}_v$ leading to node $v$, thereby reducing both computation and communication, and notably the communication when the removed branches involve inter-agent communication. Such strategies can be determined locally by each agent (Liu et al., 2021), applied to its local expanded topology $\tilde{\mathcal{G}}_k$, assumed to be known; or centrally by a server with knowledge of the global graph $\mathcal{G}$ (Chen et al., 2021; Pan & Zhu, 2022; Pan et al., 2023). The communication workload can be balanced, either at the global level, e.g., minimizing the maximum workload across agents (Pan et al., 2023), or at the individual level by balancing each agent's workload (Liu et al., 2021; Chen et al., 2022). While trimming $\mathcal{T}^v$ reduces the computational and communication load, it also discards part of the knowledge contained in the subgraph $G^v$, potentially degrading model performance. The impact of the computational tree depth on prediction accuracy has been demonstrated in real-world applications such as robot swarm localization (Zhou et al., 2021; Blumenkamp et al., 2022). The trade-off between efficiency and accuracy can be balanced by various techniques, including selecting nodes that contribute only marginally to loss reduction (Liu et al., 2021) or using reinforcement learning to guide branch removal (Chen et al., 2021). In homophilous graphs, shortcut jumps within $\mathcal{T}^v$ can be used to reach a target node $v$ in fewer communication steps (Pan & Zhu, 2022).

- *Graph partitioning.* Another approach is to optimize graph partitioning to minimize communication cost shown by $\mathcal{T}^v$. Partition refinement can be achieved by reassigning nodes dynamically to agents to reduce the number of edge cuts (Nazzal et al., 2024). Node assignment can also account for other system-level constraints such as data acquisition cost or the number of agents (Zeng et al., 2022).

Modifying the nature of the data exchanged can significantly reduce communication costs. This can be achieved by employing simpler architectures than the generic MPNN described in Section 3.4.3, achieved with techniques such as:

- *Pre-aggregating before communicating,* Embeddings can be pre-aggregated before being communicated, thus reducing the amount of communication. When the expanded subgraph $\tilde{G}_k$ is transmitted for isolated inference, communicating the full $L$-hop neighborhood may still involve a prohibitive amount of node data. A practical solution is to leverage the properties of simple GNNs such as GCN (Kipf & Welling, 2017) or SGC (Wu et al., 2019), which use non-learnable $f_{\text{Mes}}$ and $f_{\text{Agg}}$,

to pre-aggregate the required embeddings locally before transmission. For example, in a GCN, the data of an $l$-hop neighborhood of a private boundary node $v \in \mathcal{V}_{p,k}$ can be pre-aggregated as $\sum_{u \in \mathcal{V}_k | (u,v) \in \mathcal{E}_k} x_u$ instead of transmitting the set $\{x_u \mid u \in \mathcal{V}_k, (u,v) \in \mathcal{E}_k\}$.(Yao et al., 2023) This reduces the cost of communicating an $l$-hop neighborhood to that of an $(l-1)$-hop neighborhood for GCN, and to that of the node $v$ itself for SGC. For SGC, the local inference cost can be further reduced by pre-computing the non-learnable operations (Lei et al., 2023).

- *Squashing the embedding space,* The dimensionality of node embeddings $d_x$ directly impacts communication cost. In node-level prediction tasks, an agent can locally generate pseudo-labels $\hat{y}_v$ by applying a feature encoder $f_{\mathrm{Enc}}$ followed by a local predictor $f_\psi$. Instead of transmitting full embeddings, only the scalar pseudo-labels $\hat{y}_v$ are sent, substantially reducing the communication load. Although these pseudo-labels do not initially incorporate graph topology, they can be refined via non-learnable diffusion algorithms (see Section 2.4.1), which propagate labels across the graph in a topology-aware manner. Under the homophily assumption, this allows even unlabeled nodes not involved in the collaborative inference to receive meaningful predictions $y_v$ (Huang et al., 2021). This approach has been implemented in decentralized settings for node-label prediction, achieving minimal communication cost while maintaining competitive performance (Krasanakis et al., 2022).

**Model aggregation.** For model aggregation, centralized, decentralized, and hierarchical architectures are proposed, very similarly to the Euclidean scenarios. The choice, however, is largely determined by the type of graph partition and by the communication costs caused by the backpropagation of the gradients.

- *Centralized model aggregation*, inherited from Algorithms 1 and 3, is the dominant approach in cross-silo scenarios where network constraints are moderate. It is commonly used for Multiple Graph Instance settings (He et al., 2022), Vertical Subgraph settings (Chen et al., 2021), and, less frequently, for cross-silo subgraph cases (Yao et al., 2023). As in Euclidean FL, periodic aggregation trades off convergence speed and communication cost. For Euclidean data, prior work typically assumed generic data distributions and derived convergence bounds from the local objectives (2) (Stich, 2019; Yu et al., 2019). In the graph case, structural assumptions such as graph homophily allow a refined convergence characterization (Yao et al., 2023) as a function of the input graphs $\{G_k\}_{k \in \mathcal{K}}$. In particular, (Yao et al., 2023) uses a stochastic random graph model (Keriven et al., 2020) with fixed homophily for node-level tasks to analyze convergence in the subgraph edge-cut configuration using a GCN $\mathcal{F}$. Their analysis covers both i.i.d. and label-shift scenarios, showing that convergence depends on the structural shift between each $G_k$ and the global graph $G$. Higher graph homophily, lower label shift, and smaller structural shift, achieved via higher-hop graph expansion, improve convergence. As the number of agents $K$ increases, distributional shifts intensify, requiring proportionally larger hop expansions to sustain performance.

- *Decentralized synchronous model aggregation* is typically used in ego-graph scenarios (Olshevskyi et al., 2025) and has been extensively analyzed for subgraph edge-cut cases (Scardapane et al., 2020). The mixing matrix $W = [w_{kl}]_{(k,l) \in \mathcal{K}^2}$ corresponds to the adjacency matrix of the communication graph derived from the subgraph edge-cut. Convergence proofs rely on standard FL assumptions of loss similarity and training stability (Stich, 2019; Yu et al., 2019). Stability requires $W$ to be doubly stochastic and connected. Dual-based distributed aggregation inspired by ADMM (Boyd et al., 2011) has also been proposed (Scardapane et al., 2020). Consistent with Euclidean FL, frequent aggregation accelerates convergence and improves resilience to link loss. Additional robustness to link failures is achieved in (Gao et al., 2022) by centrally training a GNN to perform distributed averaging that tolerates both link loss and partial participation. Other works propose to use a different mixing matrix at each round, subsampled from the original communication graph to limit communications. This subsampling optimizes the mixing matrix at every round using an actor-critic model, trained to minimize convergence time. During training, backpropagating the gradient (19 of Algorithm 3) requires communication between agents for collaborative inference and local gradient aggregation. This communication challenge can be mitigated (Olshevskyi et al., 2025) through batching and message piggybacking: gradients are backpropagated along the graph during

each sample's backward pass with small messages, while expensive parameter consensus occurs only once per mini-batch, amortizing the communication cost across multiple samples.

- *Decentralized asynchronous model aggregation* uses gossip-based protocols (Zheng et al., 2023; Giaretta & Girdzijauskas, 2023), though the derivation of the formal convergence rates remains an open problem. (Giaretta & Girdzijauskas, 2023) propose asynchronously exchanging parameters at the layer level—aggregating different layers at different times—to enhance robustness to asynchronicity.

- *Hierarchical aggregation*, discussed in (Guo et al., 2024), improves the scalability of learning by combining local peer-to-peer aggregation with occasional centralized coordination, while collaborative inference is proposed to remain peer-to-peer, to avoid central bottlenecks.

Beyond topology choice, communication cost can be reduced by adjusting the timing of model aggregation:

- *Pre-training* performs part of the training centrally before collaborative learning, avoiding communication during early training stages (Krasanakis et al., 2022; Gao et al., 2022). A hybrid approach initializes parameters from a centrally pre-trained model (Chen et al., 2022) before fine-tuning collaboratively with Algorithm 3.

- *Local SGD* runs several local epochs between aggregation rounds. As in standard FL (Section 2.4.1), this trades reduced communication for slower model synchronization. This trade-off has also been studied in the subgraph setting (Yao et al., 2023).

**Open challenges.** While aggregation techniques are well-studied, key questions remain about aligning computational and communication topologies, establishing the viability of highly decentralized inference, and understanding the role of localized learning in GNNs leading:

- *Towards separating input, computational, and communication graphs.* Current work treats computational topology $\mathcal{G}$ and communication topology $W$ as independent. However, graph rewiring, that is, separating the input graph $\mathcal{G}$, that is, the graph observed, from the computational graph, that is, the graph defining the node computational trees $\mathcal{T}^v$ (Figure 12), improves scalability in centralized settings (Wang et al., 2019; Fatemi et al., 2021; Kazi et al., 2022; Topping et al., 2022; Gutteridge et al., 2023). Further research could explore therefore collaborative solutions that jointly optimize input, computational, and physical topologies to improve efficiency, for example by aligning message-passing with the available communication links.

- *Towards establishing viability of highly decentralized inference.* Most work assumes centralized or modestly decentralized aggregation as seen in Table 2). However, highly decentralized scenarios (millions of agents representing one graph node each) with asynchronous, resource-constrained environments remain largely unexplored. Whether decentralized message-passing can remain correct and stable under realistic conditions (e.g., asynchrony, heterogeneity, partial participation, or justify its complexity over simpler alternatives, remains uncertain (Scardapane et al., 2020; Blumenkamp et al., 2022), and further research is needed to understand performance bounds and practical benefits.

- *Towards modular frameworks with topology as post-processing.* Current work trains end-to-end MPNNs coupling feature transformation with message-passing, requiring substantial embedding communication. However, deep MPNNs may be unnecessarily complex for certain tasks, such as node classification on homophilous graphs (Krasanakis et al., 2022). Agents could train Euclidean models locally, then apply lightweight graph post-processing (Zhu & Ghahramani, 2002), substantially reducing communication. Further research is needed to understand the limitations of this simplification (Bechler-Speicher et al., 2025), but also the possibilities it brings, for example, providing a way to utilize foundation models.

**Aggregation techniques for graph-structured data** extend centralized, decentralized, and hierarchical patterns from Euclidean collaborative learning (Section 2.4.1) to accommodate dual aggregation needs: model parameters during training and node embeddings during inference. Unlike Euclidean settings, where communication topology depends solely on physical networks, message passing couples computational topology $\mathcal{G}$ with communication topology $W$, either deriving $W$ from edge cuts in $\mathcal{G}$ or maintaining independent topologies. This coupling enables graph-specific optimization: computational tree trimming removes marginally contributing branches, pre-aggregation using simple GNNs reduces transmission to lower-hop neighborhoods, and dimensionality reduction transmits pseudo-labels instead of full embeddings. Future research is needed, however, for the joint optimization of learning and network design, for the support of highly-decentralized cases, and for resource-efficient aggregation with modular learning frameworks.

**Trilemma.** Efficiency techniques (tree trimming, stale embeddings, pre-aggregation) reduce communication costs but degrade model effectiveness, with impact depending on structural properties (homophilous level, critical branches of trees). Pre-communicating expanded subgraphs exposes neighborhood structure, embedding exchanges expose intermediate representations during inference, and decentralized aggregation distributes exposure across agents rather than centralizing it, requiring privacy preserving schemes.

### 3.5.2 Asynchronous Aggregation and Delay Control

The synchronous standard Algorithms 1 and 3 assume synchronicity of agents for model aggregation and message aggregation. However, in practice, communication and computing delays can differ from agent to agent and can also vary in time, VFnot only due to device heterogeneity, but also due to structural shifts in the graph topology. This leads to stale model parameters and embeddings, which challenge the efficiency of the learning. Therefore, solutions that address the communication delays have been explored. Conversely, asynchronicity in the form of introduced delay has also been exploited to improve the effectiveness of models.

**Model aggregation under asynchronicity.** In collaborative training of GNNs, as in Algorithms 1 and 3, the model aggregation is similar to traditional FL. Consequently, techniques and convergence results from Section 2.4.2 apply to GNN. With subgraph partitioning, the communication graph often matches the computation graph, making the system topology-dependent. This ties aggregation issues to asynchronous aggregation over graphs, as studied in (Lian et al., 2018).

**Embedding aggregation** under asynchronicity. Standard GNN inference assumes synchronous message passing between nodes (18),(19), and (20), which introduces delays and requires synchronization steps. To avoid these, message passing under asynchronicity is considered, with two complementary approaches: to mitigate the effect of the delays, or conversely, to exploit delays to improve model performance.

Staleness is typically reflected in the index $t$ of the node embeddings $h_v^{(l)}(t)$, arising due to imperfect synchronization steps (lines 12 and 13 of Algorithm 3) during collaborative inference. Some works suggest to introduce topological delays or acceleration, that changes the hop count index $l$ in the message aggregation (19), either at the level of the aggregated messages $m_{v,u}^{(l)}$ or in the neighborhood $\mathcal{N}^{(l)}(v)$ (Gutteridge et al., 2023; Chen et al., 2024).

The main solutions with asynchronous embedding aggregation focus on:

- *Modifying the collaborative inference protocol* to maintain model performance despite embedding staleness. GNN training with stale embeddings is studied in (Fey et al., 2021; Peng et al., 2022). To reduce communication costs during training, these works propose to skip certain embedding transmissions and derive convergence bounds that account for the resulting staleness. Message passing under asynchronicity is also addressed in (Yu et al., 2024), where the communication and processing burden of stale embeddings is reduced by sequentially performing updates along predefined paths in the graph, referred to as update chains.

- *Modifying the MPNN to mitigate staleness.* Specifically, (18), (19), and (20) are modified to improve robustness to embedding staleness during collaborative inference. For example, Energy GNNs (Gu et al., 2020), presented as a learnable version of diffusion algorithms on networks (see Section 2.4.1) allow for asymptotic inference that is provably robust to stale embeddings (Solodova et al., 2025). Alternatively, asynchronous MPNNs (Faber & Wattenhofer, 2024) support scenarios in which a feature update at an origin node triggers asynchronous updates throughout the graph. These models propose integrating the origin node's state into the messages (18) to mitigate the effects of stale embeddings (Faber & Wattenhofer, 2024). Another asynchronous MPNN class is proposed in (Mathys et al., 2024), where node updates are propagated outward from the origin up to distance $L$ (flood phase) and then back (echo phase), a process found to be more effective than standard MPNNs.

- *Modifying the MPNN to improve expressiveness.* Variants of the MPNN have also been proposed to incorporate topological delays for enhanced expressiveness. These delays can be controlled by learnable node actions (e.g., listening, broadcasting, or remaining inactive) (Finkelshtein et al., 2024). Embedding updates may be delayed or accelerated based on topological considerations (Gutteridge et al., 2023; Chen et al., 2024), typically by modifying the aggregation step (19). Delays arise when $l$-hop embeddings from 1-hop neighbors are used, whereas accelerations occur when embeddings from nodes beyond 1-hop are included. These adjustments help address oversmoothing (Chen et al., 2020) and oversquashing (Topping et al., 2022), two expressive limitations of GNNs.

**Open challenges.** While asynchronous MPNN architectures are advancing, key questions remain about adapting to dynamic agent capabilities and thus dynamically changing staleness, and about handling extreme staleness regimes in prolonged-delay environments leading:

- *Towards dynamic capability-aware MPNNs.* Current work assumes homogeneous agents with static capabilities. However, practical deployments involve dynamic fluctuations in computational power, memory, and bandwidth due to varying workloads and network conditions. While prior work balances static capability constraints (Liu et al., 2021; Chen et al., 2022; Zeng et al., 2022), dynamically adapting message-passing depth, embedding dimensions, or aggregation responsibilities to runtime capability changes remains unexplored. For graphs, this requires the challenging coordination of computational tree traversal across agents operating at varying speeds.

- *Towards aggregation strategies for extreme staleness regimes.* Current asynchronous methods handle moderate staleness. However, prolonged delays (satellite networks, sensor networks with intermittent connectivity) create extreme staleness and cyclical connectivity patterns. Key challenges include heterogeneous embedding ages across nodes, trade-offs between recent-but-distant versus stale-but-neighbor embeddings, and exploiting predictable patterns for optimization. Developing aggregation strategies that leverage temporal patterns when staleness varies by orders of magnitude across computational trees remains unexplored.

---

**Asynchronous aggregation techniques for graph-structured data** extend asynchronicity handling from Euclidean federated learning to address staleness in both model parameters and node embeddings. Unlike Euclidean settings, where asynchronicity affects only model synchronization, graphs require managing embedding staleness during message-passing. Two strategies emerge: mitigating delays through protocol modifications (skipping transmissions, sequential updates) or exploiting delays for effectiveness (topological acceleration/deceleration addressing oversmoothing and/or oversquashing). Future research is needed to handle dynamically changing staleness and emerging systems with extreme delays.

**Trilemma.** Skipping embeddings or using stale values reduces communication but degrades effectiveness. Specifically, deep computational trees and high-degree nodes are more sensitive to staleness. Asynchronous protocols expose temporal patterns (update timing, message sequences) and topology through sequential update paths, creating attack surfaces beyond synchronous aggregation.

### 3.5.3 Communication Efficiency

For collaborative GNNs with topology partitioning, embedding vectors must be communicated among agents during the message-passing phase. These exchanges can occur in centralized, peer-to-peer, or mesh architectures and are subject to heterogeneous, unreliable, and costly communication links. Similar to model aggregation in traditional FL, the efficiency and reliability of embedding aggregation depend on the state of the communication links and the cost of transmission.

However, communication for message passing differs from conventional data exchange: embeddings are intermediate representations, and the downstream learning task typically depends on aggregated functions of these embeddings rather than their exact values. This observation motivates the design of communication schemes tailored to collaborative GNNs, optimizing transmission efficiency, adapting embedding aggregation to wireless communication, and communication patterns to the network state.

**Transmission efficiency.** Transmission efficiency quantifies the utility of the transmitted embeddings relative to the consumed communication resources, such as bandwidth and transmission power. In collaborative GNNs, this is particularly critical when each agent holds only a small portion of the graph, potentially a single node, as in fully decentralized settings. In (Lee et al., 2021), a coding and retransmission scheme is proposed to improve the reliability of embedding aggregation, aiming to maximize inference accuracy under constrained resources. The work in (Wang et al., 2025) proposes a node subsampling strategy for selecting communication edges that accommodates link bandwidth constraints, to minimize convergence time during training. Theoretical analysis for the same scenario is provided in (Gao et al., 2021), where disruptions in embedding communication are modeled by a probability of failed message passing due to wireless impairments or temporary disconnections. The results show that accurate inference remains possible if such stochastic failures are accounted for in the learning process.

**Wireless communication for embedding aggregation.** The principles of over-the-air computation, originally proposed for FL model aggregation, have been extended to embedding aggregation in collaborative GNNs (Gao & Gündüz, 2023; Lee et al., 2023; Gao & Gündüz, 2025). In this setting, the superposition property of the wireless channel is exploited to aggregate embeddings directly during transmission, reducing communication costs and allowing simultaneous updates from multiple agents. These methods have been shown to maintain robustness against channel impairments and, in some cases, operate without channel state information. Extensions to more complex and dynamic topological structures are explored in (Fiorellino et al., 2024). Practical demonstrations, such as (Blumenkamp et al., 2022), further highlight that variable delays and unreliable communication links must be considered in system design, while resource allocation in wireless mesh networks (Wang et al., 2022) has been shown to significantly affect application performance.

**Client scheduling and topology management.** Client scheduling and topology management for collaborative GNNs are mainly considered for subgraphs with topology partitioning in learning tasks over wide-area networks, where the message-passing operation is costly. In (Liu et al., 2021), collaborative GNNs in wide-area networks are considered. To reduce traffic congestion, client selection is performed based on the importance of the communicated embeddings and the communication costs. Scheduling algorithms are designed to allow interleaved message passing and local updates. A multi-tier wide-area network with edge servers is considered in (Zeng et al., 2022; Nazzal et al., 2024). The topology of the GNN is given by the application, and these papers address the problem of GNN node assignment to edge servers and edge-network topology design for communication-efficient training, for both static (Zeng et al., 2022) and dynamic (Nazzal et al., 2024) GNN topologies.

Most works on communication efficiency in collaborative GNN implementations assume that the communication graph given by the possible connections between agents is identical to the computing graph given by the adjacency matrix $A$. This assumption is not necessary. An algorithm to find the optimal communication graph for subgraphs with topology partitioning is proposed in (Scardapane et al., 2020). Conversely, (Wang et al., 2025) assumes a fully connected communication graph (i.e., agents are free to communicate with anyone in a peer-to-peer network) but samples a communication topology at each round. The authors design an

adaptive algorithm that samples communication topologies adaptively throughout training to accommodate bandwidth constraints while maintaining fast inference and training.

**Open challenges.** Existing work has started to address the challenges of communicating embeddings and model parameters over unreliable or costly communication channels or transmission paths. However, since both inference and learning involve communication, several interesting research questions remain open in the co-design of communication and computing, in utilizing the broadcast and multicast capabilities of the wireless channel, and in ensuring safe learning and inference despite the communication impairments leading:

- *Towards co-designing communication and computing.* While recent methods consider the effects of unreliable communication channels (Lee et al., 2021; Gao et al., 2021), they treat communication and computation topologies independently. Co-designing these topologies while explicitly accounting for channel quality represents an avenue for further efficiency improvements. Additionally, existing embedding aggregation methods operate on static neighbor selection policies and could be extended to dynamically exploit time-varying wireless channel quality by scheduling or selecting neighbors with temporarily favorable channel conditions.

- *Towards communication-native inference.* The wireless medium itself, with its superposition and broadcast properties, natively supports *many-to-one* and *one-to-many* communications. These properties could allow efficient embedding and model aggregation in a single operation among clusters of agents or agents connecting to the same base station or edge computing server. While initial results exist in (Fiorellino et al., 2024), further research is still needed to fully exploit these network environment properties through complex topological structures beyond simple graphs.

- *Towards safe training and inference.* Training and inference on graph data require communication in both phases. Consequently, communication impairments deteriorate inference quality even when the model itself is optimal, potentially leading to unsafe decisions. This raises the question of whether safety requirements should be incorporated into the learning process or considered only during inference, a design choice that requires further investigation.

> **Communication efficiency techniques for graph-structured data** adapt transmission optimization and topology management from Euclidean collaborative learning, but the importance of communication efficiency is increased due to the need for both message passing for inference and for backpropagation during training. Most results address learning effectiveness and efficiency directly, or suggest solutions that utilize the broadcast and superposition properties of the wireless channel. Challenges remain to increase effectiveness by additional network-specific solutions, but also to ensure safe decisions despite information loss both at inference and learning.
>
> **Trilemma.** Coding and retransmission schemes improve reliability under channel impairments but increase transmission overhead, while topology subsampling and neighbor selection reduce communication costs but risk slower convergence or degraded inference effectiveness. Over-the-air computation and wireless transmission expose intermediate embeddings to any eavesdropper within radio range.

## 3.6 Privacy Preservation Techniques for Graph Data

A high variety of privacy-preserving techniques are proposed for distributed graph data, aiming to protect training and inference data from *honest-but-curious* agents and central servers. These techniques address the vulnerabilities identified in Section 3.3.4, model parameter leakage from collaborative training, embedding, and topology leakage from collaborative inference. Anonymization has to target three main levels: *(i)* model parameters, *(ii)* exchanged node embeddings, and *(iii)* exchanged edge information. Many of these techniques build on the blind computation methods, and LDP introduced in Section 2.5. As shown in Table 2, privacy preservation is an important topic for collaborative GNNs, with most of the works focusing on cross-silo applications, with use cases handling personal information, like health data or user preferences.

**Anonymizing model parameters.** To address the inherited vulnerabilities from collaborative learning (Section 2.2.4), model parameters or gradients exchanged under GNN training can be anonymized directly with the techniques proposed for FL in Section 2.5 (Liu et al., 2022; Wu et al., 2022; Yao et al., 2023; Yan et al., 2024). These techniques can be further refined to improve the stability–privacy trade-off, such as *adaptive noise scaling* (Wu et al., 2022), in which the injected noise magnitude is adjusted according to the gradient norm to maintain a consistent perturbation throughout training.

**Anonymizing exchanged embeddings.** As discussed in Section 3.3.4, collaborative inference over graphs requires exchanging intermediate embeddings across subgraph boundaries, creating attack surfaces for feature extraction and membership inference. Early works like (Chen et al., 2021) incorrectly assumed that if local computations are performed entirely in isolation, including local model updates and isolated inference, then the raw node features $X_k$ would be inherently protected from privacy leakage. However, it is now clear that even when only intermediate embeddings $h_v^{(l)}$ are shared and aggregated, they must still be protected through some form of differential privacy (Chen et al., 2024).

- *Blind computation* methods securely aggregate embeddings (e.g., computing $f_{\text{Agg}}$ in (19)) or average representations $z_v(t)$ without revealing their values. For example, secure multiparty computation can be applied to jointly perform specific GNN functions such as aggregating $f_{Agg}$ (19) or computing matrix multiplications (Wang et al., 2023). Due to their high computational cost, these methods are often applied only to a subset of layers (Chen et al., 2021), with the remaining computations delegated to the server in a split-learning setup. Homomorphic encryption offers another option, allowing the server to aggregate results while keeping them encrypted end-to-end (Ran et al., 2022; Peng et al., 2023), using techniques such as secure multiparty computation (Chen et al., 2021) or homomorphic encryption for server-side aggregation (Ni et al., 2021).

- *LDP* techniques can be applied directly to the communicated vectors $z_v$ or $h_v^{(l)}$. The simplest approach adds Gaussian noise to the vectors (Zhang et al., 2023; Lee et al., 2023; Ni et al., 2021). More advanced methods adapt classical LDP protocols to the GNN setting. These generally follow a pipeline of noise injection, vector clipping, dimension subsampling, and privacy budget adjustment. Noise can follow a Laplace distribution (Wu et al., 2022; 2021), a Gaussian distribution (Chen et al., 2021), or a Bernoulli distribution (Zheng et al., 2023; Pan et al., 2023). Clipping may be applied to vectors or gradients (Wu et al., 2022; 2021; Chen et al., 2021; Liu et al., 2022; Yan et al., 2024) and can be combined with stochastic quantization (Lin et al., 2022; Zheng et al., 2023; Pan et al., 2023). Dimensionality reduction can be performed by random subsampling (Lin et al., 2022; Pei et al., 2023; Zheng et al., 2023; Pan et al., 2023) or top-$k$ sampling (Chen et al., 2022). In some cases, these operations are unified through complex noise functions such as piecewise noise injection (Pei et al., 2023; Wang et al., 2019). The privacy level can also be balanced against other objectives or system constraints. The noise added in LDP is known for degrading learning effectiveness (Lee et al., 2023), mitigated while aggregating high degree nodes (Guo et al., 2024). Therefore, adaptive noise strategies propose to scale the noise by the gradient norm during training to stabilize learning (Wu et al., 2022). When communication occurs over wireless channels, the trade-off between signal-to-noise ratio and DP can be jointly optimized (Lee et al., 2023). LDP can also be applied to node labels in transfer learning with adversarial losses to protect the training data of the source agent (Guan et al., 2021).

**Anonymizing edge (or topology) information.** Beyond node feature extraction, adversaries can extract existing edges from shared embeddings (Section 3.3.4), a threat known as *link stealing* (Pei et al., 2023). This represents a serious privacy attack when the existence of the edge itself represents sensitive information, or when the edges encode most of the information, as is often the case in heterogeneous graphs (Liu et al., 2022; Wu et al., 2021; 2022; Yan et al., 2024). Therefore, the edge or topology information needs to be protected in many scenarios. The concept of *Edge Local Differential Privacy (Edge LDP)* is formalized in (Wu et al., 2022; Han et al., 2024; Lin et al., 2022) and is extended to heterogeneous graphs in (Yan et al., 2024), providing privacy guarantees for graphs differing by a single edge. In some cases, protection

also covers statistics on the edge set, such as degree distributions (Pan et al., 2023). Protection strategies build both on blind computations and on LDP.

- *Blind computations* can be employed for *privacy-preserving subgraph expansion.* Each agent obtains its $L$-hop expansion $\tilde{G}_k$ of the local graph using blind computation techniques. The process begins with agents collectively running a private set intersection (PSI) protocol to identify the vertex cut $mathcalV_{cut}$ between agents. Given the knowledge of $\mathcal{V}_{cut} \cap \mathcal{V}_k$, each agent collects and communicates the missing expanded subgraphs, typically via a central server (Zhang et al., 2022; Chen et al., 2021; 2022; Agrawal et al., 2024; Qiu et al., 2022). Blind computation techniques can also be applied to aggregate useful graph statistics, such as node degree (Pan et al., 2023), using methods such as secure group aggregation (Bonawitz et al., 2017) or zero-knowledge protocols (Goldwasser et al., 1985). Such statistics may be required to compute auxiliary losses, for example, fairness losses (Agrawal et al., 2024), or may be used by the local learning algorithm (Pan et al., 2023).

- *Standard LDP* (Dwork & Roth, 2014) can be applied to edge information in set form $\mathcal{E}_k$ (Yan et al., 2024; Lin et al., 2022) or in adjacency matrix form $A_k$ (Han et al., 2024; Qiu et al., 2022). For example, (Zhang et al., 2021) projects $A_k$ into a lower-dimensional Gaussian random subspace. However, LDP introduces topological noise, which is known to degrade GNN performance even at low noise levels (Zügner et al., 2020). Excessive noise can densify the graph, leading to *oversmoothing* (Chen et al., 2020). Mitigation strategies include regularization (Lin et al., 2022), degree preservation (Yan et al., 2024), or allocating part of the privacy budget to maintain adjacency sparsity (Qiu et al., 2022).

Protecting edge information is often formulated alongside protecting the raw feature $X_k$, thus protecting the local graph data $G_k = (A_k, X_k)$. The concept of node-level privacy is introduced informally for the ego-graph case (Guo et al., 2024), consisting of protecting the raw features of nodes $v$ alongside the 1-hop topology typical of ego-graph partitions. This concept is further extended to heterogeneous graphs, specifically focusing on protecting nodes that correspond to users (Yan et al., 2024). Other works focus on protecting collections of ego-graphs sampled from $G_k$ (Zhang et al., 2023; Wu et al., 2021; 2022). Subgraph-level privacy extends this protection to entire subgraphs $G_k$ obtained from aggregating the private ego-graphs of the agents (Guo et al., 2024). Techniques that anonymize $A_k$ together with $X_k$ have been proposed in the following forms:

- *Blending ego-graphs together.* The embeddings of ego-graphs are averaged across all ego-graphs subsampled from $G_k$ in the same batch, respectively between nodes at the same depth in the computational tree, to anonymize the ego-graphs and their embeddings (Zhang et al., 2023).

- *Inferring pseudo-edges.* This common technique is used for anonymizing $A_k$ for heterogeneous graphs, usually in the vertex cut case (Liu et al., 2022; Wu et al., 2021; 2022; Yan et al., 2024). It consists of constructing unobserved yet highly probable edges based on node representation similarity, and adding them to the observed edges in $G_k$.

In hierarchical setups (e.g., device–edge and server–central server) (Guo et al., 2024) considers privacy protections between higher-rank and lower-rank entities as well as among peers of the same rank. The above anonymization techniques, focusing on model parameters, embeddings, and topology information, can then be selectively combined depending on the nature of the data exchanged at each hierarchy level.

**Open challenges.** Privacy-preserving techniques for collaborative GNNs operate at the intersection of two research domains: privacy mechanisms for GNNs and privacy-preserving collaborative training and inference. This convergence presents unique challenges and opportunities, as collaborative settings introduce multi-agent dynamics that both complicate and potentially enhance privacy guarantees, leading:

- *Towards multi-agent privacy amplification.* Most work applies LDP locally to individual training or inference data. However, LDP degrades learning effectiveness by compressing communicated

information (Smith et al., 2017; Bonawitz et al., 2022). Privacy amplification (Section 2.5) offers an alternative: agents sequentially amplify protection guarantees along collaborative computation chains rather than applying hard local anonymization. This approach fits naturally with sequential MPNN operations and could limit information loss while maintaining privacy guarantees, but is still unexplored for collaborative graph settings.

- *Towards integrating privacy with defense mechanisms.* Current anonymization techniques protect against honest-but-curious agents (Section 3.3.4) but are insufficient against byzantine malicious agents who actively manipulate graph structure or features. Privacy mechanisms are often treated as a first-line defense against attacks such as link stealing (Pei et al., 2023), embedding extraction (Chen et al., 2024), and graph rewiring (Chen et al., 2024). However, how privacy mechanisms can be combined with additional defenses (e.g., embedding filtering (He et al., 2024)) against sophisticated adversaries remains unclear, since privacy guarantees often hide the very signals needed for detecting these adversarial threats, calling for unified frameworks that jointly address honest-but-curious and malicious threat models. For instance, mitigating adversarial edge injection may require integrating verifiable private mechanisms (Daly et al., 2024) to ensure graph integrity without sacrificing privacy.

---

**Privacy-preserving techniques for graph-structured data** adapt blind computation and local differential privacy from Euclidean federated learning (Section 2.5), to protect not only the model parameters, but also the intermediate embeddings, and the topology information. This three-level protection requirement is amplified by repeated embedding exchanges across subgraph boundaries during multi-layer message-passing, creating cumulative exposure particularly severe for boundary nodes with many cross-agent connections. Additionally, graph-specific techniques are proposed to randomize edge and subgraph data. Since privacy preservation affects both efficiency and effectiveness heavily, further research is needed to fit the methods to the computation chains of GNNs and to effectively combine them with similarly costly security mechanisms.

**Trilemma.** Blind computation preserves data locality but is computationally heavy; LDP-based methods offer formal privacy guarantees for embeddings and edges but can degrade message-passing quality and inference accuracy. Edge LDP, in particular, introduces topological noise that may cause oversmoothing through graph densification, requiring mitigation strategies that can further constrain model effectiveness.

---

## 4 Conclusion

Collaborative learning has unlocked access to previously unavailable distributed data sources, leading to the success of federated learning and decentralized learning on Euclidean data, and more recently, to emerging solutions for graph-structured data. Extending collaborative learning to graph-structured data opens up opportunities that exceed those of centralized graph ML through its ability to mine data across domains ranging from sensor networks to molecular biology.

With this survey, we systematically map the design choices from Euclidean-structured to graph-structured collaborative learning. We first consolidate the foundational principles of collaborative learning on Euclidean data (Section 2), then extend this mapping to graph-structured data (Section 3). For both cases, we formulate the fundamental problem, discuss the challenges of data and system heterogeneity, and analyze how the trilemma of effectiveness, efficiency, and privacy preservation manifests in the design choices for addressing these challenges. Specifically for graph-data, we identify emerging research challenges that define its future development.

Our analysis reveals that collaborative learning on graph-structured data has given rise to a nascent, yet rich, landscape of problems and solutions. This landscape can be broadly divided into two paradigms: *isolated inference* and *collaborative inference.* Notably, collaborative inference emerges as a more central paradigm for graph-structured data due to the message passing mechanism of GNNs. Beyond challenges that persist in Euclidean collaborative learning, combining distributed relational data with collaborative training and/or

inference introduces unique constraints that require novel solutions. This includes the joint consideration of topologies, communication, privacy, and heterogeneities.

By systematically bridging collaborative learning on Euclidean and graph-structured data, this survey establishes a coherent vision to support researchers in developing the next generation of collaborative graph learning methods and to enable their practical impact across scientific and industrial domains, leading to societally important breakthroughs in drug discovery, healthcare, sustainable industries, and smart cities.

## Acknowledgement

This work was supported in part by Vinnova, Sweden´s Innovation Agency, under grant no. 2024-00648.

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

# A    Survey map, Notations and Glossary

**Organization of the solutions**

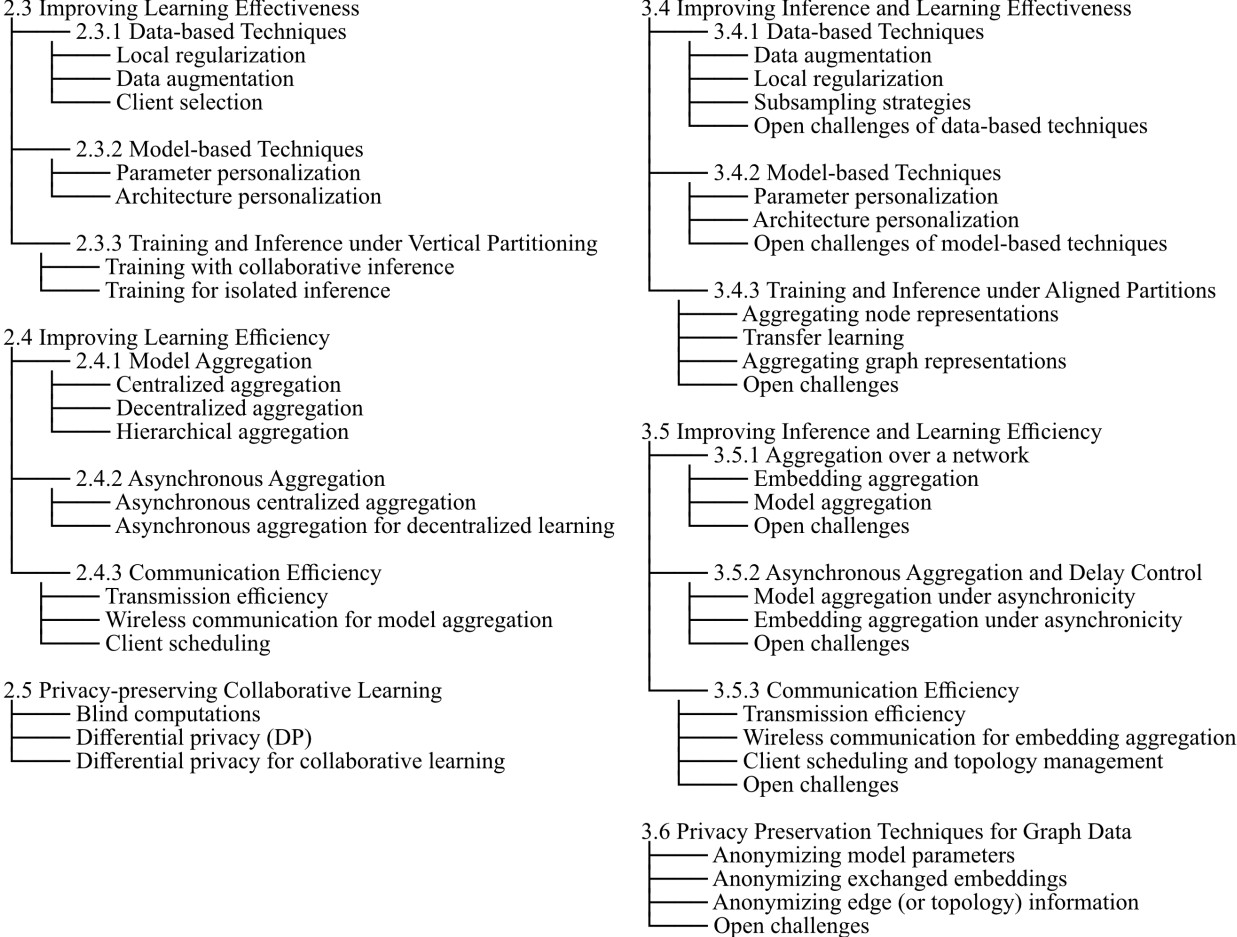

Figure 21: The taxonomy of the solutions surveyed for effective, efficient, and privacy-preserving collaborative learning for Euclidean and graph data.

**Glossary**

In this work, we use several terms whose definitions are either subjective or inconsistently used in the literature; we clarify them here for consistency and clarity. These definitions are limited to the present work.

**Agent**  An autonomous entity with local data, computational capabilities, and the ability to communicate with other agents.

**Server**  A central coordinating unit that supervises a group of agents, possessing computational capabilities and the ability to communicate with agents.

**Distributed**  A property of a system in which some or all components (e.g., data, computation, or control) are spread across multiple agents.

**Data parallelism**  A property of a distributed system where data collected centrally is distributed across multiple agents.

**Model parallelism** A property of a distributed system where parts of the model maintained centrally are distributed across multiple agents.

**Collaborative (system)** A distributed system of agents (either federated or decentralized) where agents cooperate to achieve shared objectives, though their willingness to collaborate may be limited by local constraints or incentives such as, but not limited to, local objective, computational and communication capabilities, data availability, or privacy requirements.

**Federated (system)** A distributed system of agents that are federated under the supervision of a central server coordinates.

**Decentralized (system)** A distributed system of agents where all agents operate without a central server.

**Local** Describes a property or object that pertains to a single agent (e.g., local data, local model).

**Global** Describes a property or object shared across or computed from all or many agents. It does not imply that each agent has a full view of this property (e.g., global graph).

**Topology** The discrete structure describing the communication or computational relationships between agents, typically modeled as a graph (see (154) for a formal equivalence between topology and graph).

**Sample** A single data point drawn from a dataset.

**Feature** An observed attribute or measurable property of a sample.

**Label** The target property or variable of a sample that we aim to infer from its features.

**Training** The process where models learn patterns from data by optimizing parameters to minimize prediction errors on the training dataset.

**Inference** The process by which the model predicts the label of a sample based on its features using learned parameters.

**Learning** The comprehensive framework encompasses various methodologies, algorithms, and theoretical foundations for training models to perform inference tasks effectively.

**Euclidean** Pertaining to data where each sample is represented as a feature vector in $\mathbb{R}^n$ equipped with the standard Euclidean inner product.

**Non-Euclidean** Pertaining to data with inherent relational structure that cannot be faithfully represented in $\mathbb{R}^n$ with the Euclidean inner product. Graphs are the most commonly used example of discrete non-Euclidean structures.

**Notations**

<table>
<tr><td colspan="2" align="center">Agents and Data</td></tr>
<tr><td>$\mathcal{D}_k, \mathcal{D}$</td><td>Local and global data distributions</td></tr>
<tr><td>$\mathcal{I}_k, \mathcal{I}, i$</td><td>Local and global index sets of samples, and sample index</td></tr>
<tr><td>$\mathcal{K}, K, k, k^*, s, t$</td><td>Set of agents, number of agents, agent index, active agent, source and target agent</td></tr>
<tr><td>$n_k, n$</td><td>Number of samples in local and global datasets</td></tr>
<tr><td>$\Omega$</td><td>Similarity matrix representing a latent graph between agents</td></tr>
<tr><td>$\Pi, \pi_i$</td><td>State of a Markov chain and its $i$-th element</td></tr>
<tr><td>$S_k, S$</td><td>Local and global datasets</td></tr>
<tr><td>$W \in \mathbb{R}^{K \times K}$</td><td>Communication topology matrix between agents</td></tr>
<tr><td colspan="2" align="center">Features, Labels, Representations</td></tr>
<tr><td>$d_x, d_y, d_z$</td><td>Dimensions of feature vector, label space, and embedding space</td></tr>
<tr><td>$x, \mathcal{X}$</td><td>Feature sample and feature domain</td></tr>
<tr><td>$x^i, x_k^i$</td><td>Feature of the $i$-th sample, global and local</td></tr>
<tr><td>$x_k, \mathcal{X}_k$</td><td>Local fraction of feature $x$ in local domain $\mathcal{X}_k$</td></tr>
<tr><td>$y, \mathcal{Y}$</td><td>Label and label domain</td></tr>
<tr><td>$y^i, \hat{y}^i$</td><td>Label of the $i$-th sample and its inferred value</td></tr>
<tr><td>$z, \mathcal{Z}$</td><td>Representation of sample $(x, y)$ and representation domain</td></tr>
<tr><td>$z_k, \mathcal{Z}_k$</td><td>Local fraction of global representation $z$ in local domain $\mathcal{Z}_k$</td></tr>
<tr><td colspan="2" align="center">Graphs and Topology</td></tr>
<tr><td>$A$</td><td>Adjacency matrix of $\mathcal{G}$</td></tr>
<tr><td>$\mathcal{E}_k, \mathcal{E}$</td><td>Local and global sets of edges</td></tr>
<tr><td>$\mathcal{E}_{cut,k}$</td><td>Local set of cut edges</td></tr>
<tr><td>$G = (A, X)$</td><td>Graph data sample in matrix form</td></tr>
<tr><td>$G_k, G_k^i$</td><td>Local subgraph and $i$-th local graph sample</td></tr>
<tr><td>$G^v, G_k^v$</td><td>Global and local $L$-hop neighborhoods of node $v$</td></tr>
<tr><td>$\mathcal{G}_k, \mathcal{G}$</td><td>Local and global graph structures (homogeneous)</td></tr>
<tr><td>$\tilde{G}_k, \tilde{\mathcal{G}}_k$</td><td>Local subgraph data and structure (from $G_k$ and $\mathcal{G}_k$), respectively, expanded to $L$-hop neighborhoods</td></tr>
<tr><td>$\mathcal{H}_{\mathcal{G}}$</td><td>Homogeneous projection of heterogeneous graph</td></tr>
<tr><td>$N_k, N$</td><td>Number of vertices in local and global vertex sets</td></tr>
<tr><td>$\mathcal{O}_k, \mathcal{O}$</td><td>Local and global sets of object types</td></tr>
<tr><td>$\mathcal{R}_k, \mathcal{R}$</td><td>Local and global sets of relation types</td></tr>
<tr><td>$\rho_k, \rho$</td><td>Local and global relation type mappings</td></tr>
<tr><td>$\mathcal{T}^v$</td><td>Computational tree for target node $v$</td></tr>
<tr><td>$\tau_k, \tau$</td><td>Local and global node type mappings</td></tr>
<tr><td>$\mathcal{V}_k, \mathcal{V}, v$</td><td>Local and global sets of vertices (nodes), and vertex index</td></tr>
<tr><td>$\mathcal{V}_{f,k}, \mathcal{V}_{p,k}, \mathcal{V}_{cut,k}$</td><td>Sets of foreign nodes, private nodes, and vertex cut</td></tr>
<tr><td colspan="2" align="center">Node-Level Quantities</td></tr>
<tr><td>$h_v^{(l)}, h_{v,k}^{(l)}$</td><td>Global and local embeddings of node $v$ at layer $l$</td></tr>
<tr><td>$X$</td><td>Feature matrix of the graph</td></tr>
<tr><td>$x_v, x_{v,k}$</td><td>Global and local features of node $v$</td></tr>
<tr><td>$z_v, z_{v,k}$</td><td>Global and local node representation of node $v$</td></tr>
<tr><td colspan="2" align="center">Models and Learning</td></tr>
<tr><td>$E, e$</td><td>Number of local iterations and local iteration index</td></tr>
<tr><td>$F_k, \mathcal{F}_k$</td><td>Local model and its function space</td></tr>
<tr><td>$F, \mathcal{F}$</td><td>Global model and its function space</td></tr>
<tr><td>$F_\theta, F_{\theta,\psi}$</td><td>Model $F$ parametrized by $\theta$ and by $(\theta, \psi)$</td></tr>
<tr><td>$f_{Mes}^{(l)}, f_{Agg}^{(l)}, f_{Up}^{(l)}$</td><td>Message, aggregation, and update functions at layer $l$</td></tr>
<tr><td>$f_\psi$</td><td>Predictor parametrized by $\psi$</td></tr>
<tr><td>$\mathcal{H}$</td><td>Hypernetwork</td></tr>
<tr><td>$\ell$</td><td>Pairwise loss</td></tr>
<tr><td>$L, l$</td><td>Number of layers and layer index</td></tr>
<tr><td>$\mathcal{L}_k, \mathcal{L}$</td><td>Local and global population losses</td></tr>
<tr><td>$\hat{\mathcal{L}}_k, \hat{\mathcal{L}}$</td><td>Local and global empirical losses</td></tr>
<tr><td>$\phi_k, \phi$</td><td>Local and global encoders</td></tr>
<tr><td>$\phi_{k,\theta}, \phi_\theta$</td><td>Local and global encoders parametrized by $\theta$</td></tr>
<tr><td>$\phi_{node}, \phi_{edge}, \phi_{graph}$</td><td>Node, edge, and graph encoder functions</td></tr>
<tr><td>$\psi, \Psi$</td><td>Predictor parameters and parameter space</td></tr>
<tr><td>$T, t$</td><td>Number of global iterations and global iteration index</td></tr>
<tr><td>$\theta_k, \theta, \Theta_k, \Theta$</td><td>Local and global model parameters and parameter domains</td></tr>
<tr><td colspan="2" align="center">Privacy and Differential Privacy</td></tr>
<tr><td>$\delta$</td><td>Failure probability in differential privacy</td></tr>
<tr><td>$\varepsilon$</td><td>Privacy budget in differential privacy</td></tr>
<tr><td>$\mathcal{M}$</td><td>Randomized mechanism (training algorithm) for differential privacy</td></tr>
<tr><td>$U$</td><td>Set of possible outcomes of mechanism $\mathcal{M}$</td></tr>
</table>

Table 3: Notations used in this paper, organized by comprehensive categories and alphabetical order.

