# OpenReview forum: "From Euclidean to Graph-Structured Data: A Survey of Collaborative Learning"
_TMLR — Accepted by TMLR_

### Review · Reviewer_CjWc · 2025-12-03

**Summary Of Contributions:**

This survey paper tries to addres the growing need to move beyond centralized machine learning pipelines, which suffer from scalability and privacy limitations, in the context of graph representation learning. While collaborative learning paradigms such as federated and decentralized learning have gained traction, existing research largely focuses on Euclidean data types (e.g., text, images) and overlooks graph-structured data, which are prevalent in many real-world applications. The authors argue that message-passing mechanisms in graph learning naturally align with collaborative settings but remain significantly underexplored. However, they left the Spectral GNN literature unexplored.

The survey first synthesizes foundational principles of collaborative learning on Euclidean data, organizing prior work across three fundamental dimensions: learning effectiveness, computational efficiency, and privacy preservation. It then extends this lens to graph-structured data, proposing a taxonomy for distributed graph scenarios, analyzing the resulting statistical heterogeneities, and providing unified problem definitions and algorithmic frameworks. The paper concludes by outlining open challenges and future research opportunities.

**Additional Comments:**

The paper is well-written however it is missing an important line of GNN research.

**Audience:**

Yes

**Audience Explanation:**

Yes. The paper would be of clear interest to TMLR’s audience—especially researchers working on federated learning, decentralized optimization, graph machine learning, and distributed systems. However, the survey notably omits coverage of spectral GNN architectures, an active area with several recent and influential contributions, and this gap limits the completeness of its perspective.

**Broader Impact Concerns:**

No problem in terms of  the ethical implications of the work.

**Claims And Evidence:**

Yes

**Claims Explanation:**

The authors clearly justify the need for extending collaborative learning from Euclidean to graph-structured data by highlighting the limitations of centralized graph ML and the natural compatibility of GNN message passing with collaborative environments. The paper systematically maps foundational principles from Euclidean collaborative learning to the graph domain and provides a coherent taxonomy, problem formulations, and analyses that reinforce the central claims.

Moreover, the survey presents a clear breakdown of design dimensions—effectiveness, efficiency, and privacy—and discusses how they manifest under both Euclidean and graph-structured settings. This structure supports the authors’ assertion that the field is emerging yet rich, and that graph-based collaborative learning introduces new constraints not present in traditional federated learning.

**Requested Changes:**

Major Revision: The paper overlooks a substantial portion of the message-passing literature, particularly in the area of spectral GNN architectures. Although the authors cite one reference, this significantly understates the depth of existing work. Notable models such as APPNP, GPRGNN, BernNet, ChebNetII, JacobiNet, PolyGCL, ArnoldiGCL, LORA, and related studies—several of which also incorporate contrastive learning—are missing. A thorough review of these spectral approaches is essential for a complete and balanced survey.

In addition, the paper omits important work in Graph Federated Learning, including the ICLR 2025 paper “Subgraph Federated Learning for Local Generalization.” This line of research directly tackles critical challenges such as:

Unseen Nodes – introducing new nodes with known classes that alter local graph structure,

Missing Classes – nodes whose labels have not appeared previously in a client’s graph, and

New Clients – entirely new graphs with distinct structural and label distributions.

These challenges, along with the corresponding references, should be incorporated into the survey to strengthen its completeness and relevance.

---

> ### Author Response · Authors · 2026-01-18
> **Response to reviewer CjWc (Part I)**
>
> We thank the reviewer for the insightful comments! Following them, we have extended the manuscript with a detailed discussion of spectral methods and of the dynamic scenarios. Following the suggested reference, we also included prototype-based solutions in the survey. The changes are marked in blue. Please see our detailed answer below.
>
>
> Comment 1. The paper overlooks a substantial portion of the message-passing literature, particularly in the area of spectral GNN architectures. Although the authors cite one reference, this significantly understates the depth of existing work. Notable models such as APPNP, GPRGNN, BernNet, ChebNetII, JacobiNet, PolyGCL, ArnoldiGCL, LORA, and related studies—several of which also incorporate contrastive learning—are missing. A thorough review of these spectral approaches is essential for a complete and balanced survey.
>
>
> Response to Comment 1.
> We thank the reviewer for this valuable feedback regarding spectral GNN architectures. We acknowledge that our initial treatment of spectral approaches was indeed insufficient for a comprehensive survey.
> We have conducted a thorough review of the spectral GNN literature and discovered relevant work at the intersection of spectral methods and collaborative learning. Specifically, we identified recent papers that successfully combine distributed graph learning with spectral approaches, including FedSSP (Tan et al., 2024) and S2FGL (Tan et al., 2025), as well as a comprehensive benchmark study on spectral GNNs (Liao et al., 2025).
> Regarding the centralized spectral architectures mentioned (APPNP, GPRGNN, BernNet, ChebNetII, JacobiNet), we have added a discussion in Section 3.2 explaining their relationship to message-passing approaches. We note that these methods, while often categorized as "spectral," ultimately reduce to spatial message-passing implementations through polynomial filters in the adjacency matrix. This reduction validates our survey's focus on MPNNs as a unifying framework.
> The works by Tan et al. (2024, 2025) also revealed a gap in our coverage of prototype-based techniques, which we have now addressed in Section 2.3. Regarding LoRA, we have added two new reference papers on LoRA in federated settings in Section 2.3.2 "Local adaptation." However, for graph-structured data, to the best of our knowledge, no application of LoRA has been considered in the literature.
>
>
> Changes made:
>
> Section 3.1: Added discussion of spectral GNN approaches and their relationship to spatial message-passing. Also extended discussion on GNN training using contrastive techniques.
>
> Section 3.4.1: Integrated FedSSP and S2FGL references in data augmentation and local regularization techniques
> Section 2.3.1: Added considerations on prototype-based techniques
>
> Section 2.3.2 (Knowledge distillation): Complemented with prototype-based approaches
>
> Section 2.3.2 (Local adaptation): Added two LoRA references for federated settings
>
> Table 2: Added FedSSP and S2FGL to the comparative table
>
> Bibliography: Added all mentioned references

---

> ### Author Response · Authors · 2026-01-18
> **Response to reviewer CjWc (part II)**
>
> Comment 2:
> In addition, the paper omits important work in Graph Federated Learning, including the ICLR 2025 paper “Subgraph Federated Learning for Local Generalization.” This line of research directly tackles critical challenges such as:
>
> Unseen Nodes – introducing new nodes with known classes that alter local graph structure,
>
> Missing Classes – nodes whose labels have not appeared previously in a client’s graph, and
>
> New Clients – entirely new graphs with distinct structural and label distributions.
>
> These challenges, along with the corresponding references, should be incorporated into the survey to strengthen its completeness and relevance.
>
>
> Response to Comment 2.
> We thank the reviewer for highlighting this important gap. We have now integrated "Subgraph Federated Learning for Local Generalization" (Kim et al., 2025), and included the challenges of a dynamic agent population.
>
> While our taxonomy of distributional shifts (Section 3.3.2) could theoretically encompass scenarios involving unseen nodes, missing classes, and new clients entering the system, we now make this link more explicit in this section.  We have also clarified how dynamic populations interact with system constraints under graph-structured data (Section 3.3.3). These modifications for graph-structured data have also prompted revisions for the respective sections for Euclidean data (Section 2.2.2 and Section 2.2.3)
>
> Changes made:
> Section 2.2.2: A new sentence is added to clarify the impact of new agents.
>
> Section 2.2.3: The part on device and network constraints is changed significantly, and the requirements to handle dynamic populations are added.
>
> Section 2.3.1: Data-based techniques now include prototype-based regularization with new references that Kim et al. 2025 also build on.
>
> Section 2.3.2: Under knowledge distillation, we discuss prototype-based solutions that Kim et al. 2025 also built on.
>
> Section 3.3.2: We added discusson on dynamic population under structure shift, and a discussion on characterizing shifts in evolving graphs.
>
> Section 3.3.3: The part on device and network constraints is entirely rewritten to connect system constraints with collaborative inference and explicitly link dynamic populations to challenges from Section 3.3.2. Dynamic graphs are explicitly mentioned as a primary challenge.
>
> Section 3.4.1 and Table 2: Kim et al. (2025) are cited under local regularization.

---

### Review · Reviewer_D1d5 · 2025-12-15

**Summary Of Contributions:**

This paper is a survey of Collaborative Learning on graph data. It presents Collaborative Learning (CL) as a Machine Learning problem in which independent agents want to use learned representations of other agents to improve theirs, without sharing data.

After presenting the canonical CL algorithm, Federated Averaging, they introduce the main challenge of CL, that is, the different data distribution scenarios across the learning agents. The authors guide us through the literature by explaining how different methods try to improve the learning effectiveness, efficiency and privacy of CL in the different agent data distribution scenarios.

**Strengths**
- Introducing CL using the FedAvg algorithm and then exposing its flaws is a very nice ease-in introduction.
- The categorization of the different data partitions allows to go beyond the classical Centralized, Federated Learning, Decentralized Learning categorization of Collaborative Learning.
- For each goal (effectiveness, efficiency and privacy), the authors suggest clear categorizations of the main directions in the literature.
- By presenting the euclidean CL literature with the same structure as the graph CL literature (data scenarios then classification of methods based on their goals), the authors highlight similarities between graph and euclidean CL.

**Weaknesses**
- The main motivation of CL according to the authors, i.e. using other agents knowledge without sharing data, is under-studied in the survey. Agent’s privacy is presented as an enhancement target, not as a fundamental tradeoff of knowledge exchange between the agents. The authors do mention that collaborative learning is not private, but, to a CL newcomer, the survey overall does not make it clear that the absence of data transfer does not imply good privacy. This is exacerbated by the relatively small size of the subsections on privacy enhancements compared to the effectiveness and efficiency subsections.
- While the categorization of CL methods and their goals is clear, there is little discussion on the tradeoffs (or lack thereof) they imply.
- The separation of the survey into a section on euclidean data and graph data hurts the unification claim (see the claims section) and the presentation. Since section 2 acts as an introduction to the survey structure (data scenarios, then effectiveness/efficiency/privacy goals), section 3 appears as a litany of methods, stripped of motivations and comparisons (because they were already made in section 2), which makes the reading lengthy.

**Additional Comments:**

- Categorizing frozen layers (model decoupling item p.16) into the term architecture personalization (section 2.3.2) is strange, especially since the previous paragraph is exactly about parameter personalization.
-  “Cluster-based” section 2.3.2. It is unclear what is clustered. I guess this is parameter clustering, but then what is the difference with similarity-based aggregation ?
- “in practice, agents might obtain a more effective model by collaborating under the vertical partitioning described in Section 2.2.1.”  p. 16. is the partitioning a choice of the agents ?
- Definition of the local datasets, p18: Isn’t x_k^i in X_k instead of X?
- The justification of training under vertical partition is strange: we do not want to share the data, have no label but still want to do something. Wouldn’t it be easier to sign a data sharing (w/ protection guarantees) agreement instead of implementing all of this? Asked differently, In what situation is data strictly forbidden to be shared but sharing features OK?

- Description of the two GNN paradigms p.28: the distinction between spectral and spacial filters is unclear as spectral and spacial filter as not defined.
- Strange artifact on figure 11: half a node is visible on the right
- A bit of history on the communities that worked on collaborative learning on graph would have help to better understand the context.

**Audience:**

Yes

**Audience Explanation:**

Yes, it is a thorough review of collaborative learning on graph data. The decomposition of the different data partitioning brings clarity in the challenges of collaborative learning.

**Claims And Evidence:**

No

**Claims Explanation:**

The paper is presented as a survey and makes the following claims

**Claim 1: As a survey, the paper is a pedagogical starting point to the field of collaborative learning on graphs.** Yes
- See strengths section above


**Claim 2: As a survey, it depicts a comprehensive picture of the field of collaborative learning on graphs.**  Yes
- Each of the presented research direction points to several references to learn more about it
- The table 2 is very clear, useful to follow the graph CL part of the survey and a good overview of the graph CL survey.


**Claim 3: The survey collaborative learning on euclidean data and graph data.** No
- The claim is, in my opinion, not supported enough.
- The common structure between section 2 (euclidean CL) and section 3 (graph CL) highlights the similarities between the challenges of euclidean and graph CL (effectiveness, efficiency and privacy).
- Section 3.3 highlight the greater complexity of graph data scenarios compared to euclidean data (more ways to partition the data among agents).
- But, in sections 3.4, 3.5 and 3.6, there is almost no mention of similarity, differences or unification of graph and euclidean CL methods.
- While the introduction of CL using FedAvg and its shortcomings is very pedagogical, the rest of section 2 is not re-used to understand the differences between graph and euclidean CL in section 3.


**Claim 4: Clearly presents the main open problems and promising directions in graph CL** No
- The open challenges in 3.4.1 are not motivated enough. E.g. what makes graph-specific regularization different from euclidean data?
- The open challenges on 3.4.2 are: “we need to adapt what has been done in euclidean data (personalization, foundation models, …) to graph data”. But again does not say what is different/similar between graph and euclidean CL.
- The open challenges in 3.4.3 are “towards more advanced methods”,  there is no insights on why this is necessary nor how this would be different from euclidean data

**Requested Changes:**

Typos
- “l(Fθk(t− 1 2 )(x), y”  p. 8 missing parenthesis
- “horisontal” p. 24 typo
- “decentrized”  p. 45 decentralized ?

---

> ### Author Response · Authors · 2026-01-18
> **Response to reviewer D1d5 (part I)**
>
> We thank the reviewer for the thorough review! We addressed all the comments in the manuscript; the changes are marked in blue. The most significant changes are how we discuss privacy, the methods for graph-based learning compared to those for Euclidean data, and the consistent discussion of the efficiency, effectiveness, and privacy trilemma.
> Please see our detailed answer below.
>
> Weaknesses:
>
> The main motivation of CL, according to the authors, i.e, using other agents knowledge without sharing data, is under-studied in the survey. Agent’s privacy is presented as an enhancement target, not as a fundamental tradeoff of knowledge exchange between the agents. The authors do mention that collaborative learning is not private, but, to a CL newcomer, the survey overall does not make it clear that the absence of data transfer does not imply good privacy. This is exacerbated by the relatively small size of the subsections on privacy enhancements compared to the effectiveness and efficiency subsections.
>
> Response.
> We thank the reviewer for pointing out that privacy was not adequately addressed in the survey. We now state clearly that federated learning itself does not guarantee privacy and discuss privacy vulnerabilities as a challenge both for Euclidean and graph-based data. We also present a more detailed discussion of privacy enhancements. In the graph part, we now focus more on graph-specific vulnerabilities. Finally, the trilemma of efficiency, effectiveness, and privacy is discussed in each group of methods.
>
> Changes made:
> Section 1: Small changes are made to avoid the impression that collaborative learning is private, per se.
> Section 2.2.4: A new section is added, discussing privacy vulnerabilities in collaborative learning with Euclidean data, with additional references.
>  Section 2.5: We extended the description of differential privacy techniques.
> Section 3.3.4: A new section is added, discussing privacy vulnerabilities in distributed GNN, with new references.
> Section 3.6: The description of privacy-preserving techniques is improved by adding shorter explanations for graph-specific techniques.
> After all sub-sub section: A take-home box is added that specifically discusses the trilemma.
>
>
> While the categorization of CL methods and their goals is clear, there is little discussion on the tradeoffs (or lack thereof) they imply.
>
>
> Response. We agree that the trilemma of efficiency, effectiveness, and privacy was not addressed in detail in the manuscript. The tradeoffs are often the same for several methods. Therefore, we decided to discuss them at the end of the method subsections, so that the discussions can be specific enough without being too repetitive. We also give a longer explanation of the trilemma at its introduction.
>
> Changes made:
> End of Section 2.2: The description of the collaborative trilemma is extended.
> After all sub-sub section: A take-home box is added that specifically discusses the trilemma.

---

> ### Author Response · Authors · 2026-01-18
> **Response to reviewer D1d5 (part II)**
>
> The separation of the survey into a section on Euclidean data and graph data hurts the unification claim (see the claims section) and the presentation. Since section 2 acts as an introduction to the survey structure (data scenarios, then effectiveness/efficiency/privacy goals), section 3 appears as a litany of methods, stripped of motivations and comparisons (because they were already made in section 2), which makes the reading lengthy.
>
>
> Response.
> We thank the reviewer for this important observation about the presentation of our unification claim. We acknowledge that sections 3.4, 3.5, and 3.6 initially lacked explicit connections between Euclidean and graph CL methods.
> We have substantially revised these sections to make the similarities, differences, and extensions between Euclidean and graph methods more explicit, through more detailed introductory paragraphs and some small clarifications in the description of the methods at the item level. Throughout Section 3, we now systematically: (1) reference which Euclidean techniques from Section 2 are being extended, (2) explain what is transferred and what requires graph-specific modifications, (3) identify which data partition cases are addressed and why, and (4) highlight new graph-specific aspects.
>
> Changes made:
> Section 3.4.1: We added explicit connections to Euclidean methods, explaining what results transfer and what is graph-specific for each technique.
> Throughout Sections 3.4, 3.5, 3.6: We added comparative statements (e.g. "Extend ...") to make relationships explicit. Also, we rewrote most introductory paragraphs to make the graph-specific challenges or possibilities clear.
> We introduced summary boxes at the end of each method part, where the connection with principles of Euclidean CL is also taken up.
> Open challenges subsections of Sections 3.4, 3.5, 3.6: Enhanced motivation explaining what the current state of solutions is, why it is insufficient, why challenges are different/more difficult for graph data compared to the Euclidean case, and for collaborative implementations compared to centralized solutions.
>
>
> Claim 3: The survey collaborative learning on euclidean data and graph data. No
>
> The claim is, in my opinion, not supported enough.
> The common structure between section 2 (euclidean CL) and section 3 (graph CL) highlights the similarities between the challenges of euclidean and graph CL (effectiveness, efficiency and privacy).
> Section 3.3 highlight the greater complexity of graph data scenarios compared to euclidean data (more ways to partition the data among agents).
> But, in sections 3.4, 3.5 and 3.6, there is almost no mention of similarity, differences or unification of graph and euclidean CL methods.
> While the introduction of CL using FedAvg and its shortcomings is very pedagogical, the rest of section 2 is not reused to understand the differences between graph and euclidean CL in section 3.
>
>
> Response.
> Please see the answer to the last weakness above.
>
>
> Claim 4: Clearly presents the main open problems and promising directions in graph CL No
>
> The open challenges in 3.4.1 are not motivated enough. E.g. what makes graph-specific regularization different from euclidean data?
> The open challenges on 3.4.2 are: “we need to adapt what has been done in euclidean data (personalization, foundation models, …) to graph data”. But again does not say what is different/similar between graph and euclidean CL.
> The open challenges in 3.4.3 are “towards more advanced methods”,  there is no insights on why this is necessary nor how this would be different from euclidean data
>
>
> Response.
> We thank the reviewer for pointing out that our open challenges were insufficiently motivated. We have significantly strengthened the justification for each open challenge.
> For Section 3.4.1, we now explain what makes graph-specific regularization fundamentally different from Euclidean data. For Section 3.4.2, we clarify why adapting Euclidean techniques is non-trivial for graphs and what specific challenges arise.
> For Section 3.4.3, we provide clearer insights on why more advanced methods are necessary and how graph structure creates unique requirements.
>
> Changes made:
> Section 3.4.1 open challenges: We reformulated most of the challenges and added motivations for their importance.
> Section 3.4.2 open challenges: We reformulated the challenges and discussed how new challenges emerge from developments for centralized GNNs and form general developments in ML architectures.
> Section 3.4.3 open challenges: We improved the justifications and consolidated the challenges.
> All open challenge paragraphs of Sections 3.4, 3.5, 3.6: We added an introductory paragraph explaining how the collaborative settings exacerbate challenges of centralized inference and training.

---

> ### Author Response · Authors · 2026-01-18
> **Response to reviewer D1d5 (part III)**
>
> Minor comments
>
>
> Categorizing frozen layers (model decoupling item p.16) into the term architecture personalization (section 2.3.2) is strange, especially since the previous paragraph is exactly about parameter personalization.
>
>
> Response. Thank you for this comment. We want to clarify that model decoupling does not involve frozen layers, as both shared and private components are updated during local training. What makes this architecture personalization rather than parameter personalization is that the private component can be of a different architecture in the generic case, for example, by using a deeper private component.
>
> Change made: added a clarifying sentence at the beginning of the model decoupling item (Section 2.3.2).
>
> “Cluster-based” section 2.3.2. It is unclear what is clustered. I guess this is parameter clustering, but then what is the difference with similarity-based aggregation ?
>
> Response. We thank the reviewer for this need for clarification. We have added some clarification sentences on similarity and cluster-based techniques in 2.3.2.
>
>
> “in practice, agents might obtain a more effective model by collaborating under the vertical partitioning described in Section 2.2.1.” p. 16. is the partitioning a choice of the agents?
>
> Response. In Sections 2.3.3 and 3.4.3 we have improved the introductory paragraphs to explain how the partitioning emerges and why collaboration is needed. Examples for vertical partitioning are given in Section 2.2.1. and 3.3.1. Please note that in most of the methods, data or features are not shared, only embeddings.
>
>  Definition of the local datasets, p18: Isn’t $x_k^i \in X_k$ instead of X?
>
> Response. We have fixed this typo thanks to this careful observation.
>
>
> The justification of training under vertical partition is strange: we do not want to share the data, have no label, but still want to do something. Wouldn’t it be easier to sign a data sharing (w/ protection guarantees) agreement instead of implementing all of this? Asked differently, in what situation is data strictly forbidden to be shared, but sharing features OK?
>
> Response. Please see our answer to your minor comment 3 above.
>
>
> Description of the two GNN paradigms p.28: the distinction between spectral and spatial filters is unclear, as spectral and spacial filter as
> not defined.
>
> Response.
> We significantly extended the discussion of spectral vs. spatial methods. Please see our answer to Review CjWc Comment 1.
>
>
> Strange artifact on figure 11: half a node is visible on the right
>
> Response. Thank you for this careful observation. We have fixed it.
>
>
> A bit of history on the communities that worked on collaborative learning on the graph would have helped to better understand the context.
>
> Response. We have added details on different industrial and scientific communities responsible for the three pillars and CL on graph data in Section 1.1.

---

> ### Comment · Reviewer_D1d5 · 2026-02-16
>
> Dear Authors, thank you for your thorough answers and clarifications.
> The implemented changes addressed my concerns.

---

### Review · Reviewer_Zpd3 · 2025-12-21

**Summary Of Contributions:**

The paper surveys collaborative learning (CL), with a particular focus on extending federated & decentralized learning from Euclidean data to graph settings. The paper's core contribution is a unification of CL, graph representation learning, and diffusion, along with a taxonomy of graph distribution & partition scenarios. The survey also identifies some open challenges related to efficiency, privacy, and inference in collaborative graph learning.

Despite weaknesses in organization and presentation, I believe the technical content is thorough, timely, and well researched.

**Additional Comments:**

The submission contains a large amount of high-quality & well-researched material, and reflects significant effort by the authors for which they should be commended. However, as outlined earlier, the organization & presentation limit the accessibility and impact of the survey. The current manuscript reads more like a set of comprehensive curated notes than a guided survey. Importantly, these are structural and editorial issues rather than conceptual: the underlying content is largely strong! The suggested modifications would likely resolve these issues.

For these reasons, I view the paper in its current form as lacking in clarity of presentation, but well-positioned to become a strong and valuable survey after targeted revisions.

**Audience:**

Yes

**Audience Explanation:**

The survey is comprehensive, and can be useful to researchers in multiple domains including federated learning, Graph ML, and even in privacy and security.

**Broader Impact Concerns:**

The main concern is potential over-interpretation of privacy benefits in collaborative learning settings. Clearer language around threat models and limitations would mitigate this risk. Additionally, since the paper discusses attacks in detail, it can be interpreted to inadvertently provide a detailed roadmap for adversarial behavior.

Beyond this, the work primarily serves a clarifying and organizational role and does not raise significant broader-impact concerns.

**Claims And Evidence:**

No

**Claims Explanation:**

The authors present their claims in a thorough, detailed survey with over 400 references, and they are well-supported with theory and pseudocode.

While the survey is technically sound, I strongly believe the clarity with which this evidence is synthesized and presented should be improved. In particular, organizational and presentation issues sometimes obscure otherwise well-supported arguments. I should emphasize that these concerns are largely editorial rather than indicative of missing or incorrect evidence.

The modifications which can help are outlined in the "Requested Changes" section below.

**Requested Changes:**

As mentioned earlier, the survey is well researched. Below are some changes to enhance the overall readability and completeness.

Content:

- [C1] Wouldn't there be memory bottleneck issues with GNNs, especially when storing multi-hop neighbor embeddings? It would help to see some discussion in this regard
- [C2] Section 3.6 privacy is very useful, but as with Euclidean FL, in a collaborative graph learning setting there can be a Byzantine agent who can inject adversarial edges that corrupt the embeddings of honest agents multi-hops away during message passing phase. To ensure a complete survey, it might be helpful to call out some topological defense mechanisms, ie how can an agent or server verify the structural integrity of a neighbor's sub-graph without violating privacy?
- [C3] While the choice of picking only supervised learning is fair, it would help to share any references for unsupervised or self-supervised learning
- [C4] (good to have) I believe the survey would benefit from a brief discussion on how graph density affects the communication complexity of Algorithm 3. Are there graph topologies where a single agent can become a communication bottleneck?
- [C5] (good to have) CL assumes the agents are willing to participate, and one agent's privacy can be compromised simply because their neighbor reveals a connection. There are also agents with critical/influential nodes (high centrality) while others can hold isolated nodes. This brings a challenge of "data valuation" in graphs (and with it the whole field of game theory, but that's definitely out of scope!). How can we reward agents who contribute critical topological bridges vs those who only provide redundant local clusters. This might help in linking "Effectiveness" with "System heterogeneity" of agent contributions (section 3.3.3).
- [C6] (good to have) I would clarify some language around privacy guarantees from data locality. These can be tightened to clearly distinguish architectural choices from formal privacy guarantees. The author's call for tighter integration with defense mechanisms, while responsible and necessary, should be emphasized even more strongly in the conclusion.

Editorial:

- [E1] Optimize reference management
    - [E1.1] With over 400 citations, more selective synthesis focusing on representative or contrasting works rather than exhaustive enumeration would significantly improve clarity and reduce cognitive load for readers. Table 1 is very useful in picking the top papers in a particlular field. However, I'd recommend using distinct markers to identify the pillar papers that define a field or area of study when cited in-line (like McMahan et al 2017, or Kipf & Welling 2017).  There are also many places where the references are just listed one after another for a given area (for example, in page 36 or 38), it would help to also highlight if these references also differ in some sense.
    - [E1.2] (good to have) As the authors plan to release resources, I suggest organizing the bibliography into a searchable format split into categories & use cases. There are many ways to do this, one way I can think of is having a timeline, but am open to thoughts from the authors.

- [E2] Presentation:
    - [E2.1] Heavy reliance on long, nested bullet points for complex taxonomies can make it difficult to follow at times. This substitutes for narrative structure and obscures relationships between ideas. Bullets should be reserved for summaries or compact enumerations following a clearly established framework; core concepts would benefit from being presented in prose, supported by figures or tables. It also would help to add a brief "Summary of insights" or "Key takeaways" box at the end of major sections to distill which techniques are best suited for specific cases
    - [E2.2] Given the length of the survey, while Figure 3 helps in giving an overview, a "decision tree" style branching with all the subsections listed can help readers navigate the paper better. I think this also ties back to the bullet lists, doing some reorg can help improve the overall readability. Even a detailed table of contents can help!
    - [E2.3] Table 2 is very useful and anchors the survey well, but is again tough to follow since all the references are bunched into the space and a lot more space is unused. One thought is to split it into multiple tables (probably one for each section?) and use highlights to let the readers know which papers are seminal in the area. I'm also open to hearing thoughts from the authors.

---
While all the points above would strengthen the submission, I think addressing points C1-3, E1.1 and E2 would make the paper a stronger candidate for recommendation.

---

> ### Author Response · Authors · 2026-01-18
> **Response to reviewer Zpd3 (part I)**
>
> We thank the reviewer for the thorough review! We have addressed all the comments, and improved both the content and the presentation. The most important changes are the following. To make the survey more accessible, we added key insights boxes, that summarize the methodologies in an unified way. We added further discussions of privacy challenges and solutions. Based on the reviewer's comments, we also emphasized the graph-topology related challenges and methodologies. The changes in the manuscript are marked in blue. Please see our detailed answer below.
>
>
> [C1] Wouldn't there be memory bottleneck issues with GNNs, especially when storing multi-hop neighbor embeddings? It would help to see some discussion in this regard
>
> Response. We thank the reviewer for this observation. We have modified the system challenges and added a discussion of memory issues in both the Euclidean part (Section 2.2.3) and in the GNN part, including a discussion on multi-hop aggregation (Section 3.3.3).
>
> Changes made:
>
> Section 2.2.3: Added memory constraints discussion
> Section 3.3.3: Reworked the part to include an extensive discussion on how the graph topology can indeed lead to communication bottlenecks in the form of stragglers.
>
>
> [C2] Section 3.6 privacy is very useful, but as with Euclidean FL, in a collaborative graph learning setting, there can be a Byzantine agent who can inject adversarial edges that corrupt the embeddings of honest agents multi-hops away duringthe message passing phase. To ensure a complete survey, it might be helpful to call out some topological defense mechanisms, i.e., how can an agent or server verify the structural integrity of a neighbor's sub-graph without violating privacy?
>
> Response. We thank the reviewer for this important observation about Byzantine agents and adversarial topology manipulation. We have clarified our threat model and scope, stating that we address the case of honest-but-curious agents. However, thanks to your comment, we discuss privacy vulnerabilities in more detail in Section 3.3.4 and related privacy preservation methods in Section 3.6.
>
> Changes made:
> Section 3.3.4: Created a new section to describe privacy vulnerabilities. Added a paragraph explicitly distinguishing honest-but-curious agents (our focus) from malicious Byzantine agents and acknowledged adversarial topology attacks but clarified they fall outside our survey's scope.
> Section 3.6 open challenges: Clarified the direction of verifiable mechanisms in "tighter integration with defense mechanisms".
> Added take-home boxes.
>
> [C3] While the choice of picking only supervised learning is fair, it would help to share any references for unsupervised or self-supervised learning.
>
> Response. We thank the reviewer for this comment. We have now added in 1.1  relevant reviews focusing on self-supervised and unsupervised approaches.
>
> Changes made:
> Introduction: Added references to federated unsupervised and reinforcement learning work.
> Section 3.1: Added a note that GNNs are often trained with self-supervision in practice.
>
>
> [C4] (good to have) I believe the survey would benefit from a brief discussion on how graph density affects the communication complexity of Algorithm 3. Are there graph topologies where a single agent can become a communication bottleneck?
>
> Response. We have reworked Section 3.3.3 on system heterogeneity to connect the impact of the topology and its partitioning on the memory and communication heterogeneities.
>
> Changes made:
> Section 3.3.3: Added discussion of how graph topology and density affect communication complexity, and identified scenarios where single agents become bottlenecks.
>
>
> [C5] (good to have) CL assumes the agents are willing to participate, and one agent's privacy can be compromised simply because their neighbor reveals a connection. There are also agents with critical/influential nodes (high centrality) while others can hold isolated nodes. This brings a challenge of "data valuation" in graphs (and with it the whole field of game theory, but that's definitely out of scope!). How can we reward agents who contribute critical topological bridges vs those who only provide redundant local clusters. This might help in linking "Effectiveness" with "System heterogeneity" of agent contributions (section 3.3.3).
>
> Response. We agree that valuation and incentives are important aspects of collaborative systems, and data valuation based on node centrality and topological contributionbut is an exciting topic. However, we consider this out of scope for the survey.
> However, influenced by your comment, we now discuss the centrality of nodes in Section 3.3.2 regarding distributional shifts.
>
> Change made:
> Section 3.3.2: Added discussion of node centrality in the context of distributional shifts.

---

> ### Author Response · Authors · 2026-01-18
> **Response to reviewer Zpd3 (part II)**
>
> [C6] (good to have) I would clarify some language around privacy guarantees from data locality. These can be tightened to clearly distinguish architectural choices from formal privacy guarantees. The author's call for tighter integration with defense mechanisms, while responsible and necessary, should be emphasized even more strongly in the conclusion.
>
> Response. We have added new sections (2.2.4 and 3.3.4) insisting on the possible attacks despite locality. We have also added take-home boxes at the end of 2.5 and 3.6, insisting on the distinction between privacy due to the data being local, and formal privacy guarantees.
>
> Changes made:
> Section 1: Small changes made to avoid the impression that collaborative learning is private per se.
> Section 2.2.4: New section added, discussing privacy vulnerabilities in collaborative learning with Euclidean data, with additional references.
> Section 2.5: Extended the description of differential privacy techniques.
> Section 3.3.4: New section added, discussing privacy vulnerabilities in distributed GNN, with new references.
> Section 3.6: Description of privacy-preserving techniques improved by adding shorter explanations for graph-specific techniques and a more detailed discussion of security requirements.
> After all sub-subsections: A take-home box was added that specifically discusses the trilemma.
>
> Editorial:
>
> [E1.1] With over 400 citations, more selective synthesis focusing on representative or contrasting works rather than exhaustive enumeration would significantly improve clarity and reduce cognitive load for readers. Table 1 is very useful in picking the top papers in a particlular field. However, I'd recommend using distinct markers to identify the pillar papers that define a field or area of study when cited in-line (like McMahan et al 2017, or Kipf \& Welling 2017). There are also many places where the references are just listed one after another for a given area (for example, in page 36 or 38), it would help to also highlight if these references also differ in some sense.
>
> Response. In the text, we divide the group of references up and explain them separately whenever more than four references appear.
>
> Changes made:
> Table 1: Added markers distinguishing surveys and seminal works.
> Table 2: Marked key papers in bold when more than four references appear.
> Throughout text: Divided reference groups and explained them separately when more than four references appear.
>
>
> [E1.2] (good to have) As the authors plan to release resources, I suggest organizing the bibliography into a searchable format, split into categories and use cases. There are many ways to do this, one way I can think of is having a timeline, but am open to thoughts from the authors.
>
> Response. We have added as supplementary material the resources that we will release on GitHub, listing the references in Table 2 along the two axes of the table.
>
> Changes made:
> Added as attachement to the submission, the structured bibliography organized by data partition and design choices.
>
>
> [E2.1] Heavy reliance on long, nested bullet points for complex taxonomies can make it difficult to follow at times. This substitutes for narrative structure and obscures relationships between ideas. Bullets should be reserved for summaries or compact enumerations following a clearly established framework; core concepts would benefit from being presented in prose, supported by figures or tables. It also would help to add a brief "Summary of insights" or "Key takeaways" box at the end of major sections to distill which techniques are best suited for specific cases
>
> Response. We thank the reviewer for the suggestion to include brief summaries. We have added summary boxes at the end of the method sections (Section 2.3.1, 2.3.1 and so on) to synthesize takeaways. These seem to be the right places to give insights in a short format. Specifically, in each summary box we list the key techniques, the challenges they address, and discuss the trilemma of effective, efficient, and privacy-preserving learning and inference. In the graph-data par,t future research directions are also included in the summary boxes.
>
> We acknowledge the heavy reliance on bullet points. However, we believe that this is the best solution to have text with hierarchy below subsections. We reviewed the structure of the paper and reorganized or subdivided long bullet lists. We also added introduction paragraphs that motivate the lists.
>
> Changes made:
> Throughout: Introduced key insight boxes at the end of major sections, each following the same structure.
> Throughout: Improved and extended the introduction paragraphs.
> Section 3.4.1: Cut the bullet list for data augmentation into two.

---

> ### Author Response · Authors · 2026-01-18
> **Response to reviewer Zpd3 (part III)**
>
> [E2.2] Given the length of the survey, while Figure 3 helps in giving an overview, a "decision tree" style branching with all the subsections listed can help readers navigate the paper better. I think this also ties back to the bullet lists, doing some reorg can help improve the overall readability. Even a detailed table of contents can help!
>
> Response. We have added this decision branching in Appendix A.
>
> Changes made:
> Beginning of each n.n level section: Added list of n.n.n level subsections with brief descriptions.
>
>
>  [E2.3] Table 2 is very useful and anchors the survey well, but is again tough to follow since all the references are bunched into the space, and a lot more space is unused. One thought is to split it into multiple tables (probably one for each section?) and use highlights to let the readers know which papers are seminal in the area. I'm also open to hearing thoughts from the authors.
>
> Response. We are happy that the reviewer finds Table 2 useful. To ensure that the readers return to the table, we discuss its content now at the beginning of Sections 3.4, 3.5, and 3.6, as well as in some of the subsections. However, we have preserved the unified Table 2 structure for two reasons. First, we believe that it gives a nice picture of what methodologies were addressed for the different data partitioning scenarios, and where the undiscovered topics are. The second reason is practical, the header is a significant part of the table, and we do not want to repeat that. However, to ensure that the table is used, we refer back and comment on the content at the beginning of the effectiveness, efficiency, and privacy sections.
>
> Changes made:
> Table 2: Marked seminal papers in bold when cells contain 4+ references.
> Beginning of subsections: Discuss which methodologies are considered in research contributions, based on the content of Table 2.

---

> ### Comment · Reviewer_Zpd3 · 2026-01-21
> **Updates**
>
> I commend the authors for their thoughtful and detailed response to all the reviews, and for the substantive improvements made to the manuscript. The revisions address my prior concerns, and I believe the paper is now well above the threshold for acceptance.

---

### Decision · Action_Editor_3RAt · 2026-02-17

**Recommendation:** Accept with minor revision

**Audience:**

Yes

**Audience Explanation:**

The survey might be of interest to researchers in federated learning, Graph ML, and even in privacy and security.

**Claims And Evidence:**

Yes

**Claims Explanation:**

All reviewers are in consensus that the review is well built and provides a thorough description of collaborative learning. All reviewers agree that the rebuttal and the changes to the original submission address the issues pointed out in the reviews.
Having said that, reviewer CjWc feels that the coverage of relevant spectral approaches is still lacking.
I request the authors to expand this coverage accordingly.

---

> ### Author Response · Authors · 2026-02-23
> **Response to recommendation with minor revision**
>
> Thank you for the quick recommendation. We believe that the minor revision, requesting the coverage of relevant spectral approaches, has been addressed in our initial answer to reviewer CjWc. Please see our "Response to reviewer CjWc (Part I)" below. To summarize:
>
> We have conducted a thorough review of the spectral GNN literature and discovered relevant work at the intersection of spectral methods and collaborative learning. Specifically, we identified recent papers that successfully combine distributed graph learning with spectral approaches, including FedSSP (Tan et al., 2024) and S2FGL (Tan et al., 2025), as well as a comprehensive benchmark study on spectral GNNs (Liao et al., 2025). Regarding the centralized spectral architectures mentioned (APPNP, GPRGNN, BernNet, ChebNetII, JacobiNet), we have added a discussion in Section 3.2 explaining their relationship to message-passing approaches. We note that these methods, while often categorized as "spectral," ultimately reduce to spatial message-passing implementations through polynomial filters in the adjacency matrix. This reduction validates our survey's focus on MPNNs as a unifying framework.